# Near Optimal Non-asymptotic Sample Complexity of 1-Identification

**Zitian Li** [1]   **Wang Chi Cheung** [1]

## Abstract

Motivated by an open direction in existing literature, we study the 1-identification problem, a fundamental multi-armed bandit formulation on pure exploration. The goal is to determine whether there exists an arm whose mean reward is at least a known threshold $\mu_0$, or to output None if it believes such an arm does not exist. The agent needs to guarantee its output is correct with probability at least $1 - \delta$. (Degenne & Koolen, 2019) has established the asymptotically tight sample complexity for the 1-identification problem, but they commented that the non-asymptotic analysis remains unclear. We design a new algorithm Sequential-Exploration-Exploitation (SEE), and conduct theoretical analysis from the non-asymptotic perspective. Novel to the literature, we achieve near optimality, in the sense of matching upper and lower bounds on the pulling complexity. The gap between the upper and lower bounds is up to a polynomial logarithmic factor. The numerical result also indicates the effectiveness of our algorithm, compared to existing benchmarks.

## 1. Introduction

The 1-identification problem is a fundamental multi-armed bandit formulation on pure exploration. The goal of the learning agent is to identify an arm whose mean reward is at least a known threshold $\mu_0$ if such an arm exists, and otherwise to return None if no such arm exists. We study the fixed confidence setting, where the agent aims to return the correct answer with probability at least $1 - \delta$ for a given tolerance parameter $\delta \in (0, 1)$, while ensuring the number of arm pulls, aka sample complexity, as small as possible.

The 1-identification problem models numerous real-world

problems. For example, consider a firm experimenting multiple new campaigns, and seeks to know if any new campaign is more effective than the existing one. The firm may have a long history of applying the existing campaign, with sufficient data to determine the impact and reward by utilizing this campaign. Similar scenario applies to other industries such as service operations, pharmaceutical tests, simulation which involves comparisons between a benchmark and alternatives, in terms of profit, welfare or other metrics.

**Summary of Contributions.** We make two contributions to the 1-identification problem. Firstly, existing research works only guarantee asymptotic optimality (Degenne & Koolen, 2019; Jourdan & Réda, 2023), or non-asymptotic near optimality in the case where all the mean rewards are smaller than the threshold ((Jourdan & Réda, 2023), or applying a Best Arm Identification algorithm). Novel to the literature, our proposed algorithm achieves the non-asymptotic optimality in the sample complexity both the positive case when there is a qualified arm, i.e. an arm with mean reward at least the threshold, and the negative case when there is no qualified arm. We prove matching upper and lower sample complexity bounds, and the gap between these upper and lower bounds is up to a polynomial logarithm factor.

Secondly, we conduct numeric experiments to compare the performance of 1-identification algorithms. The numeric results suggest the excellency of our proposed SEE algorithm and also the weakness of some benchmark algorithms.

**Notation.** For an integer $K > 0$, denote $[K] = \{1, \ldots, K\}$. For $\mu \in [0, 1]$, we denote $N(\mu, \sigma^2)$ as the normal distribution with mean $\mu$ and variance $\sigma^2$. Denote $\mathbb{E}$ and $\mathbb{E}_{\nu,\mathsf{alg}}$ as the expectation operator while the second one is to highlight that the expectation is determined by both the instance $\nu$ and algorithm $\mathsf{alg}$.

## 2. Model

An instance of 1-identification is specified by the tuple $Q = ([K], \nu = \{\nu_a\}_{a \in [K]}, \mu_0, \delta)$. The set $[K]$ represents the collection of $K$ arms. For each $a \in [K]$, $\nu_a$ is the probability distribution of the reward received by pulling arm $a$ once. The probability distribution $\nu_a$, and in particular its mean

---

[1]Department of Industrial Systems Engineering & Management, National University of Singapore, Singapore. Correspondence to: Zitian Li <lizitian@u.nus.edu>, Wang Chi Cheung <isecwc@nus.edu.sg>.

*Proceedings of the 42nd International Conference on Machine Learning*, Vancouver, Canada. PMLR 267, 2025. Copyright 2025 by the author(s).

$\mu_a := \mathbb{E}_{R \sim \nu_a} R$, are not known to the agent. For each arm $a \in [K]$, which has random reward $R_a \sim \nu_a$, we assume that the random noise $R_a - \mu_a$ is 1-sub-Gaussian. The parameter $\mu_0$ is a known threshold. The agent's main goal is to ascertain whether there is an arm $a \in [K]$ such that $\mu_a \geq \mu_0$. The parameter $\delta \in (0,1)$ is the tolerant probability of a wrong prediction. Unless otherwise stated, we always assume $\mu_1 \geq \mu_2 \geq \cdots \geq \mu_K$, but the order remains unknown to the agent.

**Dynamics.** The agent's pulling strategy $(\pi, \tau, \hat{a})$ is parametrized by a sampling rule $\pi = \{\pi_t\}_{t=1}^{\infty}$, a stopping time $\tau$ and a recommendation rule $\hat{a} \in [K] \cup \{\mathsf{None}\}$. When a pulling strategy is applied on a 1-identification problem instance $\nu$, the strategy generates a history $A_1, X_1, \cdots, A_\tau, X_\tau$. Action $A_t \in [K]$ is chosen contingent upon the history $H(t-1) = \{(A_s, X_s)\}_{s=1}^{t-1}$, via the function $\pi_t(H(t-1))$. In addition, we have $X_t \sim \nu_{A_t}$. The agent stops pulling at the end of time step $\tau$, where $\tau$ is a stopping time[1] with respect to the filtration $\{\sigma(H(t))\}_{t=1}^{\infty}$. Upon stopping, the agent outputs $\hat{a} \in [K] \cup \{\mathsf{None}\}$ to be the answer, using the information $H(\tau)$. Outputting $\hat{a} \in [K]$ means that the agent concludes with arm $\hat{a}$ satisfying $\mu_{\hat{a}} \geq \mu_0$, while outputting $\hat{a} = \mathsf{None}$ means that the agent concludes with no arm having mean reward $\geq \mu_0$. We allow the possibility of non-termination $\tau = \infty$, in which case there is no recommendation $\hat{a}$.

To facilitate our discussions, we introduce the definitions of positive and negative instances.

**Definition 2.1** (Positive and Negative Instances). Denote $\nu$ as a distribution vector equipped with a mean reward vector $\{\mu_a\}_{a=1}^{K}$. We call $\nu$ a positive instance, if $\mu_1 > \mu_0$. And we call $\nu$ a negative instance, if $\mu_1 < \mu_0$.

Correspondingly, we denote $\mathcal{S}^{\mathrm{pos}} = \{\nu : \mu_1 > \mu_0\}$ and $\mathcal{S}^{\mathrm{neg}} = \{\nu : \mu_1 < \mu_0\}$. For $\Delta > 0$, we further define $\mathcal{S}_{\Delta}^{\mathrm{pos}} = \{\nu : \mu_1 - \mu_0 \geq \Delta\}$ and $\mathcal{S}_{\Delta}^{\mathrm{neg}} = \{\nu : \mu_0 - \mu_1 \geq \Delta\}$. It is clear that $\mathcal{S}^{\mathrm{pos}} = \cup_{\Delta > 0} \mathcal{S}_{\Delta}^{\mathrm{pos}}$ and $\mathcal{S}^{\mathrm{neg}} = \cup_{\Delta > 0} \mathcal{S}_{\Delta}^{\mathrm{neg}}$. In this paper, we assume that the underlying instance belongs to $\mathcal{S}^{\mathrm{pos}} \cup \mathcal{S}^{\mathrm{neg}}$. For an instance $\nu \in \mathcal{S}^{\mathrm{pos}} \cup \mathcal{S}^{\mathrm{neg}}$, we define $i^*(\nu)$ as the set of correct answers. For $\nu \in \mathcal{S}^{\mathrm{pos}}$, we have $i^*(\nu) = \{a : \mu_a \geq \mu_0\}$, while for instance $\nu \in \mathcal{S}^{\mathrm{neg}}$, we have $i^*(\nu) = \{\mathsf{None}\}$. We mainly focus on analyzing PAC algorithms, with the following definitions.

**Definition 2.2.** A pulling strategy is $\delta$-PAC, if for any $\delta \in (0,1), \nu \in \mathcal{S}^{\mathrm{pos}} \cup \mathcal{S}^{\mathrm{neg}}$, it satisfies $\mathrm{Pr}_\nu(\tau < +\infty, \hat{a} \in i^*(\nu)) > 1 - \delta$.

**Definition 2.3.** A pulling strategy is $(\Delta, \delta)$-PAC, if it is $\delta$-PAC, and for any $\Delta, \delta > 0$, we have $\sup_{\nu \in \mathcal{S}_{\Delta}^{\mathrm{pos}} \cup \mathcal{S}_{\Delta}^{\mathrm{neg}}} \mathbb{E}_\nu \tau < +\infty$.

Clearly, if a 1-identification pulling strategy is $(\Delta, \delta)$-PAC,

---

[1] For any $t$, the event $\{\tau = t\}$ is $\sigma(H(t))$-measurable

it is also $\delta$-PAC.

**Objective.** The agent aims to design a $(\Delta, \delta)$-PAC pulling strategy $(\pi, \tau, \hat{a})$ that minimizes the *sampling complexity* $\mathbb{E}[\tau]$.

## 3. Literature Review

We review existing research works on the 1-identification problem. To aid our discussions, we define the following notations for describing bounds on sampling complexity. We define $\Delta_{i,j} = |\mu_i - \mu_j|$ for all $i, j \in [K] \cup \{0\}$ and

$$
\begin{aligned}
H_1^{\mathrm{neg}} &= \sum_{a=1}^{K} \frac{2}{\Delta_{0,a}^2}, \quad H_1^{\mathrm{low}} = \sum_{a: \mu_a < \mu_0} \frac{2}{\Delta_{1,a}^2}, \\
H_1^{\mathrm{pos}} &= \sum_{a=1}^{K} \frac{2}{\max\{\Delta_{0,a}^2, \Delta_{1,a}^2\}}, \quad H = \frac{2}{\Delta_{0,1}^2} \\
H_1 &= \sum_{a=2}^{K} \frac{2}{\Delta_{1,a}^2}, \quad H_0 = \sum_{a: \mu_a \geq \mu_0} \frac{2}{\Delta_{0,a}^2}, \\
H_1^{\mathrm{BAI}} &= \frac{2}{\Delta_{0,1}^2} + \sum_{a=2}^{K} \frac{2}{\Delta_{1,a}^2}.
\end{aligned}
\tag{1}
$$

Since $\Delta_{1,1} = 0$, we equivalently have $H_1^{\mathrm{pos}} = \frac{2}{\Delta_{0,1}^2} + \sum_{a=2}^{K} \frac{2}{\max\{\Delta_{0,a}^2, \Delta_{1,a}^2\}}$. Table 1 summarizes the existing algorithms and their performance guarantees with notations in (1). Details are as follows.

The 1-identification problem is a pure exploration problem with possibly multiple answers, which has been studied in (Degenne & Koolen, 2019). They consider the case when $\nu_a$ belongs to the one-parameter one-dimensional exponential family for each $a \in [K]$, and propose the Sticky-Track-and-Stop (S-TaS) algorithm. S-TaS satisfies $\limsup_{\delta \to 0} \frac{\mathbb{E}[\tau]}{\log(1/\delta)} = T^*(\mu)$, which depends not only on $\{\mu_a\}_{\{0\} \cup [K]}$ but also $\{\nu_a\}_{a \in [K]}$. With their lower bound $\liminf_{\delta \to 0} \frac{\mathbb{E}[\tau]}{\log(1/\delta)} \geq T^*(\mu)$ on any $\delta$-PAC algorithm, they conclude S-TaS achieves asymptotic optimality. We display the bound $T^*(\mu)$ in the special case when $\nu_a = N(\mu_a, 1)$ for each $a \in \mathcal{A}$, where $T^*(\mu)$ specializes to the bounds in the first row in Table 1. Nevertheless, the non-asymptotic sample complexity of the 1-identification problem remains a mystery in (Degenne & Koolen, 2019), who comment that "Both lower bounds and upper bounds in this paper are asymptotic... **A finite time analysis** with reasonably small $o(\log \frac{1}{\delta})$ terms for an optimal algorithm **is desirable**."

(Kano et al., 2017) study the Good Arm Identification problem, where the agent aims to output all arms whose mean reward is greater than $\mu_0$. This objective is different from our objective, which requires returning only a qualified arm if exists, but the expected stopping time of their first output

*Table 1.* Comparison of bounds. "pos" and "neg" in the second column correspond to $\mu_1 > \mu_0$ and $\mu_1 < \mu_0$ respectively. The definitions of different $H$'s are at the equation (1). We determine whether a sampling complexity upper bound is nearly optimal by comparing with the lower bound. An upper bound is nearly optimal iff the gap is up to a polynomial logarithm factor.

| Algorithm | Bound | | Nearly Opt in Positive | Nearly Opt in Negative | Comment |
|---|---|---|---|---|---|
| S-TaS | $\lim\limits_{\delta \to 0} \frac{\mathbb{E}\tau}{\log \frac{1}{\delta}} = \begin{cases} H & \text{pos} \\ H_1^{\text{neg}} & \text{neg} \end{cases}$ | | $\checkmark$ | $\checkmark$ | Asymptotic |
| HDoC | $\mathbb{E}\tau \leq \begin{cases} O(H \log \frac{K}{\delta} + H_1 \log\log \frac{1}{\delta} + \frac{K}{\epsilon^2}) & \text{pos} \\ O(H_1^{\text{neg}} \log \frac{K}{\delta} + \frac{K}{\epsilon^2}) & \text{neg} \end{cases}$ | | $\times$ | $\times$ | $\epsilon = \min\left\{ \min\limits_{a \in [K]} \Delta_{0,a}, \min\limits_{a \in [K-1]} \frac{\Delta_{a,a+1}}{2} \right\}$ |
| APGAI | $\mathbb{E}\tau \leq \begin{cases} O(H_0 (\log \frac{K}{\delta})) & \text{pos} \\ O(H_1^{\text{neg}} (\log \frac{K}{\delta})) & \text{neg} \end{cases}$ | | $\times$ | $\checkmark$ | Asymptotic Optimal in the case neg |
| SEE (This work) | $\mathbb{E}\tau \leq \begin{cases} O(H \log \frac{1}{\delta}) + O(H_1^{\text{pos}} \log \frac{K}{\Delta_{0,1}}) & \text{pos} \\ O(H_1^{\text{neg}} (\log \frac{1}{\delta} + \log H_1^{\text{neg}})) & \text{neg} \end{cases}$ | | $\checkmark$ | $\checkmark$ | Even the $O(\log \frac{1}{\Delta_{0,1}})$ matches lower bound in some cases |
| Lower Bound (This work) | $\mathbb{E}\tau \geq \begin{cases} \Omega(H \log \frac{1}{\delta} + \frac{1}{m} H_1^{\text{low}} - \frac{1}{\Delta_{1,m+1}^2}) & \text{pos} \\ \Omega(H_1^{\text{neg}} \log \frac{1}{\delta}) & \text{neg} \end{cases}$ | | NA | NA | In pos, $m \in [K]$ satisfies $\mu_m \geq \mu_0 > \mu_{m+1}$ |

is exactly the $\mathbb{E}\tau$ in our formulation. They propose algorithm HDoC, whose sub-optimality of their sampling complexity upper bound comes from two perspectives. Firstly, the term $\frac{1}{\epsilon^2}$ implies that the upper bound is vacuous (equal to infinity) when there exists $a \in [K]$ such that $\mu_a = \mu_0$, even in the case of $\mu_1 > \mu_0$. Secondly, when $\mu_1 > \mu_0$, (Kano et al., 2017)'s bound includes $H_1 \log\log(1/\delta)$, and $H_1$ grow linearly with $K$. (Kano et al., 2017) comment that "Therefore, in the case that ... $K = \Omega(\frac{\log \frac{1}{\delta}}{\log\log \frac{1}{\delta}})$... **there still exists a gap** between the lower bound in ... and the upper bound in ...". (Tsai et al., 2024) incorporate the LIL concentration event (Lemma 3 in (Jamieson et al., 2014)) into the algorithm HDoC with an additional warm-up stage. But the proposed algorithm lilHDoC still suffers from the same sub-optimality as HDoC.

(Jourdan & Réda, 2023) mainly focus on designing anytime algorithms on Good Arm Identification. They propose pulling rule APGAI with the stopping rule GLR. Their algorithm achieves nearly optimal asymptotic sample complexity bounds on negative instances, but is still sub-optimal on positive instances. The stopping rule GLR can be combined with other pulling rules to guarantee the $\delta$-PAC property, but in general there is no non-trivial upper bound on $\mathbb{E}\tau$. Finally, we remark that the Good Arm Identification formulation assumes $\mu_a \neq \mu_0$ for all $a \in [K]$, and their upper bounds are vacuous if there exists an arm $a$ whose mean reward is exactly $\mu_0$.

Table 1 summarizes the comparison, with imprecision on HDoC and APGAI. Details are in appendix A.2. Among them, S-TaS is $(\Delta, \delta)$-PAC, but the non-asymptotic pulling

complexity remains unclear. Appendix A.3 proves algorithm HDoC, lilHDoC, APGAI are not $(\Delta, \delta)$-PAC, and Appendix E.3 provides additional discussion on APGAI.

Besides the above work, there are other research works related to 1-identification, despite not directly solving the 1-identification. (Kaufmann et al., 2018) provide asymptotic sample complexity bounds on classifying positive and negative instances, but their algorithm does not output a qualified arm $\hat{a} \in [K]$ on a positive instance. (Katz-Samuels & Jamieson, 2020) propose the idea of "Bracketing" to solve the 1-identification problem. Their algorithm only applies for positive instances but not the negative instances, since the algorithm is not required certifying if there is no arm with mean reward at least $\mu_0$. More discussion on (Katz-Samuels & Jamieson, 2020) is provided in Appendix A.1.

The 1-identification problem can be solved by considering a Best Arm Identification(BAI) problem, which has been well studied. We can take arm 0 as a virtual arm which always returns a deterministic reward with value $\mu_0$. Applying an algorithm for the fixed confidence setting (Even-Dar et al., 2002; Gabillon et al., 2012; Jamieson et al., 2013; Kalyanakrishnan et al., 2012; Karnin et al., 2013; Jamieson & Nowak, 2014), the agent achieves a non-asymptotic high probability upper bound. For example, (Jamieson & Nowak, 2014) guarantee $\Pr_\nu(\tau < O(H_1^{\text{BAI}}(\log \frac{1}{\delta} + \log H_1^{\text{BAI}}))) > 1 - \delta$ for $\nu \in \mathcal{S}^{\text{pos}}$, $\Pr_\nu(\tau < O(H_1^{\text{neg}}(\log \frac{1}{\delta} + \log H_1^{\text{neg}}))) > 1 - \delta$ for $\nu \in \mathcal{S}^{\text{neg}}$. And the logarithmic term $H_1^{\text{BAI}}, H_1^{\text{neg}}$ can be improved. Nevertheless, the high probability upper bound cannot imply $\mathbb{E}\tau < +\infty$. To address this issue, (Chen & Li, 2015; Chen et al., 2017) develop the Parallel

Simulation Algorithm, which takes a BAI algorithm as an Oracle and converts this algorithm into a new algorithm. The new algorithm is still $\delta$-PAC, and guarantees not only the above high probability upper bound, but also finite upper bound on $\mathbb{E}\tau$. (Garivier & Kaufmann, 2016; Kaufmann et al., 2016; Garivier et al., 2019) provide asymptotic upper bounds relating to $\mathbb{E}\tau$, complemented with matching lower bounds. Though there are many existing works on the BAI problem, the comparison of asymptotic results in these two formulations (Garivier & Kaufmann, 2016; Degenne & Koolen, 2019) suggests that it is inefficient to solve the 1-identification by the corresponding BAI problem.

Our work is related to other topics, such as BAI in the fixed budget setting and the $\epsilon$-Good Arm Identification problem. We provide additional discussions in Appendix A.1.

## 4. Algorithm

### 4.1. An Informal Algorithm

---
**Algorithm 1** SEE(Informal)
---
1: **Input:** Action set $[K]$, threshold $\mu_0$, tolerance level $\delta$, $C > 1$.
2: **Tune** $\{\delta_k, T_k^{\mathrm{et}}, T_k^{\mathrm{ee}}\}_{k=1}^{+\infty}$.
3: **for** Phase $k = 1, 2, \cdots$ **do**
4:    (Exploration) Run algorithm LUCB_G with tolerance level $\delta_k$ and previous exploration history.
     Stops until one of the two conditions holds.
5:    • Total pulling times in all exploration phases is greater than $T_k^{\mathrm{ee}}$. Take $\hat{a}_k = $ Not Complete
6:    • LUCB_G stops and output $\hat{a}_k \in [K] \cup \{$None$\}$.
7:    **if** $\hat{a}_k \in [K]$ **then**
8:      (Exploitation) Keep pulling arm $\hat{a}_k$ with samples independent of exploration.
       Stops until one of the two conditions holds.
9:      • Pulling times of $\hat{a}_k$ is greater than $T_k^{\mathrm{et}}$.
10:     • LCB defined by $\delta$ is above $\mu_0$, output $\hat{a}_k$ as a qualified arm
11:    **else if** $\hat{a}_k = $ None and $\delta_k < \frac{\delta}{3}$ **then**
12:      Output the instance is negative
13:    **end if**
14: **end for**
---

Before rigorously describing the algorithm, we provide an overview with a simple algorithm sketch. This sketch presents the overall framework of SEE, but some required adaptations are explained in the full in the subsequent parts of this section.

Algorithm 1 introduces the key idea of SEE. Algorithm LUCB_G in line 4 is defined in (Kano et al., 2017), which can be considered an adapted BAI algorithm UCB in (Jamieson & Nowak, 2014). In each round, LUCB_G pulls the arm with the highest upper confidence bound (UCB),

and stops if there is an arm whose lower confidence bound (LCB) is $> \mu_0$. LUCB_G guarantees the high probability bound $\mathrm{Pr}_\nu(\mu_{\hat{a}} > \mu_0, \tau < O(H_1^{\mathrm{pos}} \log(H_1^{\mathrm{pos}}/\delta'))) > 1 - \delta'$ for $\nu \in \mathcal{S}^{\mathrm{pos}}$, and $\mathrm{Pr}_\nu(\hat{a} = $ None$, \tau < O(H_1^{\mathrm{neg}} \log(H_1^{\mathrm{neg}}/\delta'))) > 1 - \delta'$ for $\nu \in \mathcal{S}^{\mathrm{neg}}$, where $\delta' \in (0, 1)$ is the input tolerance level.

Algorithm 1 takes LUCB_G as an oracle by sequentially calling it with tolerance level $\delta_k$ at phase $k$. The sequence $\{\delta_k\}_{k \in \mathbb{N}}$ is non-increasing with $k$, and in a phase $k$ $\delta_k$ should be larger than the required tolerance level $\delta$. In line 12 of Algorithm 1, we trust the negative prediction of LUCB_G only when the $\delta_k$ is smaller than the required tolerance level $\delta$. The intuition is consistent with the conclusion in (Degenne & Koolen, 2019), which concludes lower bound $\Omega(H_1^{\mathrm{neg}} \log \frac{1}{\delta})$ is required for a negative instance. Then, it is natural to accept the negative output "None" when $\delta_k < O(\delta)$.

While meeting a positive instance, we wish LUCB_G can identify a qualified arm with pulling times $O(H_1^{\mathrm{pos}} \log \frac{H_1^{\mathrm{pos}}}{\delta_k})$, which is line 6. Then we turn to keep pulling $\hat{a}_k$ with confidence bound defined by true tolerance level $\delta$, corresponding to the line 8. Line 5 is to avoid LUCB_G from getting stuck into a non-stopping loop, which is possible when the "good event" doesn't hold.

Based on the above idea, Algorithm 1 seems to be able to solve the problem. However, there are still two main concerns in Algorithm 1, stopping us from adopting it directly. The first one is LUCB_G cannot guarantee the lower bound of $\mu_{\hat{a}_k} - \mu_0$, which makes it hard to set up maximum pulling times $T_k^{\mathrm{et}}$ in the exploitation phase. To address this issue, we introduce a tunable parameter $C > 1$ and adopt a larger radius when calculating the LCB in Algorithm 3.

The second concern is LUCB_G requires **at the start of an exploration phase** $k$, $\mathrm{LCB}_a(\delta_k) < \mu_0$ **holds for all** $a \in [K]$. But in Algorithm 1, it is possible that at the end of phase $k - 1$, the last collected sample of arm $\hat{a}_{k-1}$ is so large, such that LCB of $\hat{a}_{k-1}$ is still above $\mu_0$ at the start of phase $k$. To address this issue, we define a temporary container $Q$. When the LUCB_G is going to output an arm $\hat{a}_{k-1}$, we transfer the latest collected sample of $\hat{a}_{k-1}$ into $Q$. If we pull arm $\hat{a}_{k-1} \in [K]$ in the future exploration period, we transfer the sample back from $Q$ to history. The reason of transferring back from $Q$ is concentration inequality like (3) requires consecutive integer index summation, and we cannot skip or abandon any samples.

### 4.2. Sequential Exploration Exploitation

For simplicity, define $\lceil x \rceil^+ = \max\{\lceil x \rceil, 1\}$. We define the confidence radius

$$U(t, \delta) = \frac{\sqrt{2 \cdot 2^{\lceil \log_2 t \rceil^+} \log \frac{2(\lceil \log_2 t \rceil^+)^2}{\delta}}}{t}. \qquad (2)$$

To rigorously present the algorithm design, we split Algorithm 1 into three parts with more adaptation. Algorithm 2 is the main body, calling Algorithms 3 and 4 to conduct the pulling procedure. We elaborate on them one by one.

---

**Algorithm 2** Sequential-Explore-Exploit(SEE)

1: **Input:** Action set $[K]$, threshold $\mu_0$, confidence level $\delta$, $C > 1$, scheduling parameter $\{\delta_k, \alpha_k, \beta_k\}_{k=1}^{+\infty}$. {Default: $C = 1.01, \delta_k = 1/3^k, \alpha_k = 5^k, \beta_k = 2^k$).}
2: Calculate $T_k^{\text{ee}} = 1000(C+1)^2 K \beta_k \log(4K/\delta_k), T_k^{\text{et}} = 1000\beta_k \log(4\alpha_k K/\delta)$.
3: **Initialize:** $\mathcal{H}^{\text{ee}} = \mathcal{H}^{\text{et}} = Q = \emptyset, N_a^{\text{ee}} = N_a^{\text{et}} = 0, \forall a \in [K]$.
4: **for** Phase $k = 1, 2, \cdots$ **do**
5:     (Exploration) Call Algorithm 3 with
        $K \leftarrow K, \mu_0 \leftarrow \mu_0, \delta \leftarrow \delta_k, C \leftarrow C, \mathcal{H}^{\text{ee}} \leftarrow \mathcal{H}^{\text{ee}}$,
        $Q \leftarrow Q, T \leftarrow T_k^{\text{ee}}$,
        denote $(\hat{a}_k, \mathcal{H}^{\text{ee}}, Q, \tau_k^{\text{ee}})$ as the output.
6:     **if** $\hat{a}_k \in [K]$ **then**
7:         (Exploitation) Call Algorithm 4 with
            $K \leftarrow K, \mu_0 \leftarrow \mu_0, \delta \leftarrow \delta, \mathcal{H}^{\text{et}} \leftarrow \mathcal{H}^{\text{et}}$,
            $T \leftarrow T_k^{\text{et}}, \hat{a} \leftarrow \hat{a}_k, \alpha \leftarrow \alpha_k$,
            denote $(\text{ans}, \mathcal{H}^{\text{et}}, \tau_k^{\text{et}})$ as the output.
8:         **if** ans = Qualified **then**
9:             Return $\hat{a} = \hat{a}_k$.
10:        **end if**
11:    **else if** $\hat{a}_k =$ None and $\delta_k < \delta/3$ **then**
12:        Return $\hat{a} =$ None.
13:    **end if**
14: **end for**

---

SEE, displayed in Algorithm 2, takes $\mu_0, K, \delta, C, \{\delta_k, \alpha_k, \beta_k\}_{k=1}^{+\infty}$ as input, where $\mu_0, K, \delta$ are model parameters and $C, \{\delta_k, \alpha_k, \beta_k\}_{k=1}^{+\infty}$ are tunable parameters. Besides $C = 1.01, \delta_k = 1/3^k, \alpha_k = 5^k, \beta_k = 2^k$, other choices are also available, see Appendix B.1.

Following Algorithm 1, Algorithm 2 sequentially calls exploration and exploitation oracles in each phase, while maintaining three sample sets $\mathcal{H}^{\text{ee}}, \mathcal{H}^{\text{et}}, Q$. When calling the Exploration Algorithm 3, $\mathcal{H}^{\text{ee}}, Q$ are updated. When calling Exploitation Algorithm 4, $\mathcal{H}^{\text{et}}$ is updated.

Besides $\mathcal{H}^{\text{ee}}, \mathcal{H}^{\text{et}}, Q$, Algorithm 2 further specifies $T_k^{\text{ee}} = 1000(C + 1)^2 K \beta_k \log(4K/\delta_k)$ and $T_k^{\text{et}} = 1000\beta_k \log(4\alpha_k K/\delta)$. The coefficient 1000 is to simplify the calculation in the proof of Lemma B.1. If we replace 1000 with another fixed positive constant, the main conclusion still holds.

Algorithm 3 follows LUCB_G in (Kano et al., 2017), with three major modifications. The first is adopting (2) as the radius of the confidence interval, which is smaller than the original LUCB_G. The second is adopting $C \cdot U(\cdot, \cdot)$ for the LCB in line 2, where $C > 1$. In the case of a positive

---

**Algorithm 3** SEE-Exploration

1: **Input:** Action set $[K]$, threshold $\mu_0$, confidence level $\delta$, tunable parameter $C > 1$, History $\mathcal{H}^{\text{ee}} = \cup_{a=1}^{K} \{(a, X_{a,s}^{\text{ee}})\}_{s=1}^{N_a^{\text{ee}}}$, Temporary Container $Q$, Termination Round $T$.
2: Define $\hat{\mu}_a(\mathcal{H}^{\text{ee}}) = \hat{\mu}_{a,N_a^{\text{ee}}} = \frac{\sum_{s=1}^{N_a^{\text{ee}}} X_{a,s}^{\text{ee}}}{N_a^{\text{ee}}}, t = |\mathcal{H}^{\text{ee}} \cup Q|$,
    $\text{UCB}_a(\mathcal{H}^{\text{ee}}, \delta) = \hat{\mu}_a(\mathcal{H}^{\text{ee}}) + U(N_a^{\text{ee}}, \delta/K)$,
    $\text{LCB}_a(\mathcal{H}^{\text{ee}}, \delta) = \hat{\mu}_a(\mathcal{H}^{\text{ee}}) - C \cdot U(N_a^{\text{ee}}, \delta/K)$.
3: **while** True **do**
4:     Determine $A_t^{\text{ee}} = \arg\max_{a \in [K]} \text{UCB}_a(\mathcal{H}^{\text{ee}}, \delta)$.
5:     Get $(X, Q, \Delta t)$ by calling Sampling Rule(Alg 5), $A \leftarrow A_t^{\text{ee}}, Q \leftarrow Q$
6:     $N_{A_t^{\text{ee}}} = N_{A_t^{\text{ee}}} + 1, t = t + \Delta t, \mathcal{H}^{\text{ee}} = \mathcal{H}^{\text{ee}} \cup \{(A_t^{\text{ee}}, X)\}$
7:     **if** $t \geq T$ **then**
8:         Break and take $\hat{a} =$ Not Complete
9:     **else if** $\forall a \in [K], \text{UCB}_a(\mathcal{H}^{\text{ee}}, \delta) \leq \mu_0$ **then**
10:        Break and take $\hat{a} =$ None
11:    **else if** $\text{LCB}_{A_t^{\text{ee}}}(\mathcal{H}^{\text{ee}}, \delta) > \mu_0$ **then**
12:        $\mathcal{H}^{\text{ee}} = \mathcal{H}^{\text{ee}} \setminus \{(A_t^{\text{ee}}, X)\}$
13:        $N_{A_t^{\text{ee}}} = N_{A_t^{\text{ee}}} - 1, Q = Q \cup \{(A_t^{\text{ee}}, X)\}$
14:        Break and take $\hat{a} = a$
15:    **end if**
16: **end while**
17: Return $(\hat{a}, \mathcal{H}^{\text{ee}}, Q, t)$

---

**Algorithm 4** SEE-Exploitation

1: **Input:** Action set $[K]$, threshold $\mu_0$, confidence level $\delta$, Termination Round $T$, Predicted arm $\hat{a} \in [K]$, Tolerance Controller $\alpha$, History $\mathcal{H}^{\text{et}} = \cup_{a=1}^{K} \{(a, X_{a,s}^{\text{et}})\}_{s=1}^{N_a^{\text{et}}}$, $t = |\mathcal{H}^{\text{et}}|$
2: **while** $N_{\hat{a}}^{\text{et}} \leq T$ **do**
3:     Sample $X \sim \nu_{\hat{a}}$,
4:     $\mathcal{H}^{\text{et}} = \mathcal{H}^{\text{et}} \cup \{(\hat{a}, X)\}, N_{\hat{a}}^{\text{et}} = N_{\hat{a}}^{\text{et}} + 1, t = t + 1$
5:     **if** $\frac{\sum_{s=1}^{N_{\hat{a}}^{\text{et}}} X_{\hat{a},s}^{\text{et}}}{N_{\hat{a}}^{\text{et}}} - U(N_{\hat{a}}^{\text{et}}, \frac{\delta}{K\alpha}) > \mu_0$ **then**
6:         return (Qualified, $\mathcal{H}^{\text{et}}, t$)
7:     **end if**
8: **end while**
9: return (Unqualified, $\mathcal{H}^{\text{et}}, t$)

---

**Algorithm 5** Sampling Rule

1: **Input:** Arm $a \in [K]$, Temporary Container $Q$
2: **if** $\exists (a, X) \in Q$ such that $a = A$ **then**
3:     $Q = Q \setminus \{(a, X)\}, \Delta t = 0$, return $X, Q, \Delta t$
4: **else**
5:     Sample $X \sim \nu_A, \Delta t = 1$, return $X, Q, \Delta t$
6: **end if**

instance, factor $C > 1$ can guarantee $\mu_{\hat{a}_k} > \omega \mu_1 + (1 - \omega)\mu_0$ where $\omega = \frac{C-1}{C+3}$, conditioned on an event that holds with probability $\geq 1 - \frac{\pi^2 \delta_k}{6}$. Details are in Lemma B.4. The third is using the temporary container $Q$ to conduct the modified sampling of an arm $X$ in line 5.

Algorithm 5 conducts the modified sampling for Algorithm 3. It only collects a new sample for arm $A \in [K]$ when $Q$ does not contain a sample on $A$. We only increase the total pulling times $t$ when we collect a new sample from $\nu_A$. Algorithm 5 returns the updated $Q$ together with the sample $X$. The necessity of using $Q$ to temporarily store the latest collected sample $X$ in line 13 in Algorithm 3 is discussed in section 4.1 and Lemma B.6. From the above discussion, we know for each $a \in [K]$, $Q$ contains at most one tuple whose first element is $a$:

**Lemma 4.1.** *Throughout the pulling process, $|Q| \leq K$ holds with certainty.*

Algorithm 4 takes $(K, \mu_0, \delta, T, \hat{a})$ as the input. The parameters $K, \mu_0, \delta$ are model parameters, which remain the same in all phases. By contrast, the parameters $T$ and $\hat{a}$ generally change across different phases. During phase $k$, Algorithm 4 keeps pulling arm $\hat{a}_k$ until one of the two following conditions is met. The first is when the total pulling times of arm $\hat{a}_k$ in all exploitation periods is $T$, which should be $T_k^{\text{et}}$ at the phase $k$. In this case, the exploitation period cannot guarantee that $\mu_{\hat{a}_k} > \mu_0$ holds with probability $\geq 1 - \delta$, and requires Algorithm 2 to run the next exploration period with a smaller tolerance level $\delta_{k+1}$. The second condition is when the LCB, defined with the input tolerance level $\delta$, of $\hat{a}_k$ is above $\mu_0$. In this case, Algorithm 2 adopts this result and outputs $\hat{a}_k$ as $\hat{a}$, asserting that the instance is positive.

Throughout the pulling process, the samples collected among all the exploration periods are shared, as are the exploitation periods. By contrast, the exploration periods never share samples with the exploitation period. To distinguish between the two sets of periods, we denote $\tau_k^{\text{ee}}$ and $\tau_k^{\text{et}}$ as the total pulling times of Algorithms 3 and 4, from the start of phase 1 to the end of phase $k$. We also denote $N_a^{\text{ee}}(\tau_k^{\text{ee}}), N_a^{\text{et}}(\tau_k^{\text{et}})$ as the total pulling times of arm $a \in [K]$ stored in $\mathcal{H}^{\text{ee}}$ and $\mathcal{H}^{\text{et}}$, at the end of phase $k$. From the algorithm design, it is clear that the following Lemma holds.

**Lemma 4.2.** *At the end of phase $k$, 1.* $\tau_k^{\text{ee}} \leq T_k^{\text{ee}}$, $\tau_k^{\text{et}} \leq \sum_{p=1}^{k} T_p^{\text{et}}$. *2.* $|Q| + |\mathcal{H}^{\text{ee}}| = \tau_k^{\text{ee}}$. *3. Since* $|\mathcal{H}^{\text{ee}}| = \sum_{a=1}^{K} N_a^{\text{ee}}(\tau_k^{\text{ee}})$, *we can further conclude* $\sum_{a=1}^{K} N_a^{\text{ee}}(\tau_k^{\text{ee}}) \leq \tau_k^{\text{ee}} \leq K + \sum_{a=1}^{K} N_a^{\text{ee}}(\tau_k^{\text{ee}})$.

## 5. Main Results

In this section, we demonstrate upper bounds of $\mathbb{E}\tau$ by applying SEE, and provide lower bounds on $\mathbb{E}\tau$ for any

$\delta$-PAC algorithm. The lower bound mainly comes from the existing results in the literature.

### 5.1. Performance Guarantee of Algorithm 2

Before stating the main theorems of Algorithm 2, we introduce some notation. Define $\hat{\mu}_{a,t}^{\text{ee}} = (1/t) \sum_{s=1}^{t} X_{a,s}^{\text{ee}}$, $\hat{\mu}_{a,t}^{\text{et}} = (1/t) \sum_{s=1}^{t} X_{a,s}^{\text{et}}$, and

$$
\kappa^{\text{ee}} = \min \left\{ k \in \mathbb{N} : \forall a \in [K], \forall t \in \mathbb{N}, \right.
$$
$$
\left. \left| \hat{\mu}_{a,t}^{\text{ee}} - \mu_a \right| \leq U(t, \frac{\delta_k}{K}) \right\},
$$
$$
\kappa^{\text{et}} = \min \left\{ k \in \mathbb{N} : \forall a \in [K], \forall t \in \mathbb{N}, \right.
$$
$$
\left. \left| \hat{\mu}_{a,t}^{\text{et}} - \mu_a \right| \leq U(t, \frac{\delta}{K\alpha_k}) \right\}.
$$

(3)

Here $\kappa^{\text{ee}}, \kappa^{\text{et}}$ are the minimum phase indices such that the respective concentration inequality hold, as in phase $k$, we use $U(t, \frac{\delta_k}{K})$ and $U(t, \frac{\delta}{K\alpha_k})$ to define the UCBs and the LCBs during the exploration and exploitation periods. Not hard to see $\kappa^{\text{ee}}, \kappa^{\text{et}}$ are random variables. By Lemma D.3 in the Appendix, we have the following.

**Lemma 5.1.** *For all $k \in \mathbb{N}$, it holds that*

$$
\Pr(\kappa^{ee} \geq k) \leq \frac{\pi^2 \delta_{k-1}}{6}, \quad \Pr(\kappa^{et} \geq k) \leq \frac{\pi^2}{6} \cdot \frac{\delta}{\alpha_{k-1}}.
$$

Since $\lim_{k \to \infty} \delta_k = 0, \lim_{k \to \infty} \alpha_k = +\infty$, we observe that $\Pr(\kappa^{\text{ee}} < +\infty) = 1, \Pr(\kappa^{\text{et}} < +\infty) = 1$. We first show that Algorithm 2 is $\delta$-PAC, which is mainly due to the design of our stopping rule.

**Theorem 5.2.** *Algorithm 2 is $\delta$-PAC.*

*Sketch Proof of Theorem 5.2.* The first step is to show $\Pr(\tau = +\infty) \leq \Pr(\kappa^{\text{ee}} = +\infty) + \Pr(\kappa^{\text{et}} = +\infty) = 0$, indicating that $\tau < +\infty$ with certainty. Then, we know $\hat{a} \in [K] \cup \{\text{None}\}$ in Algorithm 2 is well defined with certainty.

If $\nu$ is positive, the event $\hat{a} \notin i^*(\nu)$ occurs only in the following two scenarios. The first scenario is when an exploration period outputs None in phase $k$ such that $\delta_k < \delta/3$. The second scenario is when an exploitation period outputs an arm with mean reward $< \mu_0$. The probabilities of both events are at most $\delta$ multiplied by an absolute constant, and we can conclude $\Pr_\nu(\hat{a} \notin i^*(\nu)) < \delta$.

If $\nu$ is negative, the event $\hat{a} \notin i^*(\nu)$ occurs only when an exploitation period outputs an arm, which is the same as the second case in the discussion of positive $\nu$. We then have $\Pr_\nu(\hat{a} \notin i^*(\nu)) < \delta$ again, hence the Lemma is proved. $\square$

The full proof is in Appendix B.2. The following theorem shows that Algorithm 2 is nearly optimal in minimizing $\mathbb{E}\tau$.

**Theorem 5.3.** *Apply Algorithm 2 to an instance $\nu$, we have*

$$\mathbb{E}\tau \leq \begin{cases} \gamma \cdot \left( \frac{\log \frac{1}{\delta}}{\Delta_{0,1}^2} + (\log \frac{K}{\Delta_{0,1}^2}) \cdot H_1^{\text{pos}} \right) & \nu \in \mathcal{S}^{\text{pos}} \\ \gamma \cdot (H_1^{\text{neg}}(\log \frac{1}{\delta} + \log H_1^{\text{neg}})) & \nu \in \mathcal{S}^{\text{neg}} \end{cases}, \text{ where}$$

*$\gamma$ only depends the constant $C$ but independent of model parameters $K$, $\delta$ and $\{\mu_a\}_{a=1}^K$, In particular, when we set $C$ to be an absolute constant such as $C = 1.01$, $\gamma$ is also an absolute constant.*

From the definition of $H_1^{\text{pos}}$, we know $K \leq H_1^{\text{pos}}$, and $1/\Delta_{0,1}^2 \leq H_1^{\text{pos}}$. Thus, we have $\log(K/\Delta_{0,1}^2) \leq 2\log H_1^{\text{pos}}$. Meanwhile, since $\max\{\Delta_{0,a}^2, \Delta_{1,a}^2\} \geq \Delta_{0,1}^2/4$, we have $H_1^{\text{pos}} \leq 8K/\Delta_{1,0}^2$, leading to $\log H_1^{\text{pos}} \leq \log 8 + \log(K/\Delta_{1,0}^2)$. Thus, it is equivalent to state the upper bound as $\mathbb{E}\tau \leq \begin{cases} \gamma \cdot \left( \frac{\log \frac{1}{\delta}}{\Delta_{0,1}^2} + (\log H_1^{\text{pos}}) \cdot H_1^{\text{pos}} \right) & \nu \in \mathcal{S}^{\text{pos}} \\ \gamma \cdot (H_1^{\text{neg}}(\log \frac{1}{\delta} + \log H_1^{\text{neg}})) & \nu \in \mathcal{S}^{\text{neg}} \end{cases}$. As Theorem 5.7 discusses the existence of $\log \frac{1}{\Delta_{0,1}}$, we adopt the current expression for further comparison. The full proof is in Appendix B.2.

*Sketch Proof of Theorem 5.3.* The main idea is to split $\tau = \tau^{\text{ee}} + \tau^{\text{et}}$, and we derive upper bounds for both $\mathbb{E}\tau^{\text{ee}}$ and $\mathbb{E}\tau^{\text{et}}$. In the following, we only summarize the main steps of upper bounding $\mathbb{E}\tau^{\text{ee}}$, when instance $\nu$ is positive. Proofs for other cases are similar.

Define $L' = \lceil \log_2 \frac{24(C+1)^2 H_1^{\text{pos}}}{K} \rceil$, $L'' = \lceil \log_2 \frac{192(C+1)^2}{\omega^2 \Delta_{0,1}^2} \rceil$, where $\omega = \frac{C-1}{C+3}$. Via routine calculations, we have $L'' \geq L'$. To prove Theorem 5.3, the most important intermediate step is to show that in phase $k \geq \max\{\kappa^{\text{ee}}, L'\}$, Algorithm 3 always outputs $\hat{a} \in [K]$, $\mu_{\hat{a}} > \omega\mu_1 + (1 - \omega)\mu_0$ with certainty. In addition,

$$\tau_k^{\text{ee}} \leq O\left( K\beta_{\max\{\kappa^{\text{ee}}, L'\}-1} \log \frac{4K}{\delta_{\max\{\kappa^{\text{ee}}, L'\}-1}} \right) + O\left( \sum_{a=1}^K \frac{\log \frac{K}{\delta_k} + \log\log \frac{1}{\max\{\Delta_{1,a}^2, \Delta_{0,a}^2\}}}{\max\{\Delta_{1,a}^2, \Delta_{0,a}^2\}} \right). \quad (4)$$

The intuition is as follows. Firstly, at the end of phase $\max\{\kappa^{\text{ee}}, L'\} - 1$, we have

$$\tau_{\max\{\kappa^{\text{ee}}, L'\}-1}^{\text{ee}} \leq O\left( K\beta_{\max\{\kappa^{\text{ee}}, L'\}-1} \log \frac{4K}{\delta_{\max\{\kappa^{\text{ee}}, L'\}-1}} \right).$$

This is guaranteed by the Lemma 4.2. Then, starting from phase $\max\{\kappa^{\text{ee}}, L'\}$, $T_k^{\text{ee}}$ is large enough, such that Algorithm 3 will not enter the case of line 8. By induction, we have

$$N_a(\tau_k^{\text{ee}}) \leq \max \left\{ N_a(\tau_{\max\{\kappa^{\text{ee}}, L'\}-1}^{\text{ee}}), \right.$$

$$\left. O\left( \frac{\log \frac{K}{\delta_k} + \log\log \frac{1}{\max\{\Delta_{1,a}^2, \Delta_{0,a}^2\}}}{\max\{\Delta_{1,a}^2, \Delta_{0,a}^2\}} \right) \right\}$$

holds for all $k \geq \max\{\kappa^{\text{ee}}, L'\}, a \in [K]$ with certainty. Summing up the above inequality for all $a \in [K]$ and using the fact that $\tau_k^{\text{ee}} \leq K + \sum_{a=1}^K N_a(\tau_k^{\text{ee}})$ (Lemma 4.2), we complete the proof of (4). Lemmas B.4, B.6, B.8 contain more details.

The next step is to show conditioned on $\mu_{\hat{a}_k} \geq \omega\mu_1 + (1 - \omega)\mu_0$, $k \geq \max\{\kappa^{\text{et}}, L''\}$ can guarantee the exploitation period output Qualified, and the algorithm must stop. This is straightforward, as $k \geq \kappa^{\text{et}}$ guarantees $U(N_{\hat{a}_k}^{\text{et}}, \frac{\delta}{K\alpha_k}) < \frac{\omega(\mu_1 - \mu_0)}{2}$ implies $\frac{\sum_{s=1}^{N_{\hat{a}_k}^{\text{et}}} X_{\hat{a}_k, s}^{\text{et}}}{N_{\hat{a}_k}^{\text{et}}} - U(N_{\hat{a}_k}^{\text{et}}, \frac{\delta}{K\alpha_k}) > \mu_0$. Meanwhile, having $k \geq L''$ guarantees $T_k^{\text{et}}$ is large enough such that Algorithm 4 will not quit the While loop before $N_{\hat{a}_k}^{\text{et}}$ is large enough such that $U(N_{\hat{a}_k}^{\text{et}}, \frac{\delta}{K\alpha_k}) < \frac{\omega(\mu_1 - \mu_0)}{2}$ holds. Combining these intermediate steps, we have

$$\tau^{\text{ee}}$$
$$\leq O\left( K\beta_{\max\{\kappa^{\text{ee}}, L'\}} \log \frac{4K}{\delta_{\max\{\kappa^{\text{ee}}, L'\}}} \right) +$$
$$O\left( \sum_{a=1}^K \frac{\log \frac{K}{\delta_{\max\{\kappa^{\text{ee}}, \kappa^{\text{et}}, L', L''\}}} + \log\log \frac{1}{\max\{\Delta_{1,a}^2, \Delta_{0,a}^2\}}}{\max\{\Delta_{1,a}^2, \Delta_{0,a}^2\}} \right)$$
$$\leq O\left( H_1^{\text{pos}} \log H_1^{\text{pos}} \right) + O\left( K\beta_{\kappa^{\text{ee}}} \log \frac{4K}{\delta_{\kappa^{\text{ee}}}} \right) +$$
$$O\left( K\beta_{L'} \log \frac{4K}{\delta_{L'}} \right) + O\left( H_1^{\text{pos}} \log \frac{1}{\delta_{\kappa^{\text{ee}}}} \right) +$$
$$O\left( H_1^{\text{pos}} \log \frac{1}{\delta_{L''}} \right) + O\left( H_1^{\text{pos}} \log \frac{1}{\delta_{\kappa^{\text{et}}}} \right) +$$
$$O\left( H_1^{\text{pos}} \log \frac{1}{\delta_{L'}} \right)$$

holds with certainty. Take expectation on both sides, we can use inequalities such as

$$\mathbb{E}\beta_{\kappa^{\text{ee}}} \log \frac{4K}{\delta_{\kappa^{\text{ee}}}} \leq \sum_{k=1}^{+\infty} \beta_k \log \frac{4K}{\delta_k} \Pr(\kappa^{\text{ee}} = k)$$
$$\stackrel{\text{Lemma 5.1}}{\leq} \sum_{k=1}^{+\infty} \frac{\pi^2 \delta_{k-1}}{6} \cdot \beta_k \log \frac{4K}{\delta_k} = O(\log K)$$

to derive the upper bound of $\mathbb{E}\tau^{\text{ee}}$. $\square$

Combining Theorems 5.2 and 5.3, we know Algorithm 2 is $(\Delta, \delta)$-PAC.

## 5.2. Lower Bounds

We derive lower bounds for both positive and negative instances based on techniques in existing works. Without extra description, we assign unit variance Gaussian noise for each of the arm in the constructed instances. The results in (Garivier & Kaufmann, 2016; Degenne & Koolen, 2019) can be adopted to show the following:

**Theorem 5.4.** *For a unit variance Gaussian instance equipped with mean reward vector $\{\mu_a\}_{a=1}^K$, $\mu_0 > \max_{1 \leq a \leq K} \mu_a$, any $\delta$-PAC 1-identification algorithm* **alg** *would satisfy $\mathbb{E}_{\text{alg}} \tau \geq \Omega(H_1^{\text{neg}} \log(1/\delta))$.*

Detailed proof can be found in Appendix C.1. Comparing with the upper bound in Theorem 5.3, the gap between the upper and lower bounds in the negative case is up to a constant and a polynomial logarithm factor polylog$(H_1^{\text{neg}})$ in the $\delta$ independent part. We also derive a lower bound for a positive instance, based on the analyses in (Degenne & Koolen, 2019) and (Katz-Samuels & Jamieson, 2020).

**Theorem 5.5.** *Consider any $\{\mu_a\}_{a=0}^K \in [0,1]^{K+1}$ satisfying $\mu_1 \geq \mu_2 \geq \cdots \geq \mu_K$, $\mu_1 > \mu_0$. Consider instance $\nu$ that takes a permutation of $\{\mu_a\}_{a=1}^K$ as the reward vector, and $\mu_0$ as the threshold. Then, for any $\delta$-PAC 1-identification* **alg**, *any $\delta \in (0,1)$, we have $\mathbb{E}_{\nu,\text{alg}} \tau \geq \Omega(H \log \frac{1}{\delta}) = \Omega(\frac{\log \frac{1}{\delta}}{\Delta_{0,1}^2})$.*

**Theorem 5.6.** *Consider any $\{\mu_a\}_{a=0}^K \in [0,1]^{K+1}$ satisfying $\mu_1 > \mu_0 \geq \mu_2 \geq \cdots \geq \mu_K$. For any $\delta$-PAC 1-identification* **alg**, *any $\delta \in (0, \frac{1}{16})$, we can find a positive instance $\nu$ whose mean reward vector is a permutation of vector $\{\mu_a\}_{a=1}^K$ and the threshold is $\mu_0$, such that $\mathbb{E}_{\nu,\text{alg}} \tau \geq \Omega(H_1)$.*

The full proof is in Appendix C.2. The last step is to combine Theorem 5.5 and 5.6. If we assume $\mu_1 > \mu_0 \geq \mu_2 \geq \cdots \mu_K$, for any $\delta$-PAC algorithm, we can find an instance $\nu$ taking $\{\mu_a\}_{a=1}^K \in [0,1]^K$ as a permutation of reward vector, such that $\mathbb{E}_\nu \tau \geq \Omega\left(\frac{\log \frac{1}{\delta}}{\Delta_{0,1}^2} + \sum_{a=2}^K \frac{1}{\max\{\Delta_{1,a}^2, \Delta_{0,a}^2\}}\right)$. The reason is $\Delta_{1,a} > \Delta_{0,a}$ holds for all $a \geq 2$. Together with Theorem 5.3, we deduce that the gap between upper and lower bounds is at most a factor poly$(\log \frac{1}{\Delta_{0,1}}, \{\log \frac{K}{\max\{\Delta_{0,a}, \Delta_{1,a}\}}\}_{a=2}^K)$, if the mean reward vector satisfies $\mu_1 > \mu_0 \geq \mu_2 \cdots \geq \mu_K$. Theorem 5.3 guarantees our bound's dependence on $\log 1/\delta$ is nearly optimal, but it still remains unclear what would be a tight upper and lower bound of the $\delta$ independent part, if there are multiple arms above $\mu_0$.

Besides the lower bounds for the total expected pulling times, we also derive a lower bound for the pulling times of arm $a$ whose $\mu_a < \mu_0$. The following theorem can only imply the possibility that the $O(\log(1/\Delta_{0,1}^2))$ in Theorem 5.3 is required.[2]

**Theorem 5.7.** *Fixed any $\mu_0 \in [0,1]$, $\{\mu_a\}_{a=2}^K \in [0,1]^{K-1}$ satisfying $\mu_0 > \mu_2 \geq \cdots \geq \mu_K$. For any $(\Delta, \delta)$-PAC 1-identification* **alg**, *any $\delta < 1/e^8$, we can find a small enough $\bar{\Delta}_{0,1} > 0$, such that for any $\mu_1 \in (\mu_0, \mu_0 + \bar{\Delta}_{0,1}]$, we can find a problem instance $\nu$ whose mean reward vec-*

*tor is $\{\mu_a\}_{a=1}^K$ and the* **alg** *must satisfy $\mathbb{E}_{\nu,\text{alg}} N_a(\tau) \geq \Omega(\log(1/\Delta_{1,0}^2)/\Delta_{1,a}^2), \forall a \geq 2$.*

The full proof is in Appendix C.3. Theorem 5.7 implies the expected pulling times of unqualified arms would be impacted by the gap between the best arm and the threshold. Nevertheless, we caution that this impact might only occur when $\Delta_{1,0}$ is sufficiently small. If $\Delta_{1,0}$ is sufficiently small, it is possible $\sum_{a=2}^K \frac{\log(1/\Delta_{1,0}^2)}{\Delta_{1,a}^2} < \frac{1}{\Delta_{1,0}^2}$. Thus, the $O(\log(1/\Delta_{1,0}^2))$ term might not occur in the upper bound of $\mathbb{E}\tau$, since the upper bound of $\mathbb{E}\tau$ usually contains $O(1/\Delta_{1,0}^2)$.

# 6. Numerical Experiments

We conduct numerical evaluations on SEE and existing benchmark algorithms on synthetic instances. The benchmark algorithms include HDoC, LUCB_G in (Kano et al., 2017), lilHDoC in (Tsai et al., 2024), Murphy Sampling (MS) in (Kaufmann et al., 2018) and TaS in (Garivier & Kaufmann, 2016). Algorithm MS and TaS are not originally designed for 1-identification, but we adapt these algorithms by applying the GLR stopping rule (see Lemma 2 in (Jourdan & Réda, 2023)). In what follows, we call these adapted versions as adapted-MS and adapted-TaS respectively. While adapted-MS and adapted-TaS are still $\delta$-PAC, there is no non-trivial theoretical guarantees on their sample complexity bounds. Hence, these adapted algorithms are heuristic. In Appendix E.3, we discuss APGAI in (Jourdan & Réda, 2023) on its numerical performance, which presents a significantly different trend from others.

We do not include algorithms S-TaS (Degenne & Koolen, 2019) and APT_G (Kano et al., 2017) in the benchmark algorithm list, since the performance of algorithm S-TaS heavily relies on the position of the qualified arm in the positive case, as shown by (Jourdan & Réda, 2023). And (Kano et al., 2017) show that empirically APT_G performs poorly compared to the HDoC, LUCB_G.

We consider six different groups of reward vectors, which are "All Worse", "Unique Qualified", "One Quarter", "Half Good", "All Good" and "Linear". The main difference among these groups is the number of qualified arms. All instances in "All Worse" group are negative, whose mean rewards of all arms are below $\mu_0$. Due to the low execution speed of algorithm MS on group "All Worse", we do not apply MS to the instance "All Worse". Other groups of instances are positive, meaning that there is at least one arm whose mean reward is $> \mu_0$. We set $10^8$ as our forced stopping threshold for each group of experiments. Every algorithm stops and outputs an answer at or before the number of arm pulls reaches $10^8$. The details of the numerical setting, tuning of the hyper-parameters in SEE, and other implementation details can be found in Appendix E.1.

---

[2]We do not find any related results in the literature, which motivates us to state the partial result in what follows.

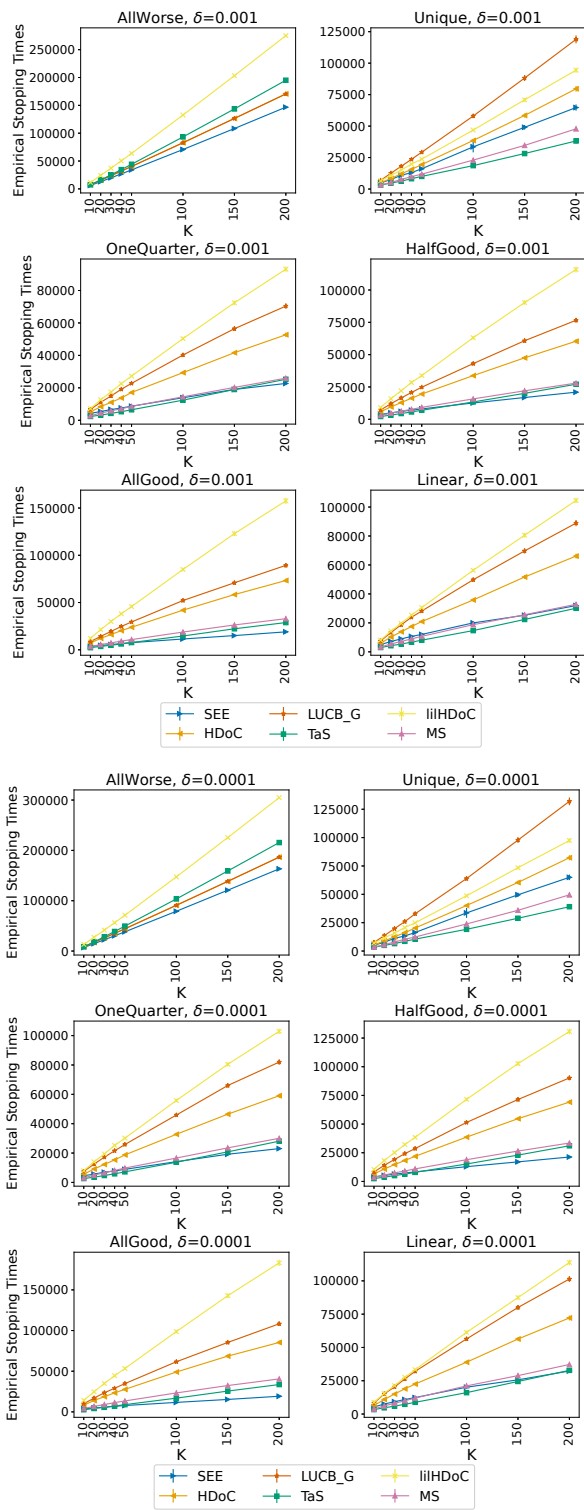

*Figure 1.* Numerical Experiments on SEE and Benchmarks

Figure 1 illustrates the numerical results. In "AllWorse", our proposed algorithm SEE outperforms all the benchmarks, and the advantage of SEE becomes more obvious as $K$ increases. In "One Quarter", "Half Good", "All Good", "Linear ", adapted-TaS and adapted-MS outperform SEE when $K \leq 50$. However, as $K$ increases, the leading gap of adapted-TaS and adapted-MS becomes smaller. When $K = 200$, SEE outperforms adapted-TaS and adapted-MS in instances "One Quarter", "Half Good", "All Good". In instances "Unique Qualified", adapted-TaS and adapted-MS consistently have better numerical performance. In comparison with benchmark algorithms in (Kano et al., 2017), our proposed SEE outperforms HDoC, lilHDoC and LUCB_G in most of the instances, except $K = 10, 20$ in the "Unique Qualified" group. When $\delta$ gets smaller, the above phenomenon is more pronounced. Altogether, SEE is competitive among all the algorithms with theoretical claims.

The numerical performance is consistent with our theoretical analysis. As Theorem 5.3 suggests, if we apply SEE to a positive instance, the empirical stopping times increase proportionally to $K$. However, our dependence on $K$ is better than the existing algorithms' sample complexity bounds, in the sense that the coefficient of the $\delta$ in our sample complexity bound is independent of $K$ on positive instances. This property does not hold for the sample complexity upper bounds for algorithms HDoC and lilHDoC. The above provides intuition on a larger leading gap of SEE in the case of smaller $\delta$.

In our numerical experiments, adapted-MS and adapted-TaS have good performances, especially in the case of a small arm number. However, from figure 2 in the appendix E.2, it can be observed that their empirical stopping times are not monotonically decreasing as the proportion of the qualified arm increases. In contrast, SEE performs better when the proportion increases, which suggests the stability and robustness of our proposed SEE.

## 7. Conclusion

We study the 1-identification problem in the fixed confidence setting, and design a new algorithm SEE. We establish a non-asymptotic sample complexity upper bound for $\mathbb{E}\tau$ under SEE, which is nearly optimal in the negative instance and positive instance with a unique qualified arm. The superior performance of SEE is also corroborated by our numerical experiments.

## Acknowledgment

This research work is funded by a Singapore Ministry of Education AcRF Tier 2 grant (A-8003063-00-00).

## Impact Statement

This paper presents work whose goal is to advance the field of Machine Learning. There are many potential societal consequences of our work, none which we feel must be specifically highlighted here.

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

# A. Additional Literature Review

## A.1. Additional Review on (Katz-Samuels & Jamieson, 2020) and Others

(Katz-Samuels & Jamieson, 2020) put forward FWER-FWPD (Algorithm 2) to solve their proposed $k$-identification problem, which is slightly different from the definition in this paper. FWER-FWPD is an anytime algorithm, returning a subset $\mathcal{R}_t \subset [K]$ at each round $t$, and never stops. The set $\mathcal{R}_t$ contains all the arms whose mean reward are believed to be greater than $\mu_0$ up to time $t$. The set $\mathcal{R}_1$ is initialized as the empty set. Algorithm FWER-FWPD sequentially adds more and more arms into $\mathcal{R}_t$. While (Katz-Samuels & Jamieson, 2020) do not provide explicit stopping time $\tau$ in their algorithms, in their analysis they consider another sequence of stopping times $\lambda_k = \min\{t : |\mathcal{R}_t \cap \{a : \mu_a > \mu_0\}| \geq k\}$, denoting the first round index whose $\mathcal{R}_t$ contains at least $k$ qualified arms. Since the set $\{a : \mu_a > \mu_0\}$ is unknown to the agent, Algorithm FWER-FWPD is unable to figure out the exact value of $\{\lambda_k\}$, which is called *Unverifiable Stopping Time*. It is evident that $\lambda_k$ is equal to infinity, if $\{a : \mu_a > \mu_0\} = \emptyset$. In this case, the algorithm FWER-FWPD may never stop with keeping $\mathcal{R}_t = \emptyset, \forall t \in \mathbb{N}$, meaning that the Algorithm FWER-FWPD cannot handle a negative instance $\nu$. For this reason, we do not conduct a direct comparison between the FWER-FWPD and the algorithms in table 1.

If we apply the Algorithm FWER-FWPD to a positive instance $\nu$, we can still derive a conclusion by adapting the upper bound of $\mathbb{E}\lambda_1$. Theorem 8 in (Katz-Samuels & Jamieson, 2020) guarantee that $\Pr(\exists t, \mathcal{R}_t \cap \{a : \mu_a < \mu_0\}) < 10\delta$. Thus, we can consider stopping time $\tau = \min\{t : \mathcal{R}_t \neq \emptyset\}$ as the termination round in our formulation. Taking $\hat{a}$ as the unique element in $\mathcal{R}_\tau$, we have $\Pr(\mu_{\hat{a}} < \mu_0) \leq \Pr(\exists t, \mathcal{R}_t \cap \{a : \mu_a < \mu_0\}) < 10\delta$. It is evident that $\tau \leq \lambda_1$. By the upper bound mentioned in Theorem 8 in (Katz-Samuels & Jamieson, 2020) we have

$$\mathbb{E}\lambda_1 \leq O\left(\left((\frac{K}{m}-1)\cdot\frac{\log\frac{1}{\delta}}{\Delta^2}+\log\frac{\frac{K}{m}}{\delta}\right)\log\left((\frac{K}{m}-1)\cdot\frac{\log\frac{1}{\delta}}{\Delta^2}+\log\frac{\frac{K}{m}}{\delta}\right)\right),$$

The above bound can be taken as an upper bound of $\mathbb{E}\tau$ of the FWER-FWPD Algorithm in (Katz-Samuels & Jamieson, 2020) coupled with the stopping time $\lambda_1$. The sub-optimality of the above upper bound mainly comes from two parts. Firstly, the dependence on $\delta$ is $\log\frac{1}{\delta}\log\log\frac{1}{\delta}$ instead of the commonly seen result $\log\frac{1}{\delta}$. Then, in the asymptotic regime, i.e. $\delta \to 0$, this upper bound is larger than all the upper bounds on positive instance in table 1. Secondly, the coefficient of $\log\frac{1}{\delta}\log\log\frac{1}{\delta}$ is the inverse minimum gap between qualified arms and $\mu_0$. In the case $\Delta << \Delta_{1,0}$, the upper bound is much larger than the lower bound, the upper bound becomes loose. This theoretical upper bound remains competitive only when the number of qualified arm $m$ is proportional to $K$, and the $\delta$ is considered a constant. These are also the assumptions adopted by (Katz-Samuels & Jamieson, 2020).

There are research works aiming to output all the arms better than the threshold $\mu_0$. As previously mentioned, (Kano et al., 2017) aim to output all the arm sequentially and its first outputting round is indeed the $\tau$. Besides the fixed confidence setting adopted by the above papers, (Locatelli et al., 2016; Mukherjee et al., 2017) work on the fixed budget setting, which aims to maximize the probability of correct output given a finite number of rounds.

Another related topic is $\epsilon$-Good Arm Identification. The agent needs to find all the arms (Mason et al., 2020) in $\{a : \mu_a \geq \mu_1 - \epsilon\}$ or only one arm (Even-Dar et al., 2002; Gabillon et al., 2012; Kalyanakrishnan et al., 2012; Kaufmann & Kalyanakrishnan, 2013; Katz-Samuels & Jamieson, 2020). In the second case, if the gap $\Delta_{0,1}$ is known to us, then the $\epsilon-$Good Arm Identification and 1-identification are equivalent by taking $\epsilon = \Delta_{0,1}$. But it is unclear whether equivalence still holds if the largest mean reward of $\{\mu_a\}_{a=0}^K$ is unknown. Besides the fixed confidence setting, (Zhao et al., 2023) works on simple regret, under the fixed budget setting.

## A.2. Comments on Table 1

Due to the space limit, we only present imprecise upper bounds for algorithm HDoC and APGAI in the table 1. Their actual upper bounds are larger, still suffering from the suboptimality discussed in the section 3. Following are more details.

Regarding algorithm HDoC, Theorem 3 in (Kano et al., 2017) proves for $\nu \in \mathcal{S}^{\text{pos}}$, we have

$$\mathbb{E}_\nu\tau \leq n_1 + \sum_{a=2}^{K}\left(\frac{\log(K\max_{j\in[K]} n_j)}{2(\Delta_{1,a}-2\epsilon)^2}+\delta n_a\right)+\frac{K^{2-\frac{\epsilon^2}{(\min_{a\in[K]}\Delta_{0,a}-\epsilon)^2}}}{2\epsilon^2}+\frac{K(5+\log\frac{1}{2\epsilon^2})}{4\epsilon^2},$$

where $n_a = \frac{1}{(\Delta_{0,a}-\epsilon)^2}\log\left(\frac{4\sqrt{K/\delta}}{(\Delta_{0,a}-\epsilon)^2}\log\frac{5\sqrt{K/\delta}}{(\Delta_{0,a}-\epsilon)^2}\right)$, $\epsilon = \min\left\{\min_{a\in[K]}\Delta_{0,a}, \min_{a\in[K-1]}\frac{\Delta_{a,a+1}}{2}\right\}$. It is easy to see $n_1 \geq$

$\Omega(\frac{\log K/\delta}{\Delta_{0,1}^2})$, $\frac{K(5+\log \frac{1}{2\epsilon^2})}{4\epsilon^2} \geq \Omega(\frac{K}{\epsilon^2})$. In addition, $\log(K \max_{j\in[K]} n_j) \geq \Omega(\log \log \frac{1}{\delta})$, suggesting that

$$\sum_{a=2}^{K} \left( \frac{\log(K \max_{j\in[K]} n_j)}{2(\Delta_{1,a} - 2\epsilon)^2} \right)$$
$$\geq \Omega \left( \sum_{a=2}^{K} \frac{\log \log \frac{1}{\delta}}{2(\Delta_{1,a} - 2\epsilon)^2} \right)$$
$$\geq \Omega \left( \sum_{a=2}^{K} \frac{\log \log \frac{1}{\delta}}{2\Delta_{1,a}^2} \right)$$
$$\geq \Omega(H_1 \log \log \frac{1}{\delta}).$$

Thus, we can conclude

$$\Omega \left( H \log \frac{K}{\delta} + H_1 \log \log \frac{1}{\delta} + \frac{K}{\epsilon^2} \right)$$
$$\leq n_1 + \sum_{a=2}^{K} \left( \frac{\log(K \max_{j\in[K]} n_j)}{2(\Delta_{1,a} - 2\epsilon)^2} + \delta n_a \right) + \frac{K^{2 - \frac{\epsilon^2}{(\min_{a\in[K]} \Delta_{0,a} - \epsilon)^2}}}{2\epsilon^2} + \frac{K(5 + \log \frac{1}{2\epsilon^2})}{4\epsilon^2},$$

and the actual upper bound for $\nu \in \mathcal{S}^{\text{pos}}$ still suffers from the suboptimality from two perspectives mentioned in the section 3. The term $\frac{1}{\epsilon^2}$ implies the upper bound can be infinity. And the term $H_1 \log \log(1/\delta)$ suggests there is still gap between the upper and lower bounds.

Regarding algorithm APGAI, (Jourdan & Réda, 2023) do not explicitly provide upper bound for $\mathbb{E}\tau$. Therorem 2 in (Jourdan & Réda, 2023) suggests

$$\mathbb{E}_\nu \tau \leq C^{\text{pos}}(\delta) + \frac{K\pi^2}{6} + 1, \nu \in \mathcal{S}^{\text{pos}}$$
$$\mathbb{E}_\nu \tau \leq C^{\text{neg}}(\delta) + \frac{K\pi^2}{6} + 1, \nu \in \mathcal{S}^{\text{neg}},$$

with definition

$$C^{\text{pos}}(\delta) = \sup \left\{ t : t \leq \left( \sum_{a:\mu_a \geq \mu_0} \frac{2}{\Delta_{0,a}^2} \right) (\sqrt{c(t,\delta)} + \sqrt{3 \log t})^2 + D^{\text{pos}}(\{\mu_a\}_{a=1}^K) \right\}$$
$$C^{\text{neg}}(\delta) = \sup \left\{ t : t \leq \left( \sum_{a=1}^{K} \frac{2}{\Delta_{0,a}^2} \right) (\sqrt{c(t,\delta)} + \sqrt{3 \log t})^2 + D^{\text{neg}}(\{\mu_a\}_{a=1}^K) \right\}$$
$$2c(t,\delta) = \bar{W}_{-1} \left( 2 \log \frac{K}{\delta} + 4 \log \log(e^4 t) + \frac{1}{2} \right)$$
$$\bar{W}_{-1}(x) = -W_{-1}(-e^{-x}) \approx x + \log x.$$

$W_{-1}$ is the negative branch of the Lambert W function. And $D^{\text{pos}}(\{\mu_a\}_{a=1}^K)$, $D^{\text{neg}}(\{\mu_a\}_{a=1}^K)$ are further defined in the appendix F in (Jourdan & Réda, 2023). From the above definition, we know $c(t,\delta) \geq \Omega(\log \frac{K}{\delta})$, suggesting that $(\sqrt{c(t,\delta)} + \sqrt{3 \log t})^2 \geq \Omega(\log \frac{K}{\delta})$. Further, we can conclude $C^{\text{pos}}(\delta) \geq \Omega(H_0(\log \frac{K}{\delta}))$ and $C^{\text{pos}}(\delta) \geq \Omega(H_1^{\text{neg}}(\log \frac{K}{\delta}))$. Thus, if there exists $a \in [K], \mu_a = \mu_0$ for an instance $\nu \in \mathcal{S}^{\text{pos}}$, $C^{\text{pos}}(\delta) = \infty$, resulting in a vacuous upper bound.

### A.3. lilHDoC, HDoC, APGAI are not $(\Delta, \delta)$-PAC

In this subsection, we are going to show lilHDoC in (Tsai et al., 2024), HDoC in (Kano et al., 2017), APGAI in (Jourdan & Réda, 2023) are not $(\Delta, \delta)$-PAC. To do this, we suffice to construct an instance $\nu \in \mathcal{S}^{\text{pos}}$, such that $\mathbb{E}_\nu \tau = +\infty$ holds for all these three algorithms.

Consider a two-arm instance $\nu$ with $\mu_1 > \mu_0 = \mu_2$. Arm 1 follows unit variance Gaussian and arm 2 returns constant reward. Easy to validate $\nu \in \mathcal{S}^{\text{pos}}$.

Algorithm lilHDoC applies uniform sampling on 2 arms, by pulling each arm $T$ times. With non-zero prob p, $\text{UCB}_1(t) < \mu_0$ holds for all $t \leq T$. Then, arm 1 will get removed from the arm set, so as algorithm HDoC. In this case, both lilHDoC and HDoC will keep pulling arm 2 without termination. In other words, we find an event with positive probability, such that $\tau = +\infty$. We can conclude $\mathbb{E}_{\nu,\text{HDoC}}\tau = +\infty, \mathbb{E}_{\nu,\text{lilHDoC}}\tau = +\infty$.

The same idea also applies to the algorithm APGAI. With non-zero probability, the first collected sample from arm 1 is below $\mu_0$. Conditioned on this event, APGAI will keep pulling arm 2 without a termination, meaning that $\mathbb{E}_{\nu,\text{APGAI}}\tau = +\infty$.

# B. Analysis of Algorithm 2

## B.1. Selection Rules of $C, \{\delta_k, \alpha_k, \beta_k\}_{k=1}^{+\infty}$

Regarding $C$, any $C > 1$ is available for the algorithm analysis. And $C$ will only impact the constant coefficient in the upper bound mentioned in Theorem 2, which is ignored as $C$ is predetermined.

Regarding $\{\delta_k, \alpha_k, \beta_k\}_{k=1}^{+\infty}$, there are some restrictions on the decreasing/increasing speed of the sequence. Generally speaking, we require $\{\delta_k\}_{k=1}^{+\infty}$ is a decreasing sequence with limit zero and $\{\alpha_k, \beta_k\}_{k=1}^{+\infty}$ are increasing sequences with limit infinity. And they should fulfill following properties

$$\beta_{k+1} \geq 2\beta_k, \beta_{k+1} \log \frac{1}{\delta_{k+1}} \geq 2\beta_k \log \frac{1}{\delta_k}$$

$$\sum_{k=1}^{+\infty} \delta_{k-1}\beta_k \log \frac{\alpha_k}{\delta_k} < +\infty$$

$$\sum_{k=1}^{+\infty} \frac{1}{\alpha_{k-1}}\beta_k \log \frac{\alpha_k}{\delta_k} < +\infty$$

$$\sum_{k=1}^{+\infty} \frac{1}{\alpha_k} \leq \frac{1}{4}.$$

With out loss of generality, we can take $\delta_0 = \beta_0 = \alpha_0 = 1$. Considering all these requirements, taking $\delta_k = \frac{1}{3^k}, \beta_k = 2^k, \alpha_k = 5^k$ is a suitable choice. It is also possible to find other sequences.

## B.2. Proof of Main Theorems

Before illustrating the proof of Theorem 5.2 and 5.3, we need some preparations. Section B.3 includes all the required lemmas to complement the proof.

We firstly prove Theorem 5.2, which asserts Algorithm 2 is $\delta$-PAC. Recall the definition of of $\delta$-PAC, we need to prove $\Pr_\nu(\tau < +\infty, \hat{a} \in i^*(\nu)) > 1 - \delta$ holds for any $\delta \in (0, 1)$ and $\nu \in \mathcal{S}^{\text{pos}} \cup \mathcal{S}^{\text{neg}}$. Here we split the proof into two steps. The first part is to show $\Pr(\tau < +\infty) = 1$, which is guaranteed by Lemma B.2. Then, given the first step, we just need to show $\Pr_\nu(\hat{a} \in i^*(\nu)) > 1 - \delta$. Equivalently, suffice to prove $\Pr_\nu(\hat{a} \notin i^*(\nu)) < \delta$ holds for any $\delta \in (0, 1)$ and $\nu \in \mathcal{S}^{\text{pos}} \cup \mathcal{S}^{\text{neg}}$.

*Proof of Theorem 5.2.* By the Lemma B.2, we know $\tau < +\infty$ with certainty. Thus, $\hat{a} \in [K] \cup \{\text{None}\}$ in Algorithm 2 is well defined with certainty.

For positive instance $\nu$,

$$\begin{aligned}
&\Pr(\hat{a} \notin i^*(\nu)) \\
&\leq \Pr(\hat{a} = \text{None}) + \Pr(\hat{a} \in [K], \mu_{\hat{a}} < \mu_0) \\
&\leq \Pr\left(\exists t \in \mathbb{N}, \exists \delta_k < \frac{\delta}{3}, \frac{\sum_{s=1}^t X_{1,s}^{\text{ee}}}{t} + U(t, \frac{\delta_k}{K}) < \mu_0\right) +
\end{aligned}$$

$$\Pr\left(\exists t \in \mathbb{N}, \exists a \in [K], \mu_a < \mu_0, \frac{\sum_{s=1}^{t} X_{a,s}^{\mathrm{et}}}{t} - U(t, \frac{\delta}{K\alpha_1}) > \mu_0\right)$$

$$\leq \Pr\left(\exists t \in \mathbb{N}, \frac{\sum_{s=1}^{t} X_{1,s}^{\mathrm{ee}}}{t} + U(t, \frac{\delta}{3K}) < \mu_0\right) +$$

$$\Pr\left(\exists t \in \mathbb{N}, \exists a \in [K], \mu_a < \mu_0, \frac{\sum_{s=1}^{t} X_{1,s}^{\mathrm{et}}}{t} - U(t, \frac{\delta}{K\alpha_1}) > \mu_a\right)$$

$$\leq \frac{\pi^2}{6}(\frac{\delta}{3} + \frac{\delta}{5}) < \delta.$$

For negative instance $\nu$,

$$\Pr(\hat{a} \notin i^*(\nu))$$
$$= \Pr(\hat{a} \in [K])$$
$$\leq \Pr\left(\exists t \in \mathbb{N}, \exists a \in [K], \mu_a < \mu_0, \frac{\sum_{s=1}^{t} X_{a,s}^{\mathrm{et}}}{t} - U(t, \frac{\delta}{K\alpha_1}) > \mu_0\right)$$
$$\leq \frac{\delta}{5}\frac{\pi^2}{6} < \delta.$$

$\square$

To prove Theorem 5.3, i.e the upper bound of $\mathbb{E}\tau$, we need more preparations. As the sketch proof in section 5 illustrated, the most important step is to find the upper bound of phase index such that that the Algorithm 3 starts to output correct answer with an appropriate upper bound of $\tau_k^{\mathrm{ee}}$.

Since the proof is too long, we split the proof of Theorem 5.3 into two parts. In the first part, we prove the upper bound $\mathbb{E}\tau \leq O\left(\frac{\log \frac{1}{\delta}}{\Delta_{0,1}^2} + (\log \frac{K}{\Delta_{0,1}^2}) \cdot H_1^{\mathrm{pos}}\right)$, for the case $\nu \in \mathcal{S}^{\mathrm{pos}}$. In the second part, we prove the upper bound $\mathbb{E}\tau \leq O(H_1^{\mathrm{neg}}(\log \frac{1}{\delta} + \log H_1^{\mathrm{neg}}))$, for the case $\nu \in \mathcal{S}^{\mathrm{neg}}$. Lemma B.4, B.6 and B.8 are required for the case $\nu \in \mathcal{S}^{\mathrm{pos}}$. And Lemma B.5, B.7 and B.9 are required for the case $\nu \in \mathcal{S}^{\mathrm{neg}}$.

*Proof of Theorem 5.3, positive case.* Denote $L' = \lceil \log_2 \frac{24(C+1)^2 H_1^{\mathrm{pos}}}{K} \rceil$, $L'' = \lceil \log_2 \frac{192(C+1)^2}{\omega^2 \Delta_{0,1}^2} \rceil$, where $\omega = \frac{C-1}{C+3}$. Easy to see $L' \leq L''$. We split $\tau = \tau^{\mathrm{ee}} + \tau^{\mathrm{et}}$ and derive an upper bound for $\mathbb{E}\tau^{\mathrm{ee}}$ and $\mathbb{E}\tau^{\mathrm{et}}$ separately.

Since we take $\beta_k = 2^k, \delta_k = \frac{1}{3^k}$, we have

$$\beta_{L'} \leq 2^{\log_2 \frac{24(C+1)^2 H_1^{\mathrm{pos}}}{K} + 1} = 2 \cdot \frac{24(C+1)^2 H_1^{\mathrm{pos}}}{K} = O\left(\frac{H_1^{\mathrm{pos}}}{K}\right)$$

$$\log \frac{1}{\delta_{L'}} \leq \log\left(3^{\log_2 \frac{24(C+1)^2 H_1^{\mathrm{pos}}}{K} + 1}\right) = \log 3 \cdot \left(\log_2 \frac{24(C+1)^2 H_1^{\mathrm{pos}}}{K} + 1\right) = O\left(\log \frac{H_1^{\mathrm{pos}}}{K}\right)$$

$$\beta_{L''} \leq 2^{1 + \log_2 \frac{192(C+1)^2}{\omega^2 \Delta_{0,1}^2}} \leq 2 \cdot \frac{192(C+1)^2}{\omega^2 \Delta_{0,1}^2} = O\left(\frac{1}{\Delta_{0,1}^2}\right)$$

$$\log \frac{1}{\delta_{L''}} \leq \log\left(3^{1 + \log_2 \frac{192(C+1)^2}{\omega^2 \Delta_{0,1}^2}}\right) = \log 3 \left(1 + \log_2 \frac{192(C+1)^2}{\omega^2 \Delta_{0,1}^2}\right) = O\left(\log \frac{1}{\Delta_{0,1}^2}\right)$$

We ignore constant $C$ as it is a predetermined constant in the Algorithm 2.

By the Lemma B.1, B.8, we know the exploration period after phase $\max\{\kappa^{\mathrm{ee}}, L'\}$ will always output $\hat{a} \in [K]$, $\mu_{\hat{a}} > \omega\mu_1 + (1-\omega)\mu_0$, and the forced termination at line 8 never triggers. By the Lemma B.3, we know the exploitation period will return "Qualified" after phase $\max\{\kappa^{\mathrm{et}}, L''\}$, conditioned on the event $\mu_{\hat{a}_k} > \omega\mu_1 + (1-\omega)\mu_0$. In summary, the algorithm must terminate no late than the end of phase $\max\{\kappa^{\mathrm{ee}}, \kappa^{\mathrm{et}}, L', L''\}$. By the Lemma B.8 we can conclude

$$\tau^{\mathrm{ee}} \leq O\left(K\beta_{\max\{\kappa^{\mathrm{ee}}, L'\}} \log \frac{4K}{\delta_{\max\{\kappa^{\mathrm{ee}}, L'\}}}\right) +$$

$$O\left(\frac{\log\frac{K}{\delta_{\max\{\kappa^{\mathrm{ee}},\kappa^{\mathrm{et}},L',L''\}}} + \log\log\frac{1}{\Delta_{0,1}^2}}{\Delta_{0,1}^2} + \sum_{a=2}^{K}\frac{\log\frac{K}{\delta_{\max\{\kappa^{\mathrm{ee}},\kappa^{\mathrm{et}},L',L''\}}} + \log\log\frac{1}{\max\{\Delta_{1,a}^2,\Delta_{0,a}^2\}}}{\max\{\Delta_{1,a}^2,\Delta_{0,a}^2\}}\right)$$

$$\leq O\left(H_1^{\mathrm{pos}}\log H_1^{\mathrm{pos}}\right) + O\left(K\beta_{\kappa^{\mathrm{ee}}}\log\frac{4K}{\delta_{\kappa^{\mathrm{ee}}}}\right) + O\left(K\beta_{L'}\log\frac{4K}{\delta_{L'}}\right) +$$

$$O\left(H_1^{\mathrm{pos}}\log\frac{1}{\delta_{\kappa^{\mathrm{ee}}}}\right) + O\left(H_1^{\mathrm{pos}}\log\frac{1}{\delta_{\kappa^{\mathrm{et}}}}\right) + O\left(H_1^{\mathrm{pos}}\log\frac{1}{\delta_{L'}}\right) + O\left(H_1^{\mathrm{pos}}\log\frac{1}{\delta_{L''}}\right)$$

$$\leq O\left(H_1^{\mathrm{pos}}\log\frac{K}{\Delta_{0,1}^2}\right) + O\left(H_1^{\mathrm{pos}}\log\frac{1}{\delta_{\kappa^{\mathrm{ee}}}}\right) + O\left(H_1^{\mathrm{pos}}\log\frac{1}{\delta_{\kappa^{\mathrm{et}}}}\right) + O\left(K\beta_{\kappa^{\mathrm{ee}}}\log\frac{4K}{\delta_{\kappa^{\mathrm{ee}}}}\right).$$

Take expectation on both side, we have

$$\mathbb{E}\beta_{\kappa^{\mathrm{ee}}}\log\frac{4K}{\delta_{\kappa^{\mathrm{ee}}}} \leq \sum_{k=1}^{+\infty}\beta_k\log\frac{4K}{\delta_k}\Pr(\kappa^{\mathrm{ee}}=k)$$

$$\leq \sum_{k=1}^{+\infty}\frac{\pi^2\delta_{k-1}}{6}\cdot\beta_k\log\frac{4K}{\delta_k}$$

$$= O(\log K).$$

$$\mathbb{E}\log\frac{1}{\delta_{\kappa^{\mathrm{ee}}}} \leq \sum_{k=1}^{+\infty}\log\frac{1}{\delta_{\kappa^{\mathrm{ee}}}}\Pr(\kappa^{\mathrm{ee}}=k)$$

$$\leq \frac{\pi^2}{6}\sum_{k=1}^{+\infty}\delta_{k-1}\log\frac{1}{\delta_{\kappa^k}}$$

$$= O(1).$$

$$\mathbb{E}\log\frac{1}{\delta_{\kappa^{\mathrm{et}}}} \leq \sum_{k=1}^{+\infty}\log\frac{1}{\delta_{\kappa^{\mathrm{et}}}}\Pr(\kappa^{\mathrm{et}}=k)$$

$$\leq \frac{\pi^2\delta}{6}\sum_{k=1}^{+\infty}\frac{1}{\alpha_{k-1}}\log\frac{1}{\delta_{\kappa^k}}$$

$$= O(1).$$

Thus, we can conclude $\mathbb{E}\tau^{\mathrm{ee}} \leq O\left(H_1^{\mathrm{pos}}\log\frac{K}{\Delta_{0,1}^2}\right)$.

The remaining work is to prove an upper bound for $\mathbb{E}\tau^{\mathrm{et}}$. Recall we take $\beta_k = 2^k, \alpha_k = 5^k$. It is not hard to see for any integer $N \in \mathbb{N}$, we have $\beta_{N+1}\log\frac{4K\alpha_{N+1}}{\delta} \geq 2\beta_N\log\frac{4K\alpha_N}{\delta}$ and $\sum_{n=1}^{N}\beta_n\log\frac{4K\alpha_n}{\delta} \leq \beta_{N+1}\log\frac{4K\alpha_{N+1}}{\delta}$. Thus, we can conclude

$$\tau^{\mathrm{et}} = \sum_{k=1}^{\{\kappa^{\mathrm{ee}},\kappa^{\mathrm{et}},L''\}} 1000\beta_k\log\frac{4K\alpha_k}{\delta}$$

$$\leq 2000\beta_{\max\{\kappa^{\mathrm{ee}},\kappa^{\mathrm{et}},L''\}+1}\log\frac{4K\alpha_{\max\{\kappa^{\mathrm{ee}},\kappa^{\mathrm{et}},L''\}+1}}{\delta}$$

$$\leq O\left(\beta_{\kappa^{\mathrm{ee}}}\log\frac{4K\alpha_{\kappa^{\mathrm{ee}}}}{\delta} + \beta_{\kappa^{\mathrm{et}}}\log\frac{4K\alpha_{\kappa^{\mathrm{et}}}}{\delta} + \beta_{L''}\log\frac{4K\alpha_{L''}}{\delta}\right)$$

holds with certainty. From the definition, we know $\beta_{L''}, \alpha_{L''} \leq O\left(\frac{1}{\Delta_{0,1}^2}\right)$. Thus, we can conclude

$$\tau^{\mathrm{et}} \leq O\left(\beta_{\kappa^{\mathrm{ee}}}\log\frac{4K\alpha_{\kappa^{\mathrm{ee}}}}{\delta} + \beta_{\kappa^{\mathrm{et}}}\log\frac{4K\alpha_{\kappa^{\mathrm{et}}}}{\delta}\right) + O\left(\frac{\log\frac{K}{\delta} + \log\frac{1}{\Delta_{0,1}^2}}{\Delta_{0,1}^2}\right).$$

Take Expectation on both side, we get

$$\mathbb{E}\beta_{\kappa^{\text{ee}}} \log \frac{4K\alpha_{\kappa^{\text{ee}}}}{\delta} \leq \sum_{k=1}^{+\infty} \beta_k \log \frac{4K\alpha_k}{\delta} \Pr(\kappa^{\text{ee}} = k)$$

$$\leq \sum_{k=1}^{+\infty} \frac{\pi^2 \delta_{k-1}}{6} \cdot \beta_k \log \frac{4K\alpha_k}{\delta}$$

$$\leq O(\log \frac{K}{\delta}).$$

$$\mathbb{E}\beta_{\kappa^{\text{et}}} \log \frac{4K\alpha_{\kappa^{\text{et}}}}{\delta} \leq \sum_{k=1}^{+\infty} \beta_k \log \frac{4K\alpha_k}{\delta} \Pr(\kappa^{\text{et}} = k)$$

$$\leq \sum_{k=1}^{+\infty} \frac{\pi^2 \delta}{6\alpha_{k-1}} \cdot \beta_k \log \frac{4K\alpha_k}{\delta}$$

$$\leq O(\log K).$$

In summary, we have $\mathbb{E}\tau^{\text{et}} \leq O\left(\frac{\log \frac{K}{\delta} + \log \frac{1}{\Delta_{0,1}^2}}{\Delta_{0,1}^2}\right)$.

Combining the following upper bounds,

$$\mathbb{E}\tau^{\text{et}} \leq O\left(\frac{\log \frac{K}{\delta} + \log \frac{1}{\Delta_{0,1}^2}}{\Delta_{0,1}^2}\right),$$

$$\mathbb{E}\tau^{\text{ee}} \leq O\left(H_1^{\text{pos}} \log \frac{K}{\Delta_{0,1}^2}\right),$$

we have proved the positive part of theorem. $\qquad\square$

The above completes the proof of inequality $\mathbb{E}\tau \leq O\left(\frac{\log \frac{1}{\delta}}{\Delta_{0,1}^2} + (\log \frac{K}{\Delta_{0,1}^2}) \cdot H_1^{\text{pos}}\right)$, for the case $\nu \in \mathcal{S}^{\text{pos}}$. The following is going to prove another inequality $\mathbb{E}\tau \leq O(H_1^{\text{neg}}(\log \frac{1}{\delta} + \log H_1^{\text{neg}})), \nu \in \mathcal{S}^{\text{neg}}$ in Theorem 5.3.

*Proof of Theorem 5.3, negative case.* Define $\tilde{L}' = \lceil \log_2 \frac{\sum_{a=1}^K \frac{24}{\Delta_{0,a}^2}}{K} \rceil, \tilde{L}'' = \min\{k : \delta_k < \frac{\delta}{3}\} = \lceil \log_3 \frac{3}{\delta} \rceil$. We split $\tau = \tau^{\text{ee}} + \tau^{\text{et}}$ and derive an upper bound for $\mathbb{E}\tau^{\text{ee}}$ and $\mathbb{E}\tau^{\text{et}}$ separately.

By the Lemma B.9 and B.5, we know

$$\tau^{\text{ee}} \leq O\left(K\beta_{\max\{\tilde{L}', \kappa^{\text{ee}}\}} \log \frac{4K}{\delta_{\max\{\tilde{L}', \kappa^{\text{ee}}\}}}\right) + O\left(\sum_{a=1}^K \frac{\log \frac{K}{\delta_{\max\{\tilde{L}', \tilde{L}'', \kappa^{\text{ee}}\}}} + \log\log \frac{1}{\Delta_{0,a}^2}}{\Delta_{0,a}^2}\right)$$

$$\leq O\left(K\beta_{\tilde{L}'} \log \frac{4K}{\delta_{\tilde{L}'}}\right) + O\left(K\beta_{\kappa^{\text{ee}}} \log \frac{4K}{\delta_{\kappa^{\text{ee}}}}\right) + O(H_1^{\text{neg}} \log H_1^{\text{neg}}) +$$

$$O\left(H_1^{\text{neg}}(\log \frac{1}{\delta_{\tilde{L}'}} + \log \frac{1}{\delta_{\tilde{L}''}} + \log \frac{1}{\delta_{\kappa^{\text{ee}}}})\right).$$

Similar to the proof of positive part, we can take expectation on both side and it is not hard to see

$$O\left(K\beta_{\tilde{L}'} \log \frac{4K}{\delta_{\tilde{L}'}}\right) = O\left(H_1^{\text{neg}} \log H_1^{\text{neg}}\right)$$

$$\mathbb{E}K\beta_{\kappa^{\text{ee}}} \log \frac{4K}{\delta_{\kappa^{\text{ee}}}} \leq \sum_{k=1}^{+\infty} \frac{\pi^2 \delta_{k-1}}{6} K\beta_k \log \frac{4K}{\delta_k} \leq O(K)$$

$$O(H_1^{\text{neg}} \log \frac{1}{\delta_{\tilde{L}'}}) \leq O\left(H_1^{\text{neg}} \log H_1^{\text{neg}}\right)$$

$$O(H_1^{\text{neg}} \log \frac{1}{\delta_{\tilde{L}''}}) \leq O\left(H_1^{\text{neg}} \log \frac{1}{\delta}\right)$$

$$\mathbb{E} H_1^{\text{neg}} \log \frac{1}{\delta_{\kappa^{\text{ee}}}} \leq \sum_{k=1}^{+\infty} \frac{\pi^2 \delta_{k-1}}{6} H_1^{\text{neg}} \log \frac{1}{\delta_k} \leq O(H_1^{\text{neg}}).$$

In summary, we can conclude $\mathbb{E}\tau^{\text{ee}} \leq O\left(H_1^{\text{neg}}(\log \frac{1}{\delta} + H_1^{\text{neg}})\right)$.

For the upper bound of $\mathbb{E}\tau^{\text{et}}$, the idea is similar. From the Lemma B.9 and B.5, we know the exploration oracle would alway output $\hat{a}_k = \mathsf{None}$ for phase index $k \geq \max\{\kappa^{\text{ee}}, \tilde{L}'\}$, and we can conclude

$$\tau^{\text{et}} \leq \sum_{k=1}^{\max\{\kappa^{\text{ee}}, \tilde{L}'\}} 1000\beta_k \log \frac{4K\alpha_k}{\delta}$$

$$\leq 2000\beta_{\max\{\kappa^{\text{ee}}, \tilde{L}'\}+1} \log \frac{4K\alpha_{\max\{\kappa^{\text{ee}}, \tilde{L}'\}+1}}{\delta}$$

$$\leq O\left(\beta_{\kappa^{\text{ee}}+1} \log \frac{4K\alpha_{\kappa^{\text{ee}}+1}}{\delta}\right) + O\left(\beta_{\tilde{L}'+1} \log \frac{4K\alpha_{\tilde{L}'+1}}{\delta}\right).$$

Apply the same calculation as above, we can assert $\mathbb{E}\tau^{\text{et}} \leq O\left(\log \frac{K}{\delta} + \frac{H_1^{\text{neg}} \log H_1^{\text{neg}}}{K}\right)$.

Combining the upper bounds, we get $\mathbb{E}\tau \leq O\left(H_1^{\text{neg}}(\log \frac{1}{\delta} + H_1^{\text{neg}})\right)$.  $\square$

### B.3. Supplement Lemmas

In this section, we illustrate more on the supplement lemmas of the main Theorems 5.2 and 5.3. The first lemma is about the upper bound the smallest round index $k$, such that the Algorithm SEE does not receive $\hat{a}_k = \mathsf{Not\ Complete}$, conditioned on the concentration inequality.

**Lemma B.1.** *Denote*

$$\tilde{T}_k^{pos} := \frac{113(\log \frac{2K\alpha_k}{\delta} + \log\log \frac{96}{\omega^2 \Delta_{0,1}^2})}{\omega^2 \Delta_{0,1}^2}$$

$$\bar{T}_k^{pos} := \frac{113(C+1)^2(\log \frac{2K}{\delta_k} + \log\log \frac{96(C+1)^2}{\Delta_{0,1}^2})}{\Delta_{0,1}^2} + \sum_{a=2}^{K} \frac{113(C+1)^2(\log \frac{K}{\delta} + \log\log \frac{96(C+1)^2}{\max\{\Delta_{1,a}^2, \Delta_{0,a}^2\}})}{\max\{\Delta_{1,a}^2, \Delta_{0,a}^2\}}$$

$$T_k^{neg} := \sum_{a=1}^{K} \frac{113(\log \frac{2K}{\delta_k} + \log\log \frac{96}{\Delta_{0,a}^2})}{\Delta_{0,a}^2},$$

*where $\omega \in (0,1)$. Define*

$$L_{ee}^{pos} = \min\left\{k : \frac{T_k^{ee}}{2} \geq \bar{T}_k^{pos}\right\},$$

$$L_{et}^{pos} = \min\left\{k : T_k^{et} \geq \tilde{T}_k^{pos}\right\},$$

$$L^{neg} = \min\left\{k : \frac{T_k^{ee}}{2} \geq T_k^{neg}\right\}.$$

*We have $L_{ee}^{pos} \leq \lceil \log_2 \frac{24(C+1)^2 H_1^{pos}}{K} \rceil$, $L_{et}^{pos} \leq \lceil \log_2 \frac{192(C+1)^2}{\omega^2 \Delta_{0,1}^2} \rceil$, $L^{neg} \leq \lceil \log_2 \frac{\sum_{a=1}^{K} \frac{24}{\Delta_{0,a}^2}}{K} \rceil$*

Before the proof, we want to highlight a reminder:

$$H_1^{\text{pos}} = \frac{2}{\Delta_{0,1}^2} + \sum_{a=2}^{K} \frac{2}{\max\{\Delta_{0,a}^2, \Delta_{1,a}^2\}} \overset{\max\{\Delta_{1,a}^2, \Delta_{0,a}^2\} \geq \frac{\Delta_{0,1}^2}{4}}{\leq} \frac{2}{\Delta_{0,1}^2} + \frac{8(K-1)}{\Delta_{0,1}^2} \leq \frac{8K}{\Delta_{0,1}^2}.$$

Thus, we have $\lceil \log_2 \frac{24(C+1)^2 H_1^{\text{pos}}}{K} \rceil \leq \lceil \log_2 \frac{192(C+1)^2}{\omega^2 \Delta_{0,1}^2} \rceil$. In the following appendix, we will continue to use the notations in Lemma B.1.

*Proof of Lemma B.1.* By the following calculation, we have

$$1000\beta_k \log \frac{4K\alpha_k}{\delta} \geq \frac{113(\log \frac{2K}{\delta} + \log\log \frac{96}{\omega^2 \Delta_{0,1}^2})}{\omega^2 \Delta_{0,1}^2}$$

$$\Leftarrow \beta_k \log \frac{4K\alpha_k}{\delta} \geq \frac{\log \frac{2K}{\delta} + \log\log \frac{96}{\omega^2 \Delta_{0,1}^2}}{\omega^2 \Delta_{0,1}^2}$$

$$\Leftarrow \beta_k \geq \frac{2}{\omega^2 \Delta_{0,1}^2}, \log \frac{4K\alpha_k}{\delta} \geq \log\log \frac{96}{\omega^2 \Delta_{0,1}^2}$$

$$\Leftarrow 2^k \geq \frac{2}{\omega^2 \Delta_{0,1}^2}, 5^k \geq \log \frac{96}{\omega^2 \Delta_{0,1}^2}$$

$$\Leftarrow k \geq \log_2 \frac{96}{\omega^2 \Delta_{0,1}^2}$$

$$\overset{C \geq 1}{\Leftarrow} k \geq \log_2 \frac{192(C+1)^2}{\omega^2 \Delta_{0,1}^2},$$

Thus, we can conclude $L_{\text{et}}^{\text{pos}} \leq \lceil \log_2 \frac{192(C+1)^2}{\omega^2 \Delta_{0,1}^2} \rceil$.

$$500(C+1)^2 K\beta_k \log \frac{4K}{\delta_k} \geq \frac{113(C+1)^2(\log \frac{2K}{\delta_k} + \log\log \frac{96(C+1)^2}{\Delta_{0,1}^2})}{\Delta_{0,1}^2} + \sum_{a=2}^{K} \frac{113(C+1)^2(\log \frac{2K}{\delta_k} + \log\log \frac{96(C+1)^2}{\max\{\Delta_{1,a}^2, \Delta_{0,a}^2\}})}{\max\{\Delta_{1,a}^2, \Delta_{0,a}^2\}}$$

$$\Leftarrow K\beta_k \log \frac{4K}{\delta_k} \geq \frac{(\log \frac{2K}{\delta_k} + \log\log \frac{96(C+1)^2}{\Delta_{0,1}^2})}{\Delta_{0,1}^2} + \sum_{a=2}^{K} \frac{(\log \frac{2K}{\delta_k} + \log\log \frac{96(C+1)^2}{\max\{\Delta_{1,a}^2, \Delta_{0,a}^2\}})}{\max\{\Delta_{1,a}^2, \Delta_{0,a}^2\}}$$

$$\Leftarrow K\beta_k \log \frac{4K}{\delta_k} \geq (\log \frac{2K}{\delta_k} + \log\log(96(C+1)^2 H_1^{\text{pos}}))H_1^{\text{pos}}$$

$$\Leftarrow \beta_k \geq \frac{2H_1^{\text{pos}}}{K}, \frac{\log\log(96(C+1)^2 H_1^{\text{pos}})}{\log \frac{4K}{\delta_k}} \leq 1$$

$$\Leftarrow \beta_k \geq \frac{2H_1^{\text{pos}}}{K}, \frac{1}{\delta_k} \geq \frac{\log(96(C+1)^2 H_1^{\text{pos}})}{4K}$$

$$\Leftarrow 2^k \geq \frac{2H_1^{\text{pos}}}{K}, 3^k \geq \frac{24(C+1)^2 H_1^{\text{pos}}}{K}$$

$$\Leftarrow k \geq \lceil \log_2 \frac{24(C+1)^2 H_1^{\text{pos}}}{K} \rceil,$$

Thus, we can conclude $L_{\text{ee}}^{\text{pos}} \leq \lceil \log_2 \frac{24(C+1)^2 H_1^{\text{pos}}}{K} \rceil$.

Similarly, by the following calculation,

$$500(C+1)^2 K\beta_k \log \frac{4K}{\delta_k} \geq \sum_{a=1}^{K} \frac{113(\log \frac{2K}{\delta_k} + \log\log \frac{96}{\Delta_{0,a}^2})}{\Delta_{0,a}^2}$$

$$\Leftarrow K\beta_k \geq \frac{\log \frac{2K}{\delta_k}}{\log \frac{4K}{\delta_k}} \sum_{a=1}^{K} \frac{1}{\Delta_{0,a}^2} + \frac{1}{\log \frac{4K}{\delta_k}} \sum_{a=1}^{K} \frac{\log\log \frac{96}{\Delta_{0,a}^2}}{\Delta_{0,a}^2}$$

$$\Leftarrow \beta_k \geq \frac{\sum_{a=1}^{K} \frac{2}{\Delta_{0,a}^2}}{K}, K \log \frac{4K}{\delta_k} \geq \log\log \frac{96}{\Delta_{0,a}^2}, \forall a \in [K]$$

$$\Leftarrow 2^k \geq \frac{\sum_{a=1}^{K} \frac{2}{\Delta_{0,a}^2}}{K}, \log \frac{4K}{\delta_k} \geq \log \log \frac{96}{\Delta_{0,a}^2}, \forall a \in [K]$$

$$\Leftarrow 2^k \geq \frac{\sum_{a=1}^{K} \frac{2}{\Delta_{0,a}^2}}{K}, \frac{4K}{\delta_k} \geq \frac{96}{\Delta_{0,a}^2}, \forall a \in [K]$$

$$\Leftarrow 2^k \geq \frac{\sum_{a=1}^{K} \frac{2}{\Delta_{0,a}^2}}{K}, K \cdot 3^k \geq \sum_{a=1}^{K} \frac{24}{\Delta_{0,a}^2}$$

$$\Leftarrow k \geq \log_2 \frac{\sum_{a=1}^{K} \frac{24}{\Delta_{0,a}^2}}{K}$$

Thus, we can conclude $L^{\text{neg}} \leq \lceil \log_2 \frac{\sum_{a=1}^{K} \frac{24}{\Delta_{0,a}^2}}{K} \rceil$. $\qquad\qquad\square$

**Lemma B.2** ($\tau$ must be finite). *Apply Algorithm 2 to a 1-Identification instance $\nu$, we have $\Pr(\tau < +\infty) = 1$.*

*Proof pf Lemma B.2.* Here we use the same notaion in Lemma B.1. Since $\beta_k = 2^k$, not hard to see $T_k^{\text{ee}} \geq 2T_{k-1}^{\text{ee}}$. Thus when $k \geq L_{\text{ee}}^{\text{pos}}$ and the algorithm call exploration algorithm 3, we have

$$T_k^{\text{ee}} = \frac{T_k^{\text{ee}}}{2} + \frac{T_k^{\text{ee}}}{2}$$

$$\geq T_{k-1}^{\text{ee}} + \frac{T_k^{\text{ee}}}{2}$$

$$\geq T_{k-1}^{\text{ee}} + \bar{T}_k^{\text{pos}}$$

Since at the start of phase $k$, the $|\mathcal{H}^{\text{ee}}| + |Q| \leq T_{k-1}^{\text{ee}}$, we can assert $k \geq L_{\text{ee}}^{\text{pos}}$ can fulfill the condition of $T \geq \sum_{a=1}^{K} N_a^0 + \frac{113(C+1)^2(\log \frac{2K}{\delta} + \log\log \frac{96(C+1)^2}{\Delta_{0,1}^2})}{\Delta_{0,1}^2} + \sum_{a=2}^{K} \frac{113(C+1)^2(\log \frac{2K}{\delta} + \log\log \frac{96(C+1)^2}{\max\{\Delta_{1,a}^2, \Delta_{0,a}^2\}})}{\max\{\Delta_{1,a}^2, \Delta_{0,a}^2\}}$ in Lemma B.4. Similarly, $k \geq L_{\text{et}}^{\text{pos}}$ can also fulfill the condition $T \geq \frac{113(\log \frac{2K\alpha}{\delta} + \log\log \frac{96}{\omega^2 \Delta_{0,1}^2})}{\omega^2 \Delta_{0,1}^2}$ in Lemma B.3. And $k \geq L^{\text{neg}}$ can fulfill the condition $T \geq \sum_{a=1}^{K} N_a^0 + \sum_{a=1}^{K} \frac{113(C+1)^2(\log \frac{K}{\delta} + \log\log \frac{96(C+1)^2}{\Delta_{0,a}^2})}{\Delta_{0,a}^2}$ in Lemma B.5.

Recall the definition

$$\kappa^{\text{ee}} = \min \left\{ k \in \mathbb{N} : \forall a \in [K], \forall t \in \mathbb{N} : \left| \frac{\sum_{s=1}^{t} X_{a,s}^{\text{ee}}}{t} - \mu_a \right| \leq \frac{\sqrt{2 \cdot 2^{\lceil \log_2 t \rceil^+} \log \frac{2K(\lceil \log_2 t \rceil^+)^2}{\delta_k}}}{t} \right\}$$

$$\kappa^{\text{et}} = \min \left\{ k \in \mathbb{N} : \forall a \in [K], \forall t \in \mathbb{N} : \left| \frac{\sum_{s=1}^{t} X_{a,s}^{\text{et}}}{t} - \mu_a \right| \leq \frac{\sqrt{2 \cdot 2^{\lceil \log_2 t \rceil^+} \log \frac{2K\alpha_k(\lceil \log_2 t \rceil^+)^2}{\delta}}}{t} \right\}$$

- For positive case, by the Lemma B.3, B.8, B.1, we know the algorithm must terminate at the phase $\max\{\kappa^{\text{ee}}, \kappa^{\text{et}}, L_{\text{et}}^{\text{pos}}\}$. Since the length of phase $k$ is bounded by $O(K\beta_k \log \frac{4\alpha_k}{\delta_k \delta})$ with certainty, we know $\Pr(\tau < +\infty) = \Pr(\kappa^{\text{ee}} = +\infty \text{ or } \kappa^{\text{et}} = +\infty) = 0$.

- For negative case, denote $L' = \min\{k : \delta_k < \frac{\delta}{3}\}$. By the Lemma B.9, B.1, we know the algorithm must terminate at the phase $\max\{L', \kappa^{\text{ee}}, L^{\text{neg}}\}$. Since the length of phase $k$ is bounded by $O(K\beta_k \log \frac{4\alpha_k}{\delta_k \delta})$ with certainty, we know $\Pr(\tau < +\infty) = \Pr(\kappa^{\text{ee}} = +\infty) = 0$.

$\qquad\qquad\square$

The following lemma states the conditions that can guarantee the output of exploitation period(Algorithm 4).

**Lemma B.3** (Correctness of Exploitation Period). *Conditioned on the event*

$$E^{et} = \left\{ \forall a \in [K], \forall t \in \mathbb{N}, \left| \frac{\sum_{s=1}^t X_{a,s}^{et}}{t} - \mu_a \right| < U(t, \frac{\delta}{K\alpha}) \right\},$$

*and* $\hat{a} \in [K], \mu_{\hat{a}} > \omega\mu_1 + (1-\omega)\mu_0$. *If* $T \geq \frac{113(\log \frac{2K\alpha}{\delta} + \log\log \frac{96}{\omega^2 \Delta_{0,1}^2})}{\omega^2 \Delta_{0,1}^2}$, *Algorithm 4 would output Qualified.*

*Proof of Lemma B.3.* For arm $\hat{a} \in [K], \mu_{\hat{a}} > \omega\mu_1 + (1-\omega)\mu_0$ and $t \in \mathbb{N}$, we have

$$\frac{\sum_{s=1}^t X_{\hat{a},s}^{et}}{t} - U(t, \frac{\delta}{K\alpha}) > \mu_0$$

$$\overset{E^{et}}{\Longleftarrow} \mu_{\hat{a}} - 2U(t, \frac{\delta}{K\alpha}) > \mu_0$$

$$\overset{\mu_{\hat{a}} > \omega\mu_1 + (1-\omega)\mu_0}{\Longleftarrow} 2U(t, \frac{\delta}{K\alpha}) < \omega(\mu_1 - \mu_0)$$

$$\Longleftarrow 2\sqrt{\frac{4\log\frac{2K\alpha\log(2t)}{\delta}}{t}} \leq \omega\Delta_{0,1}$$

$$\overset{\text{Lemma } D.2}{\Longleftarrow} t \geq \frac{112(\log\frac{2K\alpha}{\delta} + \log\log\frac{96}{\omega^2\Delta_{0,1}^2})}{\omega^2\Delta_{0,1}^2},$$

which implies the algorithm would return Qualified before the $N_{\hat{a}}^{et}$ is no less than $T$. $\qquad\square$

After the discussion on the conditions of the Algorithm 4), the following two lemmas turn to "good conditions" of the Algorithm 3.

**Lemma B.4** (Property of UCB Rule, for Positive Case). *Consider pulling process controlled by Algorithm 3. Assume the input $\mathcal{H}^{ee}$ satisfies* $\text{LCB}_a < \mu_0$ *holds for all $a \in [K]$ at line 2. Apply the pulling process to a positive instance and further assume* $T > \sum_{a=1}^K N_a^0 + \frac{113(C+1)^2(\log\frac{2K}{\delta} + \log\log\frac{96(C+1)^2}{\Delta_{0,1}^2})}{\Delta_{0,1}^2} + \sum_{a=2}^K \frac{113(C+1)^2(\log\frac{2K}{\delta} + \log\log\frac{96(C+1)^2}{\max\{\Delta_{1,a}^2, \Delta_{0,a}^2\}})}{\max\{\Delta_{1,a}^2, \Delta_{0,a}^2\}}$. *We have*

*Conditioned on the event*

$$E^{ee} = \left\{ \forall a \in [K], \forall t \in \mathbb{N}, \left| \frac{\sum_{s=1}^t X_{a,s}^{ee}}{t} - \mu_a \right| < U\left(t, \frac{\delta}{K}\right) \right\},$$

*at the end of the pulling procedure (line 17), we have*

1. $N_a(\mathcal{H}^{ee}) \leq \max\left\{ N_a^0, \frac{113(C+1)^2(\log\frac{2K}{\delta} + \log\log\frac{96(C+1)^2}{\max\{\Delta_{0,a}^2, \Delta_{1,a}^2\}})}{\max\{\Delta_{0,a}^2, \Delta_{1,a}^2\}} \right\}$, *for* $a \in [K]$.

2. $\hat{a} \in [K], \mu_{\hat{a}} > \omega\mu_1 + (1-\omega)\mu_0, \omega = \frac{C-1}{C+3}$.

*Proof of Lemma B.4.* For simplicity, denote $\mathcal{T}_a' = \frac{113(C+1)^2(\log\frac{2K}{\delta} + \log\log\frac{96(C+1)^2}{\Delta_{1,a}^2})}{\Delta_{1,a}^2}$, $\mathcal{T}_a'' = \frac{113(C+1)^2(\log\frac{2K}{\delta} + \log\log\frac{96(C+1)^2}{\Delta_{0,a}^2})}{\Delta_{0,a}^2}$. Not hard to see $\min\{\mathcal{T}_a', \mathcal{T}_a''\} = \frac{113(C+1)^2(\log\frac{2K}{\delta} + \log\log\frac{96(C+1)^2}{\max\{\Delta_{0,a}^2, \Delta_{1,a}^2\}})}{\max\{\Delta_{0,a}^2, \Delta_{1,a}^2\}}$.

Consider $a \geq 2$. For pulling times $t \in \mathbb{N}$, through the following calculation,

$$\frac{\sum_{s=1}^t X_{a,s}^{ee}}{t} + U(t, \frac{\delta}{K}) < \mu_1$$

$$\overset{E^{ee}}{\Longleftarrow} \mu_a + 2 \cdot U(t, \frac{\delta}{K}) < \mu_1$$

$$\Leftarrow 2\sqrt{\frac{4\log\frac{2K\log(2t)}{\delta}}{t}} \le \Delta_{1,a}$$

$$\stackrel{\text{Lemma } D.2}{\Longleftarrow} t > \frac{112\log\frac{2K}{\delta}}{\Delta_{1,a}^2} + \frac{64\log\left(\log\left(\frac{96}{\Delta_{1,a}^2}\right)\right)}{\Delta_{1,a}^2},$$

we know $\text{UCB}_a(t) < \mu_1 < \text{UCB}_1(t) \Leftarrow N_a(t) \ge \frac{112\log\frac{2K}{\delta}}{\Delta_{1,a}^2} + \frac{64\log\left(\log\left(\frac{96}{\Delta_{1,a}^2}\right)\right)}{\Delta_{1,a}^2} + 1.$

Similarly, consider $a \in [K]$ such that $\mu_0 < \mu_a \le \mu_1$. through the following calculation,

$$\frac{\sum_{s=1}^t X_{a,s}^{\text{ee}}}{t} - C \cdot U(t, \frac{\delta}{K}) \ge \mu_0$$

$$\stackrel{E^{\text{ee}}}{\Longleftarrow} \mu_a - (C+1) \cdot U(t, \frac{\delta}{K}) \ge \mu_0$$

$$\Leftarrow (C+1)\sqrt{\frac{4\log\frac{2K\log(2t)}{\delta}}{t}} \le \Delta_{0,a}$$

$$\stackrel{\text{Lemma } D.2}{\Longleftarrow} t > \frac{28(C+1)^2\log\frac{2K}{\delta}}{\Delta_{0,a}^2} + \frac{16(C+1)^2\log\left(\log\left(\frac{24(C+1)^2}{\Delta_{0,a}^2}\right)\right)}{\Delta_{0,a}^2}$$

we know that $\text{LCB}_a(t) > \mu_0 \Leftarrow N_a(t) > \frac{28(C+1)^2\log\frac{2K}{\delta}}{\Delta_{0,a}^2} + \frac{16(C+1)^2\log\left(\log\left(\frac{24(C+1)^2}{\Delta_{0,a}^2}\right)\right)}{\Delta_{0,a}^2} + 1.$ Once $\text{LCB}_a \ge \mu_0$ happens, the algorithm stops and take arm $a$ as the output $\hat{a}$.

We are ready to prove the first claim. According to the above discussion and the condition $\text{LCB}_a < \mu_0$, we know $N_a^0 < \mathcal{T}_a''$ holds for all $a \in [K]$. We prove the first claim through the discussion on three cases.

1. If $N_a^0 \le \mathcal{T}_a' \le \mathcal{T}_a''$, the algorithm assures $N_a(\mathcal{H}^{\text{ee}}) \le \mathcal{T}_a' = \max\left\{N_a^0, \frac{113(C+1)^2(\log\frac{2K}{\delta} + \log\log\frac{96(C+1)^2}{\max\{\Delta_{0,a}^2, \Delta_{1,a}^2\}})}{\max\{\Delta_{0,a}^2, \Delta_{1,a}^2\}}\right\}$, as

   $\text{UCB}_a < \mu_1 < \text{UCB}_1 \Leftarrow N_a(\mathcal{H}^{\text{ee}}) \ge \frac{112\log\frac{2K}{\delta}}{\Delta_{1,a}^2} + \frac{64\log\left(\log\left(\frac{96}{\Delta_{1,a}^2}\right)\right)}{\Delta_{1,a}^2} + 1$ and arm $a$ will never be the arm with highest upper confidence bound before $N_a(\mathcal{H}^{\text{ee}})$ is no less than $\mathcal{T}_a'$.

2. If $\mathcal{T}_a' < N_a^0 \le \mathcal{T}_a''$, the algorithm never pulls arm $a$ as its upper bound must below arm 1. In this case, $N_a(\mathcal{H}^{\text{ee}}) \le N_a^0 = \max\left\{N_a^0, \frac{113(C+1)^2(\log\frac{2K}{\delta} + \log\log\frac{96(C+1)^2}{\max\{\Delta_{0,a}^2, \Delta_{1,a}^2\}})}{\max\{\Delta_{0,a}^2, \Delta_{1,a}^2\}}\right\}$.

3. If If $N_a^0 \le \mathcal{T}_a'' \le \mathcal{T}_a'$, the algorithm assures $N_a(\mathcal{H}^{\text{ee}}) \le \mathcal{T}_a'' = \max\left\{N_a^0, \frac{113(C+1)^2(\log\frac{2K}{\delta} + \log\log\frac{96(C+1)^2}{\max\{\Delta_{0,a}^2, \Delta_{1,a}^2\}})}{\max\{\Delta_{0,a}^2, \Delta_{1,a}^2\}}\right\}$,

   as the algorithm will output arm $a$ before its pulling time is no less than $\mathcal{T}_a''$.

By the first claim, we know

$$\sum_{a=1}^K N_a(\mathcal{H}^{\text{ee}}) \le \sum_{a=1}^K \max\left\{N_a^0, \frac{113(C+1)^2(\log\frac{2K}{\delta} + \log\log\frac{96(C+1)^2}{\max\{\Delta_{0,a}^2, \Delta_{1,a}^2\}})}{\max\{\Delta_{0,a}^2, \Delta_{1,a}^2\}}\right\}$$

$$\le \sum_{a=1}^K N_a^0 + \sum_{a=1}^K \frac{113(C+1)^2(\log\frac{2K}{\delta} + \log\log\frac{96(C+1)^2}{\max\{\Delta_{0,a}^2, \Delta_{1,a}^2\}})}{\max\{\Delta_{0,a}^2, \Delta_{1,a}^2\}}$$

$$\le T.$$

Thus, the algorithm would not output Not Complete. From the good event $E^{\text{ee}}$, we know the $\hat{a} \neq$ None, since $\text{UCB}_1$ is always above $\mu_0$. We can conclude $\hat{a} \in [K]$. We just need to prove $\mu_{\hat{a}} > \omega\mu_1 + (1-\omega)\mu_0$.

Prove by contradiction. If $\hat{a} \in [K]$ and $\mu_{\hat{a}} < \omega\mu_1 + (1-\omega)\mu_0$, we have

$$A_{\tau^{\text{ee}}} = \hat{a}, \hat{a} = \arg\max_{1 \leq i \leq K} \hat{\mu}_{i,N_i(\tau^{\text{ee}}-1)} + U(N_i(\tau^{\text{ee}}-1), \frac{\delta}{K})$$

$$\hat{\mu}_{\hat{a},N_{\hat{a}}(\tau^{\text{ee}})} - C \cdot U(N_{\hat{a}}(\tau^{\text{ee}}), \frac{\delta}{K}) > \mu_0$$

$$\hat{\mu}_{\hat{a},N_{\hat{a}}(\tau^{\text{ee}}-1)} - C \cdot U(N_{\hat{a}}(\tau^{\text{ee}}-1)) = \hat{\mu}_{\hat{a},N_{\hat{a}}(\tau^{\text{ee}})-1} - C \cdot U(N_{\hat{a}}(\tau^{\text{ee}})-1, \frac{\delta}{K}) < \mu_0.$$

As we take $U(t, \frac{\delta}{K}) = \frac{\sqrt{2 \cdot 2^{\max\{\lceil \log_2 t \rceil, 1\}} \log \frac{2K(\lceil \log_2 t \rceil)^2}{\delta}}}{t}$, we get

$$\hat{\mu}_{\hat{a},N_{\hat{a}}(\tau^{\text{ee}})} - CU(N_{\hat{a}}(\tau^{\text{ee}})) > \mu_0$$

$$\Leftrightarrow \hat{\mu}_{\hat{a},N_{\hat{a}}(\tau^{\text{ee}})} - C\frac{\sqrt{2 \cdot 2^{\max\{\lceil \log_2 N_{\hat{a}}(\tau^{\text{ee}}) \rceil, 1\}} \log \frac{2K(\lceil \log_2 N_{\hat{a}}(\tau^{\text{ee}}) \rceil)^2}{\delta}}}{N_{\hat{a}}(\tau)} > \mu_0$$

$$\overset{E^{\text{ee}}}{\Rightarrow} \mu_{\hat{a}} - (C-1)\frac{\sqrt{2 \cdot 2^{\max\{\lceil \log_2 N_{\hat{a}}(\tau^{\text{ee}}) \rceil, 1\}} \log \frac{2K(\lceil \log_2 N_{\hat{a}}(\tau^{\text{ee}}) \rceil)^2}{\delta}}}{N_{\hat{a}}(\tau^{\text{ee}})} > \mu_0$$

$$\Rightarrow \omega\mu_1 + (1-\omega)\mu_0 - (C-1)\frac{\sqrt{2 \cdot 2^{\max\{\lceil \log_2 N_{\hat{a}}(\tau^{\text{ee}}) \rceil, 1\}} \log \frac{2K(\lceil \log_2 N_{\hat{a}}(\tau^{\text{ee}}) \rceil)^2}{\delta}}}{N_{\hat{a}}(\tau^{\text{ee}})} > \mu_0$$

$$\Leftrightarrow \omega(\mu_1 - \mu_0) > (C-1)\frac{\sqrt{2 \cdot 2^{\max\{\lceil \log_2 N_{\hat{a}}(\tau^{\text{ee}}) \rceil, 1\}} \log \frac{2K(\lceil \log_2 N_{\hat{a}}(\tau^{\text{ee}}) \rceil)^2}{\delta}}}{N_{\hat{a}}(\tau^{\text{ee}})}$$

$$\Leftrightarrow \frac{2\omega(\mu_1 - \mu_0)}{C-1} > \frac{2\sqrt{2 \cdot 2^{\max\{\lceil \log_2 N_{\hat{a}}(\tau^{\text{ee}}) \rceil, 1\}} \log \frac{2K(\lceil \log_2 N_{\hat{a}}(\tau^{\text{ee}}) \rceil)^2}{\delta}}}{N_{\hat{a}}(\tau^{\text{ee}})}$$

On the other hands,

$$\hat{\mu}_{\hat{a},N_{\hat{a}}(\tau^{\text{ee}}-1)} + U(N_{\hat{a}}(\tau^{\text{ee}}-1)) \leq \mu_1$$

$$\overset{E^{\text{ee}}}{\Leftarrow} \mu_{\hat{a}} + 2U(N_{\hat{a}}(\tau^{\text{ee}}-1)) \leq \mu_1$$

$$\Leftarrow \omega\mu_1 + (1-\omega)\mu_0 + 2U(N_{\hat{a}}(\tau^{\text{ee}}-1)) \leq \mu_1$$

$$\Leftrightarrow \frac{2\sqrt{2 \cdot 2^{\max\{\lceil \log_2 N_{\hat{a}}(\tau^{\text{ee}})-1 \rceil, 1\}} \log \frac{2K(\lceil \log_2 N_{\hat{a}}(\tau^{\text{ee}})-1 \rceil)^2}{\delta}}}{N_{\hat{a}}(\tau^{\text{ee}})-1} \leq (1-\omega)(\mu_1 - \mu_0)$$

$$\Leftrightarrow \frac{\sqrt{2 \cdot 2^{\max\{\lceil \log_2 N_{\hat{a}}(\tau^{\text{ee}})-1 \rceil, 1\}} \log \frac{2K(\lceil \log_2 N_{\hat{a}}(\tau^{\text{ee}})-1 \rceil)^2}{\delta}}}{N_{\hat{a}}(\tau^{\text{ee}})-1} \leq \frac{(1-\omega)(\mu_1 - \mu_0)}{2}.$$

Notice that

$$\frac{\sqrt{2 \cdot 2^{\max\{\lceil \log_2 N_{\hat{a}}(\tau^{\text{ee}})-1 \rceil, 1\}} \log \frac{2K(\lceil \log_2 N_{\hat{a}}(\tau^{\text{ee}})-1 \rceil)^2}{\delta}}}{N_{\hat{a}}(\tau^{\text{ee}})-1}$$

$$\leq \frac{N_{\hat{a}}(\tau^{\text{ee}})}{N_{\hat{a}}(\tau^{\text{ee}})-1} \frac{\sqrt{2 \cdot 2^{\max\{\lceil \log_2 N_{\hat{a}}(\tau^{\text{ee}}) \rceil, 1\}} \log \frac{2K(\lceil \log_2 N_{\hat{a}}(\tau^{\text{ee}}) \rceil)^2}{\delta}}}{N_{\hat{a}}(\tau^{\text{ee}})}$$

$$\leq \frac{2\sqrt{2 \cdot 2^{\max\{\lceil \log_2 N_{\hat{a}}(\tau^{\text{ee}}) \rceil, 1\}} \log \frac{2K(\lceil \log_2 N_{\hat{a}}(\tau^{\text{ee}}) \rceil)^2}{\delta}}}{N_{\hat{a}}(\tau^{\text{ee}})}$$

and by $\omega = \frac{C-1}{C+3}$

$$\frac{(1-\omega)}{2} = \frac{1 - \frac{C-1}{C+3}}{2} = \frac{4}{2(C+3)} = \frac{2\omega}{C-1},$$

we can conclude

$$\hat{\mu}_{\hat{a}, N_{\hat{a}}(\tau)} - C \cdot U(N_{\hat{a}}(\tau^{\text{ee}}), \frac{\delta}{K}) > \mu_0$$

$$\Rightarrow \hat{\mu}_{\hat{a}, N_{\hat{a}}(\tau^{\text{ee}}-1)} + U(N_{\hat{a}}(\tau^{\text{ee}}-1), \frac{\delta}{K}) \leq \mu_1$$

$$\overset{E^{\text{ee}}}{\Rightarrow} \hat{\mu}_{\hat{a}, N_{\hat{a}}(\tau^{\text{ee}}-1)} + U(N_{\hat{a}}(\tau^{\text{ee}}-1)) < \hat{\mu}_{1, N_1(\tau^{\text{ee}}-1)} + U(N_1(\tau^{\text{ee}}-1))$$

$$\Rightarrow A_{\tau^{\text{ee}}} \neq \hat{a}.$$

We have found a contradiction. And we can conclude $\mu_{\hat{a}} \geq \frac{C-1}{C+3}\mu_1 + \frac{4}{C+3}\mu_0$, conditioned on the event $E^{\text{ee}}$.  $\square$

**Lemma B.5** (Property of UCB Rule, for Negative Case). *Consider pulling process controlled by Algorithm 3. Apply the pulling process to a negative instance and further assume $T > \sum_{a=1}^{K} N_a^0 + \sum_{a=1}^{K} \frac{113(C+1)^2 (\log \frac{K}{\delta} + \log \log \frac{1}{\Delta_{0,a}^2})}{\Delta_{0,a}^2}$.*

*Conditioned on the event*

$$E^{ee} = \left\{ \forall a \in [K], \forall t \in \mathbb{N}, \left| \frac{\sum_{s=1}^{t} X_{a,s}^{ee}}{t} - \mu_a \right| < U\left(t, \frac{\delta}{K}\right) \right\},$$

*at the end of the pulling procedure (line 17), we have*

1. $N_a(\mathcal{H}^{ee}) \leq \max \left\{ N_a^0, \frac{113(\log \frac{2K}{\delta} + \log \log \frac{96}{\Delta_{0,a}^2})}{\Delta_{0,a}^2} \right\}$, *for* $\forall a \in [K]$.

2. $\hat{a} \in \mathsf{None}$

*Proof of Lemma B.5.* For simplicity, denote $\mathcal{T}_a = \frac{113(\log \frac{2K}{\delta} + \log \log \frac{96}{\Delta_{0,a}^2})}{\Delta_{0,a}^2}$.

Consider $a \geq 1$. For pulling times $t \in \mathbb{N}$, through the following calculation,

$$\frac{\sum_{s=1}^{t} X_{a,s}^{\text{ee}}}{t} + U(t, \frac{\delta}{K}) < \mu_0$$

$$\overset{E^{\text{ee}}}{\Leftarrow} \mu_a + 2 \cdot U(t, \frac{\delta}{K}) < \mu_0$$

$$\Leftarrow 2\sqrt{\frac{4 \log \frac{2K \log(2t)}{\delta}}{t}} \leq \Delta_{0,a}$$

$$\overset{\text{Lemma } D.2}{\Leftarrow} t > \frac{112(\log \frac{2K}{\delta} + \log \log \frac{96}{\Delta_{0,a}^2})}{\Delta_{0,a}^2},$$

we know $\text{UCB}_a(t) < \mu_0 \Leftarrow N_a(t) \geq \frac{112(\log \frac{2K}{\delta} + \log \log \frac{96}{\Delta_{0,a}^2})}{\Delta_{0,a}^2} + 1$. If arm $a$ is still the arm with highest upper confidence bound while $\text{UCB}_a < \mu_0$, the algorithm would stop and take $\hat{a} = \mathsf{None}$. Thus, arm $a$ will never get pulled once $\text{UCB}_a < \mu_0$ holds.

We are ready to prove the first claim, by analyzing the following two cases. For $a \geq 1$,

1. If $N_a^0 \leq \mathcal{T}_a$, the algorithm assures $N_a(\mathcal{H}^{\text{ee}}) \leq \mathcal{T}_a = \max\{N_a^0, \mathcal{T}_a\}$, as the above discussion indicates.

2. If $N_a^0 > \mathcal{T}_a$, then the upper confidence bound of arm $a$ is smaller than the $\mu_0$ at the start of the algorithm. This arm will never get pull till the end of the algorithm. Thus, $N_a(\mathcal{H}^{ee}) = N_a^0 = \max\{N_a^0, \mathcal{T}_a\}$.

Then, we turn to the second claim. From the good event, we know $\text{LCB}_a < \mu_a < \mu_0$ holds for arm $a \in [K]$. That means the algorithm will not output $\hat{a} \in [K]$. In addition, from the first claim, we know

$$\sum_{a=1}^{K} N_a(\mathcal{H}^{ee}) \leq \sum_{a=1}^{K} \max \left\{ N_a^0, \frac{113(\log \frac{2K}{\delta} + \log\log \frac{96}{\Delta_{0,a}^2})}{\Delta_{0,a}^2} \right\}$$
$$\leq \sum_{a=1}^{K} N_a^0 + \sum_{a=1}^{K} \frac{113(\log \frac{2K}{\delta} + \log\log \frac{96}{\Delta_{0,a}^2})}{\Delta_{0,a}^2}$$
$$\leq T.$$

Then the algorithm must terminate before the round $T$. Since the upper confidence bounds of all arm $a \in [K]$ are below $\mu_0$, the algorithm will output $\hat{a} = \text{None}$. $\square$

The following lemma shows the condition of Lemma B.4 can be fulfilled in phase $k \geq \max\left\{\kappa^{ee}, \lceil \log_2 \frac{24(C+1)^2 H_1^{pos}}{K} \rceil\right\} =: L$.

**Lemma B.6.** *Apply Algorithm 2 to a positive 1-identification instance $\nu$, for phase index $k \geq \max\left\{\kappa^{ee}, \lceil \log_2 \frac{24(C+1)^2 H_1^{pos}}{K} \rceil\right\} =: L$, before running the exploration (at the Line 2 of Algorithm 3), we have*

$$T_k^{ee} \geq |\mathcal{H}^{ee}| + 113(C+1)^2 \left( \frac{(\log \frac{2K}{\delta_k} + \log\log \frac{96(C+1)^2}{\Delta_{0,1}^2})}{\Delta_{0,1}^2} + \sum_{a=2}^{K} \frac{(\log \frac{K}{\delta} + \log\log \frac{96(C+1)^2}{\max\{\Delta_{1,a}^2,\Delta_{0,a}^2\}})}{\max\{\Delta_{1,a}^2, \Delta_{0,a}^2\}} \right)$$

*and $\text{LCB}_a(\mathcal{H}^{ee}, \delta_k) \leq \mu_0$ holds for all $a \in [K]$.*

*Proof of Lemma B.6.* By the Lemma B.1, we know $k \geq \lceil \log_2 \frac{24(C+1)^2 H_1^{pos}}{K} \rceil$ implies

$$\frac{T_k^{ee}}{2} \geq 113(C+1)^2 \left( \frac{(\log \frac{2K}{\delta_k} + \log\log \frac{96(C+1)^2}{\Delta_{0,1}^2})}{\Delta_{0,1}^2} + \sum_{a=2}^{K} \frac{(\log \frac{K}{\delta} + \log\log \frac{96(C+1)^2}{\max\{\Delta_{1,a}^2,\Delta_{0,a}^2\}})}{\max\{\Delta_{1,a}^2, \Delta_{0,a}^2\}} \right).$$

Also, from the algorithm design, we know at the start of phase $k$, $|\mathcal{H}^{ee}| \leq T_{k-1}^{ee}$. Since we take $\beta_k = 2^k$, we have $T_{k-1}^{ee} \leq \frac{T_k^{ee}}{2}$.

Combining these two results, we have

$$T_k^{ee} \geq |\mathcal{H}^{ee}| + 113(C+1)^2 \left( \frac{(\log \frac{2K}{\delta_k} + \log\log \frac{96(C+1)^2}{\Delta_{0,1}^2})}{\Delta_{0,1}^2} + \sum_{a=2}^{K} \frac{(\log \frac{K}{\delta} + \log\log \frac{96(C+1)^2}{\max\{\Delta_{1,a}^2,\Delta_{0,a}^2\}})}{\max\{\Delta_{1,a}^2, \Delta_{0,a}^2\}} \right).$$

The remaining work is to prove $\text{LCB}_a(\mathcal{H}^{ee}, \delta_k) \leq \mu_0$ holds for all $a \in [K]$, before the start of phase $k$. We complete this by discussing the value of $\hat{a}_{k-1}$.

- If $\hat{a}_{k-1} = \text{None}$, we have $\text{LCB}_a(\mathcal{H}^{ee}, \delta_k) < \text{LCB}_a(\mathcal{H}^{ee}, \delta_{k-1}) < \text{LCB}_a(\mathcal{H}^{ee}, \delta_{k-1}) \leq \mu_0$, forall $a \in [K]$.

- If $\hat{a}_{k-1} = \text{Not Complete}$, we can also assert $\text{LCB}_a(\mathcal{H}^{ee}, \delta_{k-1}) < \mu_0, \forall a \in [K]$, or the algorithm would take $\hat{a}_{k-1} \in [K]$. Then we can further assert $\text{LCB}_a(\mathcal{H}^{ee}, \delta_k) < \text{LCB}_a(\mathcal{H}^{ee}, \delta_{k-1}) < \mu_0$.

- If $\hat{a}_{k-1} \in [K]$, then we can firstly assert $\text{LCB}_a(\mathcal{H}^{ee}, \delta_{k-1}) < \mu_0, \forall a \neq \hat{a}_{k-1}$. Or the algorithm would output other arms instead of $\hat{a}_{k-1}$. Thus, we have $\text{LCB}_a(\mathcal{H}^{ee}, \delta_k) < \text{LCB}_a(\mathcal{H}^{ee}, \delta_{k-1}) < \mu_0, \forall a \neq \hat{a}_{k-1}$.

Then, we turn to analyze $\hat{a}_{k-1}$. Before the execution of line 13, the following inequalities must hold

$$\frac{X + \sum_{s=1}^{N_{\hat{a}_{k-1}}^{\text{ee}} - 1} X_{\hat{a}_{k-1},s}}{N_{\hat{a}_{k-1}}^{\text{ee}}} - C \cdot U\left(N_{\hat{a}_{k-1}}^{\text{ee}}, \frac{\delta_{k-1}}{K}\right) > \mu_0$$

$$\frac{\sum_{s=1}^{N_{\hat{a}_{k-1}}^{\text{ee}} - 1} X_{\hat{a}_{k-1},s}}{N_{\hat{a}_{k-1}}^{\text{ee}} - 1} - C \cdot U\left(N_{\hat{a}_{k-1}}^{\text{ee}} - 1, \frac{\delta_{k-1}}{K}\right) \leq \mu_0$$

Since we put the last collected sample $X$ into $Q$, we can assert $\hat{\mu}_{\hat{a}_{k-1}}$ before the start of phase $k$ must be $\frac{\sum_{s=1}^{N_{\hat{a}_{k-1}}^{\text{ee}} - 1} X_{\hat{a}_{k-1},s}}{N_{\hat{a}_{k-1}}^{\text{ee}} - 1}$. Since $U\left(N_{\hat{a}_{k-1}}^{\text{ee}} - 1, \frac{\delta_k}{K}\right) > U\left(N_{\hat{a}_{k-1}}^{\text{ee}} - 1, \frac{\delta_{k-1}}{K}\right)$, we know before the start of phase $k$, $\text{LCB}_{\hat{a}_{k-1}}(\mathcal{H}^{\text{ee}}, \delta_k) < \mu_0$ must hold.

$\square$

The following lemma shows the condition of Lemma B.5 can be fulfilled in phase $k \geq \max\left\{\kappa^{\text{ee}}, \lceil \log_2 \frac{\sum_{a=1}^{K} \frac{24}{\Delta_{0,a}^2}}{K} \rceil\right\} =: L$.

**Lemma B.7.** *Apply Algorithm 2 to a negative 1-identification instance $\nu$, for phase index $k \geq \max\left\{\kappa^{ee}, \lceil \log_2 \frac{\sum_{a=1}^{K} \frac{24}{\Delta_{0,a}^2}}{K} \rceil\right\} =: L$, before running the exploration (at the Line 2 of Algorithm 3), we have*

$$T_k^{ee} \geq |\mathcal{H}^{ee}| + \sum_{a=1}^{K} \frac{114(\log \frac{2K}{\delta_k} + \log\log \frac{96}{\Delta_{0,a}^2})}{\Delta_{0,a}^2}$$

*Proof of Lemma B.7.* By the Lemma B.1, we know $k \geq \lceil \log_2 \frac{\sum_{a=1}^{K} \frac{24}{\Delta_{0,a}^2}}{K} \rceil$ implies

$$\frac{T_k^{\text{ee}}}{2} \geq \sum_{a=1}^{K} \frac{114(\log \frac{2K}{\delta_k} + \log\log \frac{96}{\Delta_{0,a}^2})}{\Delta_{0,a}^2}.$$

Also, from the algorithm design, we know at the start of phase $k$, $|\mathcal{H}^{\text{ee}}| \leq T_{k-1}^{\text{ee}}$. Since we take $\beta_k = 2^k$, we have $T_{k-1}^{\text{ee}} \leq \frac{T_k^{\text{ee}}}{2}$.

Combining these two results, we have

$$T_k^{\text{ee}} \geq |\mathcal{H}^{\text{ee}}| + \sum_{a=1}^{K} \frac{114(\log \frac{2K}{\delta_k} + \log\log \frac{96}{\Delta_{0,a}^2})}{\Delta_{0,a}^2}$$

$\square$

Given the above preparations, we are now ready to bound $\tau_k^{\text{ee}}$ with certainty, for large enough phase index $k$. The following two lemmas correspond to positive and negative instances separately. These two lemmas are also the final preparations for two main theorems 5.2, 5.3.

**Lemma B.8.** *Apply Algorithm 2 to a positive 1-identification instance $\nu$, for phase index $k \geq \max\left\{\kappa^{ee}, \lceil \log_2 \frac{24(C+1)^2 H_1^{pos}}{K} \rceil\right\} =: L$, we have*

$$\tau_k^{ee} \leq 1000(C+1)^2 K \beta_{L-1} \log \frac{4K}{\delta_{L-1}} +$$

$$114(C+1)^2 \left( \frac{(\log \frac{2K}{\delta_k} + \log\log \frac{96(C+1)^2}{\Delta_{0,1}^2})}{\Delta_{0,1}^2} + \sum_{a=2}^{K} \frac{(\log \frac{K}{\delta} + \log\log \frac{96(C+1)^2}{\max\{\Delta_{1,a}^2, \Delta_{0,a}^2\}})}{\max\{\Delta_{1,a}^2, \Delta_{0,a}^2\}} \right)$$

*holds with certainty, and $\hat{a}_k \in [K], \mu_{\hat{a}_k} \geq \omega\mu_1 + (1-\omega)\mu_0$, where $\omega = \frac{C-1}{C+3}$.*

Reminder: $\tau_k^{\text{ee}}$ is the total pulling times in all the exploration periods up to end of phase $k$.

*Proof of Lemma B.8.* It is not hard to see $\frac{T_k^{\text{ee}}}{2} = 500(C+1)^2 K\beta_k \log \frac{4K}{\delta_k} \geq T_{k-1}^{\text{ee}}$ hold for all $k \geq 2$, as $\beta_k = 2^k$. By the Lemma B.1, we also know

$$k \geq \lceil \log_2 \frac{24(C+1)^2 H_1^{\text{pos}}}{K} \rceil$$

$$\Rightarrow \frac{T_k^{\text{ee}}}{2} \geq 113(C+1)^2 \left( \frac{(\log \frac{2K}{\delta_k} + \log \log \frac{96(C+1)^2}{\Delta_{0,1}^2})}{\Delta_{0,1}^2} + \sum_{a=2}^{K} \frac{(\log \frac{K}{\delta} + \log \log \frac{96(C+1)^2}{\max\{\Delta_{1,a}^2, \Delta_{0,a}^2\}})}{\max\{\Delta_{1,a}^2, \Delta_{0,a}^2\}} \right).$$

Thus, we can conclude for $k \geq \lceil \log_2 \frac{24(C+1)^2 H_1^{\text{pos}}}{K} \rceil$, we have

$$T_k^{\text{ee}} = \frac{T_k^{\text{ee}}}{2} + \frac{T_k^{\text{ee}}}{2}$$

$$\geq T_{k-1}^{\text{ee}} + 113(C+1)^2 \left( \frac{(\log \frac{2K}{\delta_k} + \log \log \frac{96(C+1)^2}{\Delta_{0,1}^2})}{\Delta_{0,1}^2} + \sum_{a=2}^{K} \frac{(\log \frac{2K}{\delta} + \log \log \frac{96(C+1)^2}{\max\{\Delta_{1,a}^2, \Delta_{0,a}^2\}})}{\max\{\Delta_{1,a}^2, \Delta_{0,a}^2\}} \right).$$

From the algorithm design, we know the total pulling times in the exploration period up to phase $k-1$ is at most $T_{k-1}^{\text{ee}}$. By the Lemma B.6, we can validate the conditions in Lemma B.4 holds for $k \geq \max\left\{\kappa^{\text{ee}}, \lceil \log_2 \frac{24(C+1)^2 H_1^{\text{pos}}}{K} \rceil\right\}$. Thus, we can assert $\hat{a}_k \in [K], \mu_{\hat{a}_k} \geq \omega\mu_1 + (1-\omega)\mu_0, \omega = \frac{C-1}{C+3}$ for $k \geq L$. The remaining work is to prove the upper bound of $\tau_k^{\text{ee}}$.

In the following, we use induction to prove

$$N_a(\tau_k^{\text{ee}}) \leq \max\left\{ N_a(\tau_{L-1}^{\text{ee}}), 113(C+1)^2 \left( \frac{\log \frac{2K}{\delta_k} + \log \log \frac{96(C+1)^2}{\max\{\Delta_{1,a}^2, \Delta_{0,a}^2\}}}{\max\{\Delta_{1,a}^2, \Delta_{0,a}^2\}} \right) \right\}$$

holds for all $k \geq L, a \in [K]$. By the Lemma B.4, we know the above inequality holds for $k = L, \forall a \in [K]$. Then if $k$ holds, for the case of $k+1$, we firstly derive

$$N_a(\tau_{k+1}^{\text{ee}}) \leq \max\left\{ N_a(\tau_k^{\text{ee}}), 113(C+1)^2 \left( \frac{\log \frac{2K}{\delta_{k+1}} + \log \log \frac{96(C+1)^2}{\max\{\Delta_{1,a}^2, \Delta_{0,a}^2\}}}{\max\{\Delta_{1,a}^2, \Delta_{0,a}^2\}} \right) \right\}.$$

The reason is similar to the proof in the Lemma B.4.

- If $N_a(\tau_k^{\text{ee}}) \geq 113(C+1)^2 \left( \frac{\log \frac{2K}{\delta_{k+1}} + \log \log \frac{96(C+1)^2}{\max\{\Delta_{1,a}^2, \Delta_{0,a}^2\}}}{\max\{\Delta_{1,a}^2, \Delta_{0,a}^2\}} \right)$, from the condition $k \geq \kappa^{\text{ee}}$, we know

$$\left| \frac{\sum_{s=1}^{t} X_{a,s}^{\text{ee}}}{t} - \mu_a \right| < U(t, \frac{\delta_{k+1}}{K})$$

holds for all $t \in \mathbb{N}, a \in [K]$. Following the same discussion in Lemma B.4, the algorithm will never pull arm $a$ due to its upper confidence bound is below $\mu_1$, while the upper confidence bound of arm 1 is always above $\mu_1$. Thus, $N_a(\tau_{k+1}^{\text{ee}}) = N_a(\tau_k^{\text{ee}})$.

- If $N_a(\tau_k^{\text{ee}}) < 113(C+1)^2 \left( \frac{\log \frac{2K}{\delta_{k+1}} + \log \log \frac{96(C+1)^2}{\max\{\Delta_{1,a}^2, \Delta_{0,a}^2\}}}{\max\{\Delta_{1,a}^2, \Delta_{0,a}^2\}} \right)$, we can still conclude

$$\left| \frac{\sum_{s=1}^{t} X_{a,s}^{\text{ee}}}{t} - \mu_a \right| < U(t, \frac{\delta_{k+1}}{K})$$

from the condition $k \geq \kappa^{\text{ee}}$. Then arm $a$ would be either being output or stopping pulling before $N_a(\tau^{\text{ee}}_{k+1})$ is no less than $113(C+1)^2 \left( \dfrac{\log \frac{2K}{\delta_{k+1}} + \log\log \frac{96(C+1)^2}{\max\{\Delta^2_{1,a}, \Delta^2_{0,a}\}}}{\max\{\Delta^2_{1,a}, \Delta^2_{0,a}\}} \right)$.

Use the induction, we can conclude

$$
\begin{aligned}
N_a(\tau^{\text{ee}}_{k+1}) &\leq \max\left\{ N_a(\tau^{\text{ee}}_k), 113(C+1)^2 \left( \frac{\log \frac{2K}{\delta_{k+1}} + \log\log \frac{96(C+1)^2}{\max\{\Delta^2_{1,a}, \Delta^2_{0,a}\}}}{\max\{\Delta^2_{1,a}, \Delta^2_{0,a}\}} \right) \right\} \\
&\leq \max\left\{ N_a(\tau^{\text{ee}}_{L-1}), 113(C+1)^2 \left( \frac{\log \frac{2K}{\delta_k} + \log\log \frac{96(C+1)^2}{\max\{\Delta^2_{1,a}, \Delta^2_{0,a}\}}}{\max\{\Delta^2_{1,a}, \Delta^2_{0,a}\}} \right), \right. \\
&\qquad\qquad \left. 113(C+1)^2 \left( \frac{\log \frac{2K}{\delta_{k+1}} + \log\log \frac{96(C+1)^2}{\max\{\Delta^2_{1,a}, \Delta^2_{0,a}\}}}{\max\{\Delta^2_{1,a}, \Delta^2_{0,a}\}} \right) \right\} \\
&= \max\left\{ N_a(\tau^{\text{ee}}_{L-1}), 113(C+1)^2 \left( \frac{\log \frac{2K}{\delta_{k+1}} + \log\log \frac{96(C+1)^2}{\max\{\Delta^2_{1,a}, \Delta^2_{0,a}\}}}{\max\{\Delta^2_{1,a}, \Delta^2_{0,a}\}} \right) \right\}
\end{aligned}
$$

The induction is completed.

Then, for $k \geq L$, we can conclude

$$
\begin{aligned}
\tau^{\text{ee}}_k &\overset{\text{Lemma 4.2}}{\leq} K + \sum_{a=1}^{K} N_a(\tau^{\text{ee}}_k) \\
&\leq K + \sum_{a=1}^{K} N_a(\tau^{\text{ee}}_{L-1}) + \sum_{a=1}^{K} 113(C+1)^2 \left( \frac{\log \frac{2K}{\delta_{k+1}} + \log\log \frac{96(C+1)^2}{\max\{\Delta^2_{1,a}, \Delta^2_{0,a}\}}}{\max\{\Delta^2_{1,a}, \Delta^2_{0,a}\}} \right) \\
&\leq 1000(C+1)^2 K \beta_{L-1} \log \frac{4K}{\delta_{L-1}} + \\
&\qquad 114(C+1)^2 \left( \frac{(\log \frac{2K}{\delta_k} + \log\log \frac{96(C+1)^2}{\Delta^2_{0,1}})}{\Delta^2_{0,1}} + \sum_{a=2}^{K} \frac{(\log \frac{K}{\delta} + \log\log \frac{96(C+1)^2}{\max\{\Delta^2_{1,a}, \Delta^2_{0,a}\}})}{\max\{\Delta^2_{1,a}, \Delta^2_{0,a}\}} \right).
\end{aligned}
$$

$\square$

**Lemma B.9.** *Apply Algorithm 2 to a negative 1-identification instance $\nu$, for phase index $k \geq \max\left\{ \kappa^{ee}, \lceil \log_2 \frac{\sum_{a=1}^{K} \frac{24}{\Delta^2_{0,a}}}{K} \rceil \right\} =: L$, we have*

$$
\tau^{ee}_k \leq 1000(C+1)^2 K \beta_{L-1} \log \frac{4K}{\delta_{L-1}} + \sum_{a=1}^{K} \frac{114(\log \frac{2K}{\delta_k} + \log\log \frac{96}{\Delta^2_{0,a}})}{\Delta^2_{0,a}}.
$$

*holds with certainty. And $\hat{a}_k =$ None.*

*Proof of Lemma B.9.* Similar to the argument in the proof of Lemma B.8, we can conclude for phase index $k \geq \lceil \log_2 \frac{\sum_{a=1}^{K} \frac{24}{\Delta^2_{0,a}}}{K} \rceil$, we have

$$
T^{\text{ee}}_k = \frac{T^{\text{ee}}_k}{2} + \frac{T^{\text{ee}}_k}{2}
$$

$$\geq T^{\text{ee}}_{k-1} + \sum_{a=1}^{K} \frac{113(\log \frac{2K}{\delta_k} + \log \log \frac{96}{\Delta_{0,a}^2})}{\Delta_{0,a}^2}.$$

From the algorithm design, we know the total pulling times in the exploration period up to phase $k-1$ is at most $T^{\text{ee}}_{k-1}$. By the Lemma B.5, we can validate the conditions of Lemma B.5 hold for $k \geq \left\{ \kappa^{\text{ee}}, \lceil \log_2 \frac{\sum_{a=1}^{K} \frac{24}{\Delta_{0,a}^2}}{K} \rceil \right\}$. Thus, we have proved $\hat{a}_k = \text{None}$ for $k \geq \left\{ \kappa^{\text{ee}}, \lceil \log_2 \frac{\sum_{a=1}^{K} \frac{24}{\Delta_{0,a}^2}}{K} \rceil \right\}$. The remaining work is to prove the upper bound of $N_a(\tau^{\text{ee}}_k)$.

In the following, we use induction to prove

$$N_a(\tau^{\text{ee}}_k) \leq \max \left\{ N_a(\tau^{\text{ee}}_{L-1}), \frac{113(\log \frac{2K}{\delta_k} + \log \log \frac{96}{\Delta_{0,a}^2})}{\Delta_{0,a}^2} \right\}$$

holds for all $k \geq L, a \in [K]$. By the Lemma B.5, we know the above inequality holds for $k = L, \forall a \in [K]$. Then if $k$ holds, for the case of $k+1$, we can further derive

$$N_a(\tau^{\text{ee}}_{k+1}) \leq \max \left\{ N_a(\tau^{\text{ee}}_k), \frac{113(\log \frac{2K}{\delta_{k+1}} + \log \log \frac{96}{\Delta_{0,a}^2})}{\Delta_{0,a}^2} \right\}.$$

The reason is similar to the proof in the Lemma B.5.

- If $N_a(\tau^{\text{ee}}_k) \geq \frac{113(\log \frac{2K}{\delta_{k+1}} + \log \log \frac{96}{\Delta_{0,a}^2})}{\Delta_{0,a}^2}$, from the condition $k \geq \kappa^{\text{ee}}$, we know

$$\left| \frac{\sum_{s=1}^{t} X^{\text{ee}}_{a,s}}{t} - \mu_a \right| < U(t, \frac{\delta_{k+1}}{K})$$

holds for all $t \in \mathbb{N}, a \in [K]$. Following the same discussion in Lemma B.5, the algorithm will never pull arm $a$ due to its upper confidence bound is below $\mu_0$. If its upper confidence bound is the highest, the algorithm would output None. Thus, $N_a(\tau^{\text{ee}}_{k+1}) = N_a(\tau^{\text{ee}}_k)$.

- If $N_a(\tau^{\text{ee}}_k) < \frac{113(\log \frac{2K}{\delta_{k+1}} + \log \log \frac{96}{\Delta_{0,a}^2})}{\Delta_{0,a}^2}$, we can still conclude

$$\left| \frac{\sum_{s=1}^{t} X^{\text{ee}}_{a,s}}{t} - \mu_a \right| < U(t, \frac{\delta_{k+1}}{K})$$

from the condition $k \geq \kappa^{\text{ee}}$. Then arm $a$ will never get pulled before $N_a(\tau^{\text{ee}}_{k+1})$ is no less than $\frac{113(\log \frac{2K}{\delta_k} + \log \log \frac{96}{\Delta_{0,a}^2})}{\Delta_{0,a}^2}$, as its upper confidence bound is already smaller than $\mu_0$ before that happens.

Use the induction, we can conclude

$$N_a(\tau^{\text{ee}}_{k+1}) \leq \max \left\{ N_a(\tau^{\text{ee}}_k), 1 \frac{113(\log \frac{2K}{\delta_{k+1}} + \log \log \frac{96}{\Delta_{0,a}^2})}{\Delta_{0,a}^2} \right\}$$

$$\leq \max \left\{ N_a(\tau^{\text{ee}}_{L-1}), \frac{113(\log \frac{2K}{\delta_k} + \log \log \frac{96}{\Delta_{0,a}^2})}{\Delta_{0,a}^2}, \frac{113(\log \frac{2K}{\delta_{k+1}} + \log \log \frac{96}{\Delta_{0,a}^2})}{\Delta_{0,a}^2} \right\}$$

$$= \max \left\{ N_a(\tau^{\text{ee}}_{L-1}), \frac{113(\log \frac{2K}{\delta_{k+1}} + \log \log \frac{96}{\Delta_{0,a}^2})}{\Delta_{0,a}^2} \right\}$$

The induction is completed.

Then, for $k \geq L$, we can derive

$$\tau_k^{\text{ee}} \overset{\text{Lemma } 4.2}{\leq} K + \sum_{a=1}^{K} N_a(\tau_k^{\text{ee}})$$

$$\leq K + \sum_{a=1}^{K} N_a(\tau_{L-1}^{\text{ee}}) + \sum_{a=1}^{K} \frac{113(\log \frac{2K}{\delta_{k+1}} + \log\log \frac{96}{\Delta_{0,a}^2})}{\Delta_{0,a}^2}$$

$$\leq 1000(C+1)^2 K \beta_{L-1} \log \frac{4K}{\delta_{L-1}} + \frac{114(\log \frac{2K}{\delta_{k+1}} + \log\log \frac{96}{\Delta_{0,a}^2})}{\Delta_{0,a}^2}$$

$\square$

## C. Lower Bound

### C.1. Negative Case

*Proof of Theorem 5.4.* Following the section 2.1 in (Garivier & Kaufmann, 2016), define

$$\text{Alt}(\nu) = \{\nu' : i^*(\nu') \neq \textsf{None}\},$$

and the kl-divergence between two Gaussian distribution $d(N(\mu_a, 1), N(\lambda_a, 1)) = \frac{(\mu_a - \lambda_a)^2}{2}$, the kl-diverence between two bernoulli distribution $kl(\delta, 1-\delta) = \delta \log \frac{\delta}{1-\delta} + (1-\delta) \log \frac{1-\delta}{\delta}$.

Following the same step in the proof of Theorem 1, from (Garivier & Kaufmann, 2016), we can conclude

$$kl(\delta, 1-\delta) \leq \mathbb{E}_\nu \tau \sup_{w \in \Sigma_K} \inf_{\lambda \in \text{Alt}(\nu)} \sum_{a=1}^{K} w_a \frac{(\mu_a - \lambda_a)^2}{2}.$$

By the example 1 in the (Degenne & Koolen, 2019), we can derive

$$\frac{1}{\sup_{w \in \Sigma_K} \inf_{\lambda \in \text{Alt}(\nu)} \sum_{a=1}^{K} w_a \frac{(\mu_a - \lambda_a)^2}{2}} = \sum_{a=1}^{K} \frac{2}{\Delta_{0,a}^2},$$

which means $\mathbb{E}_\nu \tau \geq kl(\delta, 1-\delta) \sum_{a=1}^{K} \frac{2}{\Delta_{0,a}^2} = \Omega(H_1^{\text{neg}} \log \frac{1}{\delta})$.  $\square$

### C.2. Positive Case

In this section, we slightly adapt the conclusion in(Katz-Samuels & Jamieson, 2020) and (Kaufmann et al., 2016) to prove theorem 5.5 and 5.6.

*Proof of Theorem 5.5.* For this instance $\nu$, take $\{a_i\}_{i=1}^{K}$ as its permutation of the mean reward vector, such that the mean reward of the $i^{th}$ arm is $\mu_{a_i}$. Then, we consider an alternative instance $\nu'$, whose mean reward of $i^{th}$ arm is

$\mu_i' = \begin{cases} \mu_{a_i} & \mu_{a_i} \leq \mu_0 \\ \mu_0 - \Delta & \mu_{a_i} > \mu_0 \end{cases}$ for some $\Delta > 0$. The answer set $i^*(\nu') = \{\textsf{None}\}$. Apply the Lemma 1 in (Kaufmann et al., 2016), we get

$$\sum_{i:\mu_i > \mu_0} \frac{(\mu_i - \mu_0 + \Delta)^2}{2} \mathbb{E}_\nu N_{a_i}(\tau) \geq kl(\delta, 1-\delta),$$

where $kl(\delta, 1-\delta) = \delta \log \frac{\delta}{1-\delta} + (1-\delta) \log \frac{1-\delta}{\delta}$.

By the assumption, we have $\mu_1 \geq \mu_2 \geq \cdots \geq \mu_K$, which implies $\frac{(\mu_i - \mu_0 + \Delta)^2}{(\mu_1 - \mu_0 + \Delta)^2} \leq 1$ holds for all $i$ such that $\mu_i \geq \mu_0$. Thus, we can conclude

$$
\begin{aligned}
\mathbb{E}_\nu \tau &\geq \sum_{i:\mu_i > \mu_0} \mathbb{E}_\nu N_{a_i}(\tau) \\
&\geq \sum_{i:\mu_i > \mu_0} \frac{(\mu_i - \mu_0 + \Delta)^2}{(\mu_1 - \mu_0 + \Delta)^2} \mathbb{E}_\nu N_{a_i}(\tau) \\
&\geq \frac{2kl(\delta, 1-\delta)}{(\mu_1 - \mu_0 + \Delta)^2}.
\end{aligned}
$$

Since $kl(\delta, 1-\delta) = \Omega(\log \frac{1}{\delta})$ and $\mathbb{E}_\nu \tau \geq \frac{2kl(\delta, 1-\delta)}{(\mu_1 - \mu_0 + \Delta)^2}$ holds for all $\Delta > 0$, we can conclude $\mathbb{E}_{\nu,\text{alg}} \tau \geq \Omega(\frac{\log \frac{1}{\delta}}{\Delta_{0,1}^2})$. $\qquad\square$

*Proof of Theorem 5.6.* The algorithm might take $\mu_1 > \mu_0 > \mu_2 \geq \cdots \geq \mu_K$ as prior knowledge and solve the problem by identifying 1 arm among 1 good arm. We can directly apply the Theorem 1 in (Katz-Samuels & Jamieson, 2020) to derive a lower bound, which asserts we can find a positive instance $\nu$ whose mean reward vector is a permutation of vector $\{\mu_a\}_{a=1}^K$ and the threshold is $\mu_0$, such that

$$
\mathbb{E}_\nu \tau \geq \frac{1}{64} \left( -\frac{1}{\Delta_{1,2}^2} + \sum_{a=3}^K \frac{1}{\Delta_{1,a}^2} \right). \tag{5}
$$

By the Theorem 5.5, we can also conclude $\mathbb{E}_\nu \tau \geq \Omega(\frac{\log \frac{1}{\delta}}{\Delta_{0,1}^2}) \geq \Omega(\frac{1}{\Delta_{1,2}^2})$ by the assumption $\mu_1 > \mu_0 > \mu_2$ and $\delta < \frac{1}{16}$. Combining the result with (5), we get

$$
\mathbb{E}_\nu \tau \geq \Omega \left( \sum_{a=2}^K \frac{1}{\Delta_{1,a}^2} \right) = \Omega\left(H_1\right).
$$

$\qquad\square$

## C.3. Lower Bound for a Suboptimal Arm

Before proving Theorem 5.7, we need to firstly introduce a lemma, talking about a "high probability lower bound" of the optimal arm, if it is also the uniqe qualified arm.

**Lemma C.1** (High Probability Lower Bound). *Denote $\nu$ as a 1-dentification Gaussian problem instance with fixed variance 1. Denote $K$, $\{\mu_a\}_{a=0}^K$ as the number of alternative arms and mean reward of all the arms in $\nu$. If $\mu_1 > \mu_0 > \mu_2 > \cdots > \mu_K$, then for any $\delta \in (0,1)$, and any $\delta$-PAC algorithm, we have*

$$
\Pr_\nu(N_1(\tau) \geq \frac{\log \frac{1}{\delta}}{C(\mu_1 - \mu_0)^2}) \geq 1 - \delta - \delta^{1 - \frac{1}{4} - \frac{1}{2C}} - \delta^{\frac{C}{32}},
$$

*where $C$ can be any values greater than 1.*

*Proof of Lemma C.1.* Define instance $\nu'$ with mean rewards $\tilde{\mu}_1, \mu_2, \cdots, \mu_K$ where $\tilde{\mu}_1 < \mu_0$. In other words, the only difference between $\nu$ and $\nu'$ is the mean reward of arm 1, while others are all the same. Since the algorithm is $\delta$-PAC, we know

$$
\Pr_\nu(\text{output arm } 1) > 1 - \delta
$$
$$
\Pr_{\nu'}(\text{output none}) > 1 - \delta.
$$

Apply the transportation equality, we get

$$
\delta > \Pr_{\nu'}(\text{output arm } 1) = \mathbb{E}_\nu \mathbb{1}(\text{output arm } 1) \exp\left( -\sum_{s=1}^{N_1(\tau)} Z_{1,s} \right)
$$

where $Z_{1,s}$ is the realized KL divergence, $Z_{1,s} = \log \frac{\exp(-\frac{(X_{1,s}-\mu_1)^2}{2})}{\exp(-\frac{(X_{1,s}-\tilde{\mu}_1)^2}{2})} = \frac{(\mu_1-\tilde{\mu}_1)(2X_{1,s}-\mu_1-\tilde{\mu}_1)}{2}$, $X_{1,s} \sim N(\mu_1, 1)$. Or we can directly assume $Z_{1,s} \overset{i.i.d}{\sim} N(\frac{(\mu_1-\tilde{\mu}_1)^2}{2}, (\mu_1-\tilde{\mu}_1)^2)$. Before moving on, we firstly prove a concentration result for the sum $\sum_{s=1}^{t} Z_{1,s}$. For any fixed number $N, I$, define $\xi_{N,I} = \left\{\max_{1\le t\le N} \sum_{s=1}^{t} \left(Z_{1,s} - \frac{(\mu_1-\tilde{\mu}_1)^2}{2}\right) < I\right\}$. We can assert $\Pr_{Z_{1,s} \overset{i.i.d}{\sim} N(\frac{(\mu_1-\tilde{\mu}_1)^2}{2},(\mu_1-\tilde{\mu}_1)^2)} (\xi_{N,I}) \ge 1 - \exp\left(-\frac{I^2}{2N(\mu_1-\tilde{\mu}_1)^2}\right)$, by the following application of maximal inequality (Theorem 3.10 in (Lattimore & Szepesvári, 2020)).

$$\Pr_{Z_{1,s} \overset{i.i.d}{\sim} N(\frac{(\mu_1-\tilde{\mu}_1)^2}{2},(\mu_1-\tilde{\mu}_1)^2)} \left(\max_{1\le t\le N} \sum_{s=1}^{t} \left(Z_{1,s} - \frac{(\mu_1-\tilde{\mu}_1)^2}{2}\right) > I\right)$$

$$= \Pr_{\tilde{Z}_{1,s} \overset{i.i.d}{\sim} N(0,(\mu_1-\tilde{\mu}_1)^2)} \left(\max_{1\le t\le N} \sum_{s=1}^{t} \tilde{Z}_{1,s} > I\right)$$

$$\overset{\lambda=\frac{I}{N(\mu_1-\tilde{\mu}_1)^2}}{=} \Pr_{\tilde{Z}_{1,s} \overset{i.i.d}{\sim} N(0,(\mu_1-\tilde{\mu}_1)^2)} \left(\max_{1\le t\le N} \prod_{s=1}^{t} \exp\left(\lambda\tilde{Z}_{1,s}\right) > \exp(\lambda I)\right)$$

$$\overset{\text{Maximal Inequality}}{\le} \frac{\mathbb{E}_{\tilde{Z}_{1,s} \overset{i.i.d}{\sim} N(0,(\mu_1-\tilde{\mu}_1)^2)} \prod_{s=1}^{N} \exp\left(\lambda\tilde{Z}_{1,s}\right)}{\exp(\lambda I)}$$

$$= \frac{\exp\left(\frac{N\lambda^2(\mu_1-\tilde{\mu}_1)^2}{2}\right)}{\exp(\lambda I)} = \exp\left(-\frac{I^2}{2N(\mu_1-\tilde{\mu}_1)^2}\right).$$

To apply the Maximal Inequality, we need to validate $\{\prod_{s=1}^{t} \exp\left(\lambda\tilde{Z}_{1,s}\right)\}_{t=1}^{+\infty}$ is a submartingale, which is not hard. By the Jensen Inequality, we have

$$\mathbb{E}\left[\prod_{s=1}^{t+1} \exp\left(\lambda\tilde{Z}_{1,s}\right) \mid \prod_{s=1}^{t} \exp\left(\lambda\tilde{Z}_{1,s}\right)\right]$$

$$= \mathbb{E}\left[\exp\left(\lambda\tilde{Z}_{1,t+1}\right)\right] \prod_{s=1}^{t} \exp\left(\lambda\tilde{Z}_{1,s}\right)$$

$$\ge \exp\left(\mathbb{E}\left[\lambda\tilde{Z}_{1,t+1}\right]\right) \prod_{s=1}^{t} \exp\left(\lambda\tilde{Z}_{1,s}\right)$$

$$= \prod_{s=1}^{t} \exp\left(\lambda\tilde{Z}_{1,s}\right).$$

Adding the concentration term into the transportation equality, we get

$$\delta > \Pr_{\nu'}(\text{output arm } 1)$$

$$= \mathbb{E}_{\nu} \mathbb{1}(\text{output arm } 1) \exp\left(-\sum_{s=1}^{N_1(\tau)} Z_{1,s}\right)$$

$$\ge \mathbb{E}_{\nu} \mathbb{1}(\text{output arm } 1) \mathbb{1}(N_1(\tau) < \frac{\log\frac{1}{\delta}}{C(\mu_1-\tilde{\mu}_1)^2}) \mathbb{1}(\xi_{\frac{\log\frac{1}{\delta}}{C(\mu_1-\tilde{\mu}_1)^2}, I})$$

$$\exp\left(\sum_{s=1}^{N_1(\tau)} \left(\frac{(\mu_1-\tilde{\mu}_1)^2}{2} - Z_{1,s}\right)\right) \exp\left(-N_1(\tau)\frac{(\mu_1-\tilde{\mu}_1)^2}{2}\right)$$

$$\ge \exp(-I) \exp\left(-\frac{\log\frac{1}{\delta}}{2C}\right) \mathbb{E}_{\nu} \mathbb{1}(\text{output arm } 1) \mathbb{1}(N_1(\tau) < \frac{\log\frac{1}{\delta}}{C(\mu_1-\tilde{\mu}_1)^2}) \mathbb{1}(\xi_{\frac{\log\frac{1}{\delta}}{C(\mu_1-\tilde{\mu}_1)^2}, I}).$$

The last inequality is equivalent to

$$\exp(I)\exp\left(\frac{\log\frac{1}{\delta}}{2C}\right)\delta \geq \mathbb{E}_\nu\mathbb{1}(\text{output arm }1)\mathbb{1}(N_1(\tau) < \frac{\log\frac{1}{\delta}}{C(\mu_1-\tilde{\mu}_1)^2})\mathbb{1}(\xi_{\frac{\log\frac{1}{\delta}}{C(\mu_1-\tilde{\mu}_1)^2},I})$$

$$\overset{\Pr(A\cap B)\geq\Pr(A)-\Pr(\neg B)}{\Longrightarrow}\exp(I)\exp\left(\frac{\log\frac{1}{\delta}}{2C}\right)\delta \geq \left(\Pr_\nu(N_1(\tau) < \frac{\log\frac{1}{\delta}}{C(\mu_1-\tilde{\mu}_1)^2}) - \Pr_\nu(\neg\text{output arm }1) - \Pr_{\nu_1}(\neg\xi_{\frac{\log\frac{1}{\delta}}{C(\mu_1-\tilde{\mu}_1)^2},I})\right)$$

$$\Leftrightarrow \Pr_\nu(\neg\text{output arm }1) + \exp(I)\exp\left(\frac{\log\frac{1}{\delta}}{2C}\right)\delta + \Pr_{\nu_1}(\neg\xi_{\frac{\log\frac{1}{\delta}}{C(\mu_1-\tilde{\mu}_1)^2},I}) \geq \Pr_\nu(N_1(\tau) < \frac{\log\frac{1}{\delta}}{C(\mu_1-\tilde{\mu}_1)^2})$$

$$\Rightarrow \delta + \exp(I)\exp\left(\frac{\log\frac{1}{\delta}}{2C}\right)\delta + \exp\left(-\frac{CI^2}{2\log\frac{1}{\delta}}\right) \geq \Pr_\nu(N_1(\tau) < \frac{\log\frac{1}{\delta}}{C(\mu_1-\tilde{\mu}_1)^2})$$

$$\Leftrightarrow \Pr_\nu(N_1(\tau) \geq \frac{\log\frac{1}{\delta}}{C(\mu_1-\tilde{\mu}_1)^2}) \geq 1 - \delta - \exp(I)\exp\left(\frac{\log\frac{1}{\delta}}{2C}\right)\delta - \exp\left(-\frac{CI^2}{2\log\frac{1}{\delta}}\right).$$

Take $I = \frac{1}{4}\log\frac{1}{\delta}$, we get

$$\Pr_\nu(N_1(\tau) \geq \frac{\log\frac{1}{\delta}}{C(\mu_1-\tilde{\mu}_1)^2}) \geq 1 - \delta - \delta^{1-\frac{1}{4}-\frac{1}{2C}} - \delta^{\frac{C}{32}}.$$

As the above inequality holds for any $\tilde{\mu}_1 < \mu_0$, we can take $\tilde{\mu}_1 \to \mu_0$. Thus, $\Pr_\nu(N_1(\tau) \geq \frac{\log\frac{1}{\delta}}{C(\mu_1-\mu_0)^2}) \geq 1 - \delta - \delta^{1-\frac{1}{4}-\frac{1}{2C}} - \delta^{\frac{C}{32}}$. We complete the proof. $\square$

Then, we are ready to prove Theorem 5.7.

*Proof of Theorem 5.7.* Take $M_2 = 512 * 3 = 1536$, $M_1 = 8M_2$.

Let $\bar{\Delta}_{0,1}$ to be small enough, such that

$$\frac{\exp\left(-4\sqrt{3}-2\right)}{2\bar{\Delta}_{0,1}} > \sup_{\nu'\in\mathcal{S}^{\text{pos}}_{\Delta_{0,a}}\cup\mathcal{S}^{\text{neg}}_{\Delta_{0,a}}}\mathbb{E}_{\nu'}\tau, \forall a \geq 2.$$

In the following, we take $\mu_1$ as any value in $(\mu_0, \mu_0 + \bar{\Delta}_{0,1}]$, and take $\Delta_{0,1} = \mu_1 - \mu_0$. We are going to prove, for all $a \geq 2$, we have $\mathbb{E}_\nu N_a(\tau) \geq \frac{\log\frac{1}{\Delta^2_{1,0}}}{M_1\Delta^2_{1,a}}$.

Prove by contradiction. If there exists an arm $a \geq 2$, such that $\mathbb{E}_\nu N_a(\tau) < \frac{\log\frac{1}{\Delta^2_{0,1}}}{M_1\Delta^2_{0,a}}$, define instance $\nu'_a$ by taking the mean reward of $i$th arm as $\begin{cases} 2\mu_0 - \mu_1 & i = 1 \\ 2\mu_0 - \mu_a & i = a \\ \mu_i & i \neq 1, a \end{cases}$ for all arm $a \in [K], a \geq 2$. In other words, we "flip" the mean reward of arm 1 and $a$, while keeping others the same as the instance $\nu$. From this definition, we know $\nu'_a \in \mathcal{S}^{\text{pos}}_{\Delta_{0,a}}$, $i^*(\nu'_a) = \{a\}$.

By the Markov Inequality, we know

$$\Pr_\nu\left(N_a(\tau) < \frac{\log\frac{1}{\Delta^2_{0,1}}}{M_2\Delta^2_{0,a}}\right) \geq 1 - \frac{M_2}{M_1}.$$

Define $\tilde{\tau} = \min\{\tau, \min\{t : N_1(t) \geq \frac{1}{\Delta^2_{0,1}}\}\}$. Then we get $N_1(\tau) \geq \frac{1}{\Delta^2_{0,1}} \Rightarrow N_1(\tilde{\tau}) = \frac{1}{\Delta^2_{0,1}}$. According to the lemma C.1, we have

$$\Pr_\nu\left(N_1(\tau) \geq \frac{\log\frac{1}{\delta}}{8\Delta^2_{0,1}}\right) > 1 - 3\sqrt{\delta}.$$

Through simple calculation, we can derive

$$\Pr_{\nu}\left(N_a(\tilde{\tau}) \leq \frac{\log\frac{1}{\Delta_{0,1}^2}}{M_2\Delta_{0,a}^2} \text{ and } N_1(\tilde{\tau}) = \frac{1}{\Delta_{0,1}^2}\right)$$

$$= \Pr_{\nu}\left(N_a(\tilde{\tau}) \leq \frac{\log\frac{1}{\Delta_{0,1}^2}}{M_2\Delta_{0,a}^2}, N_1(\tau) \geq \frac{1}{\Delta_{0,1}^2}\right)$$

$$\overset{\delta < \min\{\frac{1}{e^8}, \frac{1}{24^2}\}}{\geq} \Pr_{\nu}\left(N_a(\tau) \leq \frac{\log\frac{1}{\Delta_{0,1}^2}}{M_2\Delta_{0,a}^2}, N_1(\tau) \geq \frac{\log\frac{1}{\delta}}{8\Delta_{0,1}^2}\right)$$

$$\geq \Pr_{\nu}\left(N_a(\tau) \leq \frac{\log\frac{1}{\Delta_{0,1}^2}}{M_2\Delta_{0,a}^2}\right) - \Pr_{\nu}\left(N_1(\tau) < \frac{\log\frac{1}{\delta}}{8\Delta_{0,1}^2}\right)$$

$$\geq 1 - \frac{M_2}{M_1} - 3\sqrt{\delta}.$$

Denote $\mathcal{G} = \left\{N_a(\tilde{\tau}) \leq \frac{\log\frac{1}{\Delta_{0,1}^2}}{M_2\Delta_{0,a}^2} \text{ and } N_1(\tilde{\tau}) = \frac{1}{\Delta_{0,1}^2}\right\}$ as the event, we can apply the transportation equality in the Lemma 18 of (Kaufmann et al., 2016), we have

$$\Pr_{\nu_a'}(\mathcal{G}) = \mathbb{E}_{\nu}\mathbb{1}(\mathcal{G})\exp\left(-\sum_{s=1}^{N_1(\tilde{\tau})} Z_{1,s} - \sum_{s=1}^{N_a(\tilde{\tau})} Z_{a,s}\right),$$

where

$$Z_{1,s} = \log\frac{\exp(-\frac{(X_{1,s}-\mu_1)^2}{2})}{\exp(-\frac{(X_{1,s}+\mu_1-2\mu_0)^2}{2})} = \frac{(X_{1,s}+\mu_1-2\mu_0)^2 - (X_{1,s}-\mu_1)^2}{2} = \frac{(2X_{1,s}-2\mu_0)(2\mu_1-2\mu_0)}{2},$$

$$Z_{a,s} = \log\frac{\exp(-\frac{(X_{a,s}-\mu_a)^2}{2})}{\exp(-\frac{(X_{a,s}+\mu_a-2\mu_0)^2}{2})} = \frac{(X_{a,s}+\mu_a-2\mu_0)^2 - (X_{a,s}-\mu_a)^2}{2} = \frac{(2X_{a,s}-2\mu_0)(2\mu_a-2\mu_0)}{2},$$

and $X_{1,s} \sim N(\mu_1, 1), X_{a,s} \sim N(\mu_a, 1)$. In other words, we can directly assume $Z_{1,s} \sim N(2\Delta_{0,1}^2, 4\Delta_{0,1}^2)$ and $Z_{a,s} \sim N(2\Delta_{0,a}^2, 4\Delta_{0,a}^2)$.

Let $I_1, I_a$ be two positive integers to be determined. Denote the concentration event of realized KL-divergence as

$\xi_1 = \left\{\max_{1 \leq t \leq \frac{1}{\Delta_{0,1}^2}} \sum_{s=1}^{t}\left(Z_{1,s} - 2\Delta_{0,1}^2\right) \leq I_1\right\}, \xi_a = \left\{\max_{1 \leq t \leq \frac{\log\frac{1}{\Delta_{0,1}^2}}{M_2\Delta_{0,a}^2}} \sum_{s=1}^{t}\left(Z_{a,s} - 2\Delta_{0,a}^2\right) \leq I_a\right\}$. Similar to

the proof of Lemma C.1, we can derive a probability bound for both events $\xi_1, \xi_a$. Notice that

$$\Pr_{Z_s \sim N(\mu,\sigma^2)}\left(\max_{1 \leq t \leq T} \sum_{s=1}^{t}(Z_s - \mu) > I\right)$$

$$\overset{\lambda = \frac{I}{T\sigma^2}}{\leq} \frac{\mathbb{E}_{Z_s \sim N(\mu,\sigma^2)}\prod_{s=1}^{T}\exp(\lambda(Z_s - \mu))}{\exp(\lambda I)}$$

$$= \frac{\exp(\frac{T\lambda^2\sigma^2}{2})}{\exp(\lambda I)}$$

$$= \exp\left(-\frac{I^2}{2T\sigma^2}\right),$$

holds for all positive $T$ and $I$. The first inequality is guaranteed by the maximal inequality of submartingale (Theorem 3.10

in (Lattimore & Szepesvári, 2020)). By the above inequality, we have

$$\Pr_{\nu}(\xi_1) \geq 1 - \exp\left(-\frac{I_1^2}{2 \cdot \frac{1}{\Delta_{0,1}^2} \cdot 4\Delta_{0,1}^2}\right) = 1 - \exp\left(-\frac{I_1^2}{8}\right)$$

$$\Pr_{\nu}(\xi_a) \geq 1 - \exp\left(-\frac{I_a^2}{2 \cdot \frac{\log \frac{1}{\Delta_{0,1}^2}}{M_2 \Delta_{0,a}^2} \cdot 4\Delta_{0,a}^2}\right) = 1 - \exp\left(-\frac{M_2 I_a^2}{8 \log \frac{1}{\Delta_{0,1}^2}}\right)$$

Plug in these inequalities to the transportation equality, we get

$$\Pr_{\nu_a'}(\mathcal{G})$$

$$\geq \mathbb{E}_\nu \mathbb{1}(\mathcal{G})\mathbb{1}(\xi_1)\mathbb{1}(\xi_a) \exp\left(-N_1(\tilde\tau) \cdot 2\Delta_{0,1}^2 - N_a(\tilde\tau) \cdot 2\Delta_{0,a}^2\right)$$

$$\exp\left(-\sum_{s=1}^{N_1(\tilde\tau)}\left(Z_{1,s} - 2\Delta_{0,1}^2\right) - \sum_{s=1}^{N_a(\tilde\tau)}\left(Z_{2,s} - 2\Delta_{0,a}^2\right)\right)$$

$$\geq \mathbb{E}_\nu \mathbb{1}(\mathcal{E})\mathbb{1}(\xi_1)\mathbb{1}(\xi_a) \exp\left(-N_1(\tilde\tau)2\Delta_{0,1}^2 - N_a(\tilde\tau)2\Delta_{0,a}^2\right) \exp\left(-I_1 - I_a\right)$$

$$\geq \mathbb{E}_\nu \mathbb{1}(\mathcal{E})\mathbb{1}(\xi_1)\mathbb{1}(\xi_a)$$

$$\exp\left(-\frac{\log \frac{1}{\Delta_{0,1}^2}}{M_2 \Delta_{0,a}^2} \cdot 2\Delta_{0,a}^2 - \frac{1}{\Delta_{0,1}^2} \cdot 2\Delta_{0,1}^2\right) \exp\left(-I_1 - I_2\right)$$

$$\geq \mathbb{E}_\nu \mathbb{1}(\mathcal{E})\mathbb{1}(\xi_1)\mathbb{1}(\xi_a) \exp\left(-I_1 - I_a\right) \exp\left(-\frac{2\log \frac{1}{\Delta_{0,1}^2}}{M_2} - 2\right)$$

$$\geq \exp\left(-I_1 - I_a\right) \exp\left(-\frac{2\log \frac{1}{\Delta_{0,1}^2}}{M_2} - 2\right)\left(\Pr_{\nu}(\mathcal{E}) - \Pr_{\nu}(\neg\xi_1) - \Pr_{\nu}(\neg\xi_a)\right)$$

$$\geq \exp\left(-I_1 - I_a\right) \exp\left(-\frac{2\log \frac{1}{\Delta_{0,1}^2}}{M_2} - 2\right)$$

$$\left(1 - \frac{M_2}{M_1} - 3\sqrt{\delta} - \exp\left(-\frac{M_2 I_a^2}{8\log \frac{1}{\Delta_{0,1}^2}}\right) - \exp(-\frac{I_1^2}{8})\right).$$

Take $I_a^2 = \frac{1}{16}\log \frac{1}{\Delta_{0,1}^2}$, $I_1 = 4\sqrt{3}$, $3\sqrt{\delta} < \frac{1}{8}$, $M_2 = 512 * 3 = 1536$, $M_1 = 8M_2$ we get

$$\Pr_{\nu_a'}(\mathcal{E})$$

$$\geq \exp\left(-\frac{1}{4}\sqrt{\log \frac{1}{\Delta_{0,1}^2}} - 4\sqrt{3}\right) \exp\left(-\frac{2\log \frac{1}{\Delta_{0,1}^2}}{M_2} - 2\right)$$

$$\left(1 - \frac{1}{8} - \frac{1}{8} - \exp(-\frac{M_2}{512}) - \exp(-3)\right)$$

$$\geq \exp\left(-4\sqrt{3}\right) \exp\left(-\left(\frac{2}{M_2} + \frac{1}{4}\right)\log \frac{1}{\Delta_{0,1}^2} - 2\right)\left(1 - \frac{1}{8} - \frac{1}{8} - \exp(-\frac{M_2}{512}) - \exp(-3)\right)$$

$$\geq \frac{1}{2}\exp\left(-4\sqrt{3}\right) \exp\left(-\frac{1}{2}\log \frac{1}{\Delta_{0,1}^2} - 2\right),$$

further

$$\Pr_{\nu'_a}\left(N_1(\tau) \geq \frac{1}{\Delta_{0,1}^2}\right)$$

$$\geq \Pr_{\nu}\left(N_a(\tilde{\tau}) \leq \frac{\log\frac{1}{\Delta_{0,1}^2}}{M_2\Delta_{0,a}^2} \text{ and } N_1(\tilde{\tau}) = \frac{1}{\Delta_{0,1}^2}\right)$$

$$\geq \frac{1}{2}\exp\left(-4\sqrt{3}\right)\exp\left(-\frac{1}{2}\log\frac{1}{\Delta_{0,1}^2} - 2\right)$$

$$= \frac{1}{2}\exp\left(-4\sqrt{3} - 2\right)\exp\left(-\log\frac{1}{\Delta_{0,1}^2} + \frac{1}{2}\log\frac{1}{\Delta_{0,1}^2}\right)$$

$$= \frac{1}{2}\exp\left(-4\sqrt{3} - 2\right)\exp\left(\frac{1}{2}\log\frac{1}{\Delta_{0,1}^2}\right)\Delta_{0,1}^2.$$

By the Markov Inequality, we have

$$\mathbb{E}_{\nu'_a}N_1(\tau) \geq \frac{1}{2}\exp\left(-4\sqrt{3} - 2\right)\exp\left(\frac{1}{2}\log\frac{1}{\Delta_{0,1}^2}\right) = \frac{\exp\left(-4\sqrt{3} - 2\right)}{2\Delta_{0,1}}.$$

From the construction of $\mu_1$, we know $\mathbb{E}_{\nu'_a}\tau \geq \frac{\exp(-4\sqrt{3}-2)}{2\Delta_{0,1}} > \sup_{\nu' \in \mathcal{S}_{\Delta_{0,a}}^{\mathrm{pos}} \cup \mathcal{S}_{\Delta_{0,a}}^{\mathrm{neg}}} \mathbb{E}_{\nu'}\tau$, contradicting to the fact that $\nu'_a \in \mathcal{S}_{\Delta_{0,a}}^{\mathrm{pos}}$. We complete the proof. $\qquad\square$

## D. Technical Inequality

### D.1. Inequality about $x$ and $\log\log x$

This section includes some mathematics inequalities for simplifying calculation.

**Lemma D.1.** *For any $b \geq a \geq 0$,*

- *If $b \geq e^2$, we have $x \geq b + 2a\log\log b \Rightarrow x \geq a\log\log(x) + b$.*

- *If $b, a \geq e$, we have $e \leq x \leq b + a\log\log b \Rightarrow x < a\log\log(x) + b$.*

The second inequality also implies $x \geq a\log\log(x) + b \Rightarrow x \geq b + a\log\log b$.

*Proof.* We prove the first claim. Easy to see $\frac{d(x - a\log\log x - b)}{dx} = 1 - \frac{a}{x\log x} = \frac{x\log x - a}{x\log x}$. Take $x_0 = b + 2a\log\log b$, then $x_0\log x_0 > x_0\log(e^2 + 2a\log\log e^2) > x_0\log e^2 = 2x_0 > a$. Thus $x - a\log\log(x) - b$ increases in the interval $(x_0, +\infty)$. On the other hand, easy to check

$$x_0 - a\log\log x_0 - b$$
$$= b + 2a\log\log b - a\log\log(b + 2a\log\log b) - b$$
$$= 2a\log\log b - a\log\log(b + 2a\log\log b)$$
$$= a\left(2\log\log b - \log\log(b + 2a\log\log b)\right)$$
$$= a\left(\log(\log b)^2 - \log\log(b + 2a\log\log b)\right)$$
$$= a\log\frac{(\log b)^2}{\log(b + 2a\log\log b)}$$
$$\geq a\log\frac{(\log b)^2}{\log(b + 2b\log\log b)}$$

$$= a \log \frac{(\log b)^2}{\log b + \log(1 + 2 \log \log b)},$$

Notice that

$$(\log b)^2 - \log b - \log(1 + 2 \log \log b)$$
$$= \log b(\log b - 1) - \log(1 + 2 \log \log b)$$
$$\overset{b \geq e^2}{\geq} \log b - \log(1 + 2 \log \log b)$$
$$\geq \log b - 2 \log \log b$$
$$\overset{x > 2 \log x, \forall x > 0}{>} 0,$$

we can conclude $x_0 - a \log \log x_0 - b > 0$ holds for all $x \geq b + 2a \log \log b$.

Then we turn to prove the second claim. As $\frac{d(x - a \log \log x - b)}{dx} = 1 - \frac{a}{x \log x} = \frac{x \log x - a}{x \log x}$, we know there is at most 1 zero point of $\frac{x \log x - a}{x \log x}$ in the interval $(e, b + a \log \log b)$. Thus,

$$\max_{e \leq x \leq b + a \log \log b} x - a \log \log x - b = \max\{x - a \log \log x - b|_{x=e}, x - a \log \log x - b|_{x=b+a \log \log b}\}.$$

Easy to see

$$e - a \log \log e - b = e - b < 0,$$

and

$$b + a \log \log b - a \log \log(b + a \log \log b) - b$$
$$= a \log \log b - a \log \log(b + a \log \log b)$$
$$< a \log \log b - a \log \log(b)$$
$$= 0.$$

That means $\max_{e \leq x \leq b + a \log \log b} x - a \log \log x - b < 0$. The second conclusion is done. $\qquad \square$

**Lemma D.2.** *For any* $\Delta \in (0, 1]$, $K \geq 2$, $\delta \in (0, \frac{1}{2}]$, $C \geq 1$, *we can conclude*

$$t > \frac{28C^2 \log \frac{2K}{\delta}}{\Delta^2} + \frac{16C^2 \log \left(\log \left(\frac{24C^2}{\Delta^2}\right)\right)}{\Delta^2}$$

$$\Rightarrow 2t > \frac{8C^2 \log \frac{2K}{\delta} + 8C^2 \log \log_2 e + 16C^2 \log \log 2t}{\Delta^2}$$

$$\Leftrightarrow C\sqrt{\frac{4 \log \frac{2K(\log_2 2t)^2}{\delta}}{t}} < \Delta$$

*Proof.* By simple calculation, we can derive

$$t > \frac{28C^2 \log \frac{2K}{\delta}}{\Delta^2} + \frac{16C^2 \log \left(\log \left(\frac{24C^2}{\Delta^2}\right)\right)}{\Delta^2}$$

$$\Leftrightarrow 2t > \frac{56C^2 \log \frac{2K}{\delta}}{\Delta^2} + \frac{32C^2 \log \left(\log \left(\frac{24C^2}{\Delta^2}\right)\right)}{\Delta^2}$$

$$\Leftrightarrow 2t > \frac{24C^2 \log \frac{2K}{\delta}}{\Delta^2} + \frac{32C^2 \log \left(\log \left(\frac{24C^2}{\Delta^2}\right)\right) + 32C^2 \log(\frac{2K}{\delta})}{\Delta^2}$$

$$\overset{\log(x+y) \leq \log x + \log y, \forall x,y \geq 2}{\Rightarrow} 2t > \frac{24C^2 \log \frac{2K}{\delta}}{\Delta^2} + \frac{32C^2 \log \left(\log \left(\frac{24C^2}{\Delta^2}\right) + \frac{2K}{\delta}\right)}{\Delta^2}$$

$$\Rightarrow 2t > \frac{24C^2 \log \frac{2K}{\delta}}{\Delta^2} + \frac{32C^2 \log \left( \log \left( \frac{24C^2}{\Delta^2} \right) + \log \left( \log \frac{2K}{\delta} \right) \right)}{\Delta^2}$$

$$\Leftrightarrow 2t > \frac{24C^2 \log \frac{2K}{\delta}}{\Delta^2} + \frac{32C^2 \log \log \left( \frac{24C^2 \log \frac{2K}{\delta}}{\Delta^2} \right)}{\Delta^2}$$

$$\overset{\text{Lemma } D.1, \text{ as } 24C^2 \log \frac{2K}{\delta} > e^2}{\Rightarrow} 2t > \frac{24C^2 \log \frac{2K}{\delta}}{\Delta^2} + \frac{16C^2 \log \log (2t)}{\Delta^2}$$

$$\Rightarrow 2t > \frac{8C^2 \log \frac{2K}{\delta} + 16C^2 \log \log_2 e + 16C^2 \log \log(2t)}{\Delta^2}$$

$$\Leftrightarrow t > \frac{4C^2 \log \frac{2K}{\delta} + 8C^2 \log \frac{\log(2t)}{\log 2}}{\Delta^2}$$

$$\Leftrightarrow t > \frac{4C^2 \log \frac{2K}{\delta} + 8C^2 \log(\log_2 2t)}{\Delta^2}$$

$$\Leftrightarrow C \sqrt{\frac{4 \log \frac{2K(\log_2 2t)^2}{\delta}}{t}} < \Delta.$$

$\square$

## D.2. Probability Bound of Good Event

**Lemma D.3** (Adapt Lemma 3 in (Jamieson et al., 2014)). *Denote $\{X_i\}_{i=1}^{+\infty}$ as i.i.d $\sigma^2$-subgaussian random variable with true mean reward $\mu = 0$. For any $\delta \in (0,1)$, we have*

$$\Pr \left( \exists t, |\sum_{s=1}^{t} X_s| \geq \sqrt{2\sigma^2 2^{\lceil \log_2 t \rceil^+} \log \frac{2(\log_2 2^{\lceil \log_2 t \rceil^+})^2}{\delta}} \right) < \frac{\pi^2}{6} \delta.$$

*Or equivalently,*

$$\Pr \left( \exists t, |\sum_{s=1}^{t} X_s| \geq \sqrt{2\sigma^2 2^{\lceil \log_2 t \rceil^+} \log \frac{2(\lceil \log_2 t \rceil^+)^2}{\delta}} \right) < \frac{\pi^2}{6} \delta.$$

Lemma D.3 is fundamentally the same as the lemma 3 in (Jamieson et al., 2014). The only different part is the constant outside the square root. But for simplicity and completeness, we rewrite part of proof and leave it here.

*Proof of Lemma D.3.* Define $u_k = 2^k$, $k \geq 1$. Define $x = \sqrt{2\sigma^2 u_k \log \frac{2(\log_2 u_k)^2}{\delta}}$, $S_t = \sum_{i=1}^{t} X_i$ and the event

$$E_k = \left\{ \max_{1 \leq t \leq u_k} S_t > \sqrt{2\sigma^2 u_k \log \frac{2(\log_2 u_k)^2}{\delta}} \right\} \cup \left\{ \min_{1 \leq t \leq u_k} S_t < -\sqrt{2\sigma^2 u_k \log \frac{2(\log_2 u_k)^2}{\delta}} \right\},$$

For $\lambda > 0$, notice that

$$\mathbb{E} \left[ \exp(\lambda(\sum_{s=1}^{t} X_s)) | X_1, \cdots, X_{t-1} \right]$$

$$= \exp(\lambda(\sum_{s=1}^{t-1} X_s)) \mathbb{E} \exp(\lambda X_t)$$

$$\geq \exp(\lambda(\sum_{s=1}^{t-1} X_s)) \exp(\mathbb{E}\lambda X_t)$$

$$= \exp(\lambda(\sum_{s=1}^{t-1} X_s)).$$

Take $\lambda = \frac{x}{u_k \sigma^2}$, we can conclude $\{\exp(\lambda S_t)\}$ is a submartingale. Then,

$$
\Pr\left(\max_{1 \le t \le u_k} S_t \ge x\right)
$$

$$
= \Pr\left(\max_{1 \le t \le u_k} \exp(\lambda S_t) > \exp(\lambda x)\right)
$$

$$
\overset{*}{\le} \frac{\mathbb{E}\exp(\lambda S_{u_k})}{\exp(\lambda x)}
$$

$$
\le \frac{\exp(\frac{u_k \lambda^2 \sigma^2}{2})}{\exp(\lambda x)}
$$

$$
\overset{\lambda = \frac{x}{u_k \sigma^2}}{=} \exp(-\frac{x^2}{2 u_k \sigma^2}).
$$

Step * is by the maximal inequality for the submartingale. Take $x = \sqrt{2\sigma^2 u_k \log \frac{2(\log_2 u_k)^2}{\delta}}$, we have

$$
\Pr\left(\max_{1 \le t \le u_k} S_t \ge \sqrt{2\sigma^2 u_k \log \frac{2(\log_2 u_k)^2}{\delta}}\right) \le \exp\left(-\log \frac{2(\log_2 u_k)^2}{\delta}\right) = \frac{\delta}{2(\log_2 u_k)^2} = \frac{\delta}{2k^2}
$$

For the part of $\Pr\left(\min_{1 \le t \le u_k} S_t < -x\right)$, the proof is similar. We can conclude $\Pr(E_k) \le \frac{\delta}{k^2}$ and further $\Pr(\cup_{k=1}^{+\infty} E_k) \le \frac{\pi^2 \delta}{6}$.

Thus,

$$
\Pr\left(\exists t, |\sum_{s=1}^{t} X_s| \ge \sqrt{2\sigma^2 \max\{2^{\lceil \log_2 t \rceil}, 2\} \log \frac{2(\log_2 \max\{2^{\lceil \log_2 t \rceil}, 2\})^2}{\delta}}\right)
$$

$$
\le \Pr\left(\exists k, \max_{1 \le t' \le u_k} |\sum_{s=1}^{t'} X_s| \ge \sqrt{2\sigma^2 u_k \log \frac{2(\log_2 u_k)^2}{\delta}}\right)
$$

$$
\le \Pr(\cup_{k=1}^{+\infty} E_k) \le \frac{\pi^2 \delta}{6}.
$$

$\square$

Some Comments are as follows.

- We can similarly prove $\Pr\left(\exists t, |\sum_{s=1}^{t} X_s| \ge \sqrt{2\sigma^2 2^{\lceil \log_2 t \rceil^+} \log \frac{2\pi^2 (\log_2 2^{\lceil \log_2 t \rceil^+})^2}{6\delta}}\right) < \delta$ holds for all $\delta \in (0,1)$.

- Since $\lceil \log_2 t \rceil^+ \le 1 + \log_2 t$, we have

$$
\frac{\pi^2 \delta}{6} \ge \Pr\left(\exists t, |\sum_{s=1}^{t} X_s| \ge \sqrt{2\sigma^2 2^{\max\{\lceil \log_2 t \rceil, 1\}} \log \frac{2(\log_2 2^{\lceil \log_2 t \rceil})^2}{\delta}}\right)
$$

$$
\ge \Pr\left(\exists t, |\sum_{s=1}^{t} X_s| \ge \sqrt{4\sigma^2 t \log \frac{2(\log_2 2t)^2}{\delta}}\right)
$$

# E. Numerical Experiment

## E.1. Settings of Numerical Experiments

The parameter setting of SEE is $\delta_k = \frac{1}{3^k}, \beta_k = 2^k/4, \alpha_k = 5^k, C = 1.01$. In all numerical experiments, all the algorithms in section 6 achieve 100% accuracy in identifying a correct answer of either a qualified arm in a positive instance or outputting None in a negative instance.

We took arm number $K = 10, 20, 30, 40, 50, 100, 150, 200$ and the tolerance level $\delta = 0.001, 0.0001$. Fix $\mu_0 = 0.5, \Delta = 0.15$, we set up 6 instances, by considering different number of arms whose mean rewards are above $\mu_0$. For an arm number $K$, we define (1) AllWorse, whose mean reward vector is $\mu_1 = \mu_2 = \cdots = \mu_K = 0.25$; (2) Unique Qualified, whose mean reward vector is $\mu_1 = \mu_0 + \Delta, \mu_2 = \mu_3 = \cdots = \mu_K = \mu_0$; (3) One Quarter Qualified, whose mean reward vector is $\mu_1 = \mu_2 = \cdots = \mu_{\lfloor K/4 \rfloor} = \mu_0 + \Delta, \mu_{\lfloor K/4 \rfloor + 1} = \mu_{\lfloor K/4 \rfloor + 2} = \cdots = \mu_K = \mu_0$; (4) Half Good, whose mean reward vector is $\mu_1 = \mu_2 = \cdots = \mu_{\lfloor K/2 \rfloor} = \mu_0 + \Delta, \mu_{\lfloor K/2 \rfloor + 1} = \mu_{\lfloor K/2 \rfloor + 2} = \cdots = \mu_K = \mu_0$; (5) All Good, whose mean reward vector is $\mu_1 = \mu_2 = \cdots = \mu_K = \mu_0 + \Delta$ (6) Linear, whose mean reward vector is $\mu_i = \mu_0 - \Delta + \frac{2(i-1)\Delta}{K-1}, 1 \le i \le K$. For instance "AllWorse", the set of correct answer is {None}, while for the other instances which are positive, the answer set $i^*(\nu)$ contains at least one arm. In each experiment setting, we run 1000 independent trials and compute the empirical average of the stopping times.

To avoid an infinite loop in a trial, we set a forced stopping threshold $10^8$ in each instance. All the algorithms, including SEE, HDoC, lilHDoC, LUCB_G, adapted-TaS, adapted-MS, stop in all trials before the total pulling times reach $10^8$.

HDoC and LUCB_G's performance on "All Worse" instances are very similar, leading to two nearly overlapping curves. The radius of each error bar is 3 times the standard error of the empirical stopping times across 1000 repeated trials. The error bars do not appear to be visible, as they are much smaller than the empirical stopping times.

When implementing the algorithm lilHDoC, its originally proposed length of warm-up stage is larger than the total pulling times of some of the benchmark algorithms. To allow comparisons, in our numerical experiment, we only uniformly pull all the arms 200 times for the algorithm lilHDoC.

### E.2. Supplement Figure

Figure 2 compares the trend of empirical stopping times, when the proportion of qualified arms increases in positive instances. For algorithms adapted-TaS and adapted-MS, the empirical stopping times increase as the proportion increases, while for our proposed SEE, the trend is inverse.

### E.3. Correctness of APGAI algorithm

*Table 2.* Number of Failure, APGAI, $\delta = 0.001$

| Instance Type \ K | 10 | 20 | 30 | 40 | 50 |
|---|---|---|---|---|---|
| AllWorse | 0 | 0 | 0 | 0 | 0 |
| Linear | 0 | 0 | 4 | 2 | 1 |
| Unique | 373 | 449 | 462 | 520 | 549 |
| OneQuarter | 152 | 32 | 22 | 7 | 6 |

*Table 3.* Emprical Stopping times $\pm$ 3*Std, APGAI, $\delta = 0.001$

| Instance Type \ K | 10 | 20 | 30 | 40 | 50 |
|---|---|---|---|---|---|
| AllWorse | $5099 \pm 54$ | $10703 \pm 78$ | $16524 \pm 96$ | $22450 \pm 114$ | $28501 \pm 129$ |
| Linear | $11800 \pm 2757$ | $12814 \pm 5496$ | $18145 \pm 11685$ | $11979 \pm 8601$ | $11819 \pm 9723$ |
| Unique | $201716 \pm 22404$ | $485772 \pm 45444$ | $749871 \pm 68136$ | $1096819 \pm 91215$ | $1454700 \pm 112848$ |
| OneQuarter | $90246 \pm 17187$ | $46501 \pm 17532$ | $46890 \pm 22569$ | $25511 \pm 17547$ | $23000 \pm 19224$ |

In compared to other benchmarks, APGAI's numerical performance is significantly different, in the sense that the stopping time it is either very small or very large, which makes its curve of stopping times either much below or much higher than all the others. And sometimes it might get stuck in a non-stopping loop.

We tune some of the numeric experiment setting to see the non-stopping phenomenon more clearly. For APGAI, we set the forced stopping threshold as $50000 * K$, where $K$ is the arm number. Once the total pulling times is no less than the forced stopping threshold, we mark this experiment as a failure, and take the forced stopping threshold as the total pulling times of APGAI. Table 2 records the number of failure for experiments $K = 10, 20, 30, 40, 50$ and instance type "All Worse", "Unique Qualified", "One Quarter", "Linear". The tolerance level is $\delta = 0.001$, with repeating times 1000. Except the

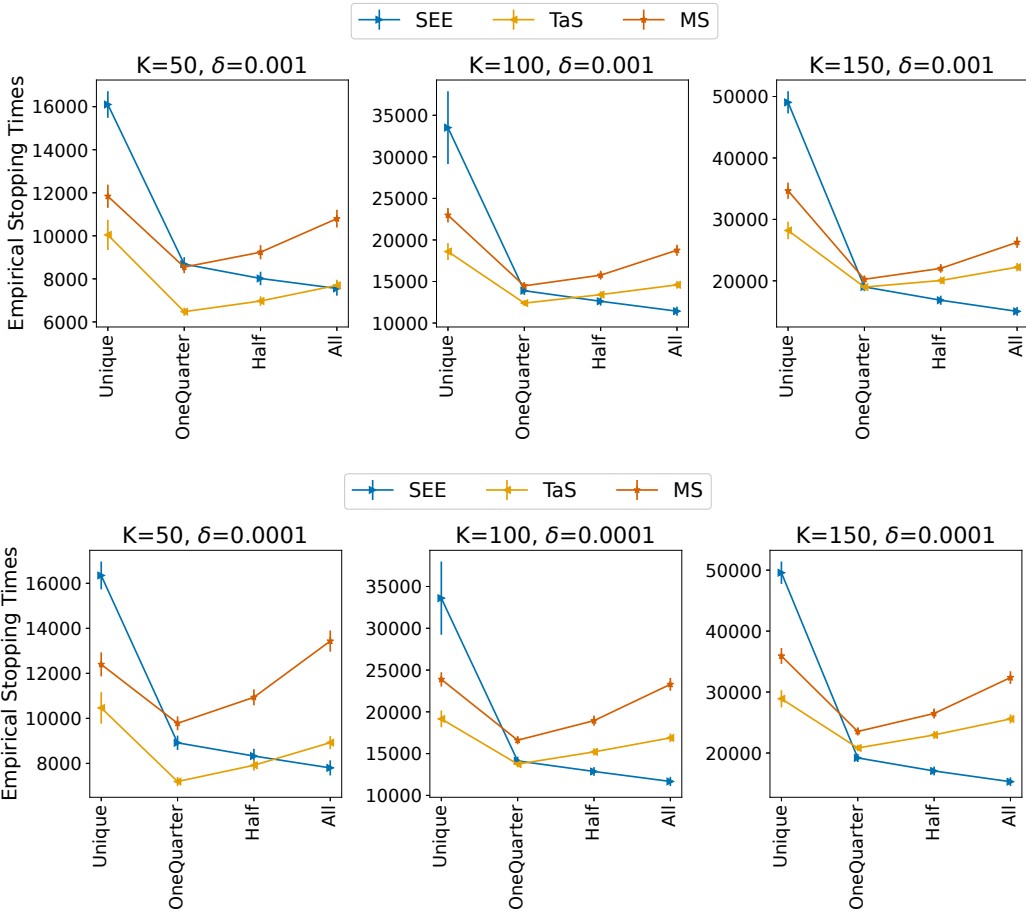

*Figure 2.* Numerical Experiments on SEE and Benchmarks

negative instance "All Worse", failure frequently occurs in some of positive instances. In the "Unique Qualified" group, at least 35% of experiments end up with "failure". While in other groups, failure also occurs. In group "One Quarter", the number of failures decreases as arm number $K$ increases, suggesting APGAI's good performance relies more on the number of qualified arms, instead of the fraction of qualified arms.

The huge number of failure experiment also affects the estimation of the $\mathbb{E}_{\nu,\text{APGAI}}\tau$. Table 3 records the confidence interval for estimating $\mathbb{E}_{\nu,\text{APGAI}}\tau$, ignoring the decimal. In group "All Worse", there aren't any failure, and the empirical mean outperforms all the other benchmarks. In group "Unique Qualified", failure frequently occurs. Both cases result in a relatively small standard error in the numeric experiment. While for the group "Linear" and "One Quarter", the less frequently occurrence of failure greatly increase the standard error of the empirical mean stopping times. In some cases, 3*Std is nearly the same as the empirical mean value. In addition, because of the existence of the forced stopping threshold, the empirical mean we recorded is only a lower bound estimation for the true $\mathbb{E}_{\nu,\text{APGAI}}\tau$. Due to large standard error, we cannot conclude whether the current repeating times is sufficient to approximate $\mathbb{E}_{\nu,\text{APGAI}}\tau$.

Given the time-consuming simulation for APGAI and its significantly different pattern, we do not apply APGAI to instances with larger arm number. And we believe it is unsuitable to compare APGAI with other benchmarks, given the difficulties of estimating $\mathbb{E}_{\nu,\text{APGAI}}\tau$.

