# OpenReview forum: "Near Optimal Non-asymptotic Sample Complexity of 1-Identification"
_ICML.cc/2025/Conference — ICML 2025 poster_

### Official Review · Reviewer_d2eW · 2025-02-27

**Overall Recommendation:** 3

**Summary:**

This paper studies fixed-confidence $1$-identification for sub-Gaussian distributions, where the learner should output one arm whose mean exceed a given threshold if it exists, and output none otherwise. This pure exploration problem with multiple correct answers has been considered under different names in the literature, e.g., Good Arm Identification or any low arm, and with slightly different variants. The authors propose an algorithm named  Sequential-Exploration-Exploitation (SEE), see Algorithm 1. This is a phase-based algorithm that call two subroutines at each phase, an exploration algorithm whose sampling rule is based on UCB indices (Algorithm 2) and an exploitation algorithm that only pulls the candidate answer from Algorithm 2 to verify it is actually a good arm (Algorithm 3). A non-asymptotic upper bound on its expected sample complexity of SEE is given in Theorem 5.3. For positive instance, the upper bound remarkably only features the gap between the mean of the best arm and the threshold. Two lower bounds have been derived on the expected sample complexity for negative instances (Theorem 5.4) and positive instances with only one good arm (Theorem 5.5). They also derived a lower bound on the number of pulls of suboptimal arms for positive instances with only one good arm (Theorem 5.6). The empirical performance of SEE is compared with the ones of several algorithms from the literature on several instances.

**## update after rebuttal**
Including the discussions will improve the paper in its revised version, hence I maintain my positive score.

**Claims And Evidence:**

**Non-asymptotic lower bounds only on instances with a unique correct answer.** Theorem 5.4 holds for negative instances where the only correct answer is None. Theorems 5.5 and 5.6 holds for positive instances with a unique good arm. Therefore, there is no non-asymptotic lower bound when the instance has multiple correct answer, which seems to be restrictive. Could the authors comment on the challenges posed by instances with an arbitrary number of good arms ? Taking two arms as an example would already be valuable. My interest is sparked by the fact that non-asymptotic lower bounds are quite challenging to obtain for a pure exploration problem with multiple correct answers, as being correct on one instance doesn't necessarily imply that we are incorrect on an alternative answer due to non-unique correct answers. For example, in terms of $\log(1/\delta)$ dependency, the only tight lower bound in this case is asymptotic in $\delta \to 0$. As I am less familiar with lower bounds based on the random permutation model, I would appreciate having insights on the limitations of multiple correct answer for this kind of lower bounds.

**Misleading statement on the matching upper and lower bounds on the pulling complexity.** Given the above comment on the fact that there is no lower bound for positive instances with more than one good arm, it is important to reformulate the statement that suggests matching upper and lower bounds for any instances. For example, Line 27-28 “achieve near optimality, in the sense of matching upper and lower bounds on the pulling complexity”  or “our proposed algorithm achieves the non-asymptotic optimality in the sample complexity both the positive case when there is a qualified arm, i.e. an arm with mean reward at least the threshold, and the negative case when there is no qualified arm. We prove matching upper and lower sample complexity bounds, and the gap between these upper and lower bounds is up to a polynomial logarithm factor.”

**Gaussian instances.** To the best of my understanding based on reading the proofs, Theorems 5.5 and 5.6 require the instances to be Gaussian and not solely sub-Gaussian (as in Theorem 5.4). Therefore, this should be added to the statements, or the proof should be adapted. For Theorem 5.5, this comes from Lines 1444-1446. For Theorem 5.6, this comes from lemma C.1 which is used in the proof.

**Non-asymptotic upper bound of Theorem 5.3.** Even though $\gamma$ seems to be an absolute constant, it would be valuable to have a more explicit closed-form constant. This would allow to better understand the impact of each of the hyperparameters used in SEE. Based on a detailed reading of the proofs, it is not possible to clearly extract a formula from the proofs that heavily rely on $O(\cdot)$ notation. Moreover, it also seems like this constant will be very large.

**Non-asymptotic lower bound of Theorem 5.5.** While I understand the need to restrict the parameter $\delta$, it is not clear why the authors restrict themselves to mean parameters in $[0,1]^{K+1}$. Could the authors comment on where they used this condition and why it is important ? This seems unnecessary based on the existing literature proving adopts the random permutation model to prove lower bounds, e.g. Simchowitz et al. (2017, The simulator: Understanding adaptive sampling in the moderate-confidence regime), Chen et al. (2017, Nearly instance optimal sample complexity bounds for top-k arm selection), Al Marjani et al. (2022, On the complexity of all $\epsilon$-best arms identification) or Poiani et al. (2024, Best-Arm Identification in Unimodal Bandits).

**Non-asymptotic lower bound of Theorem 5.6.** Similar question, namely why the boundedness of the mean parameters is important. In that case, the condition $\delta< e^{-8}\approx 3. 10^{4}$ starts to become restrictive for practical application. Could the authors comment on whether finer analysis would allow to get a weaker condition on $\delta$.

**Essential References Not Discussed:**

To the best of my understanding, the authors totally omitted one key related work, namely Tsai et al. (2024, “lil’HDoC: An Algorithm for Good Arm Identification Under Small Threshold Gap”). This paper proposes lil’HDoC, which builds on HDoC by using the law of iterative logarithm. Similarly, as for HDoC, lil’HDoC is designed to sequentially return all the arms above a given threshold. It is $\delta$-correct for this task (Theorem 1) and enjoys guarantees on the sample complexity to return the $k$-th best arm above the threshold (Theorem 3). Using their Theorem 3 for the best arm of a positive instance yields a high probability upper bound on the sample complexity scaling as $O(\Delta_{0,1}^{-2} \log(\log(\Delta_{0,1}^{-2}/\delta)/\delta))$. While being guarantees in high-probability, it seems reasonable to expect that lil’HDoC is also $(\Delta, \delta)$-PAC. Moreover, given the improved performance of lil’HDoC compared to HDoC, it is quite important to compare it with SEE.

**Experimental Designs Or Analyses:**

For varying dimension ($K \in \{10,20,30,40,50\}$) and confidence level ($\delta \in \{10^{-k}\}_{k \in \{2,3,4\}}$), the experiments consider multiple instances with and varying that address specific cases of interests: one negative instance (AllWorse) and several positive instances. The positive instances have either two groups of arms and varying number of good arms (Unique versus OneQuarter), or have linearly decreasing mean (Linear). The experiments are repeated $1000$ times and errors bars are shown.

While the experimental setup is convincing, three important benchmarks are missing: Track-and-Stop, Murphy Sampling and lil’HDoC. Updating the plots with those benchmarks will go a long way to provide satisfying empirical evidence on the performance of SEE compared to existing algorithms.  This is important and possible during the author-reviewer discussion, at least to have preliminary results.
- Track-and-Stop (TaS) algorithm from Garivier and Kaufmann (2016) adapter to GAI. Namely, on top of forced exploration and based on C-Tracking, the optimal allocation is the Dirac distribution on the empirical best arm when the empirical best mean exceeds the threshold, and it is inversely proportional to the inverse square mean gap to the threshold otherwise.  A more detailed description exists in Appendix I.2.3 of Jourdan and Reda (2023). As TaS is a reference algorithm for fixed-confidence pure exploration problems, it is important to include it. Appendix I.5 of Jourdan and Reda (2023) suggests that it performs well.
- Murphy Sampling (MS) algorithm from Kaufmann et al. (2018). The improved stopping rule of MS is tailored to a slightly different pure exploration problem (Section 5), i.e., return whether the instance is positive without returning a good arm. However, the sampling rule of MS can be used for GAI when combined with the  recommendation/stopping rule in Section 4 of Jourdan and Reda (2023). Appendix I.5 of Jourdan and Reda (2023) suggests that MS performs well.
- lil’HDoC algorithm from Tsai et al. (2024, “lil’HDoC: An Algorithm for Good Arm Identification Under Small Threshold Gap”). Since it numerically outperforms HDoC, it is important to understand whether SEE outperforms it. See “Relation To Broader Scientific Literature” for more details.

**Methods And Evaluation Criteria:**

See “Experimental Designs Or Analyses” section for details on the empirical evaluation.

**Other Comments Or Suggestions:**

- Theorem 5.4 could be stated more precisely instead of using the $\Omega(\cdot)$ notation. The authors show that $\mathbb E[\tau] \ge \log(1/(2.4 \delta)) H_{1}^{neg}$.

- Theorem 5.5 could be stated more precisely instead of using the $\Omega(\cdot)$ notation. The authors actually have a result that is not too difficult to state in details.

- End of page 12 to beginning of page 13. The $\delta_{k-1}$ and $\alpha_{k-1}$ should not be moved outside the sum over $k$.

- Line 617. $\beta_k = 2^k$.

- Lines 618-621. There seems to be an error in the block equation. Since $3^{\log_2(x)} = x^{\alpha}$ where $\alpha = \ln(3)/\ln(2) >1$, the authors should obtain that $1/\delta_{L’} = O((H_{1}^{pos}/K)^\alpha)$ and $1/\delta_{L’’} = O(\Delta_{0,1}^{-2\alpha})$. Instead, they claim that the same upper bound holds with $\alpha=1$. This doesn’t seem to change the main result, as only $log(1/\delta_{k})$ terms appear.

- Lines 702-705. The first $\delta$ should be a $\delta_{k-1}$ and the last inequality should be $\log(K/\delta)$.

**Other Strengths And Weaknesses:**

**Lack of conclusion.** The paper cruelly lacks a conclusion that summarizes the contributions and discusses open problems. Could the author write a sketch of the conclusion that they would add in a potential camera-ready version ? Even in the submitted version, some spaces should have been saved to write a proper conclusion. For example, one could put in Appendices some plots showing the impact of $\delta$ for all the instances in Section 6.

**Lack of algorithmic simplicity.** It’s worth noting that SEE is quite convoluted algorithmically speaking, and some components exist only for the sake of analysis.
First, this is a phased-based algorithm which calls two distinct subroutines at each phase (Algorithms 2 and 3) that do not share samples. This seems to be wasteful. Could the authors elaborate on whether the lack of sharing between observations is only to facilitate the analysis, or whether it is rooted in more fundamental reasons ?
Second, the authors mention that the temporary container $Q$ exists only to “facilitate our theoretical analysis”. Could the authors discuss the challenges that arise from simplifying the algorithm by removing the temporary container $Q$ ? It would be great to actually remove this altogether.
Third, there are four hyperparameters with arguably “subjective” choice of default values, i.e., $C$ and $(\alpha_{k}, \beta_{k}, \delta_{k})$.
Fourth, it is unclear to me what is the intuition behind the exploration and exploitation horizons $(T_{k}^{ee}, T_{k}^{et})$, which are very large. For example, with default hyperparameters values, we obtain $T^{ee}_{1}/K \approx 20.000$. Therefore, the first exploration horizon is actually larger than the empirical stopping time in the experiments of Section 6.
Given the simplicity of the GAI setting, it seems unsatisfactory to solve it with a convoluted approach.

**Questions For Authors:**

1. Could the authors discuss in more details what are the differences between their sampling rule and stopping rule and the ones used previously in the literature, e.g., HDoC, LUCB-G, APGAI and lil’HDoC.

2. To the best of my understanding, the sampling rule used in Algorithm 2 seems to bear similarity with lil’UCB from Jamieson et al. (2014). Could the author highlight the differences ?

3. Could the authors compare the lower boundof Theorem 5.6 with Proposition 2 in Kaufmann et al. (2018) ? Both seems to say something on the expected number of selection of suboptimal arms.

Several other questions have been asked in the previous sections.

**Relation To Broader Scientific Literature:**

The authors should rephrase the following false (or at least misleading) statement on the literature (L 147-150). “HDoC and APGAI are both $\delta$−PAC algorithm, but they are not $(\Delta, \delta)$-PAC, as they both suffer from the infinite complexity issue, i.e. the upper bound of $\mathbb E[\tau] = + \infty$ if there exists an arm $a$ whose $\mu_a = \mu_0$”. While the known upper bound on the expected sample complexity of both algorithms tends to infinity, it doesn’t imply that the expected sample complexity goes to infinity. In order to prove such a statement, one would need to have a lower bound on the expected sample complexity that tends to infinity when there exists an arm $a$ whose $\mu_a = \mu_0$. Therefore, the current known results don’t imply that  HDoC and APGAI are not $(\Delta, \delta)$-PAC. While it seems legitimate to conjecture that APGAI might not be $(\Delta, \delta)$-PAC based on the empirical evidence, HDoC might still be $(\Delta, \delta)$-PAC.

**Theoretical Claims:**

I checked the correctness of the theoretical results. The depth of my proofreading is detailed in the question on “Supplementary Material”. I highlighted some typos or minor errors in the question “Other Comments Or Suggestions”.

---

> ### Author Rebuttal · Authors · 2025-03-29
>
> Thank you very much for your suggestions, and we will correct the typos in the revision. Due to length limit, we can only answer part of the questions in comments. We are looking forward to discuss more in the next iterations.
>
> 1. "no non-asymptotic lower bound when the instance has multiple correct answer"
>     + Please check the 1st and 2nd point in the response to the reviewer M2Aq.
>     + If there are multiple qualified arms, our upper bound is nearly optimal regarding the $\delta$-dependent part, but the $\delta$-independent part is loose.
>     + Despite that, **none of the existing algorithms can achieve same theoretical guarantee as us**.
>
> 2. "Motivation of temporary container Q and complicated alg design"
>
>     Please check the 3rd point in our response to Reviewer eELW.
>
> 3. "Theorems 5.5 and 5.6 require the instances to be Gaussian and not solely sub-Gaussian"
>
>     + Gaussian dist with variance $\leq 1$ is 1-subgaussian. Theorems 5.2, 5.3 work for all subgaussian instances, while Lower bound(Theorems 5.5, 5.6) show it is nearly tight in a sub-class of the subgaussian dist.
>     + It is generally impossible to expect the lower bound works for all 1-subgaussian dist. For example, if all the reward are deterministic, an algorithm pulling all the arms once can be $\delta$-PAC with sample complexity $K$.
>
> 4. "differences of sampling and stopping rules among HDoC, LUCB-G, APGAI, lil’HDoC, SEE"
>
>     Please check the last part in our reply to the reviewer M2Aq. The main difference between ours and existing work is
>     + we adopt increasing radius of the confidence interval.
>     + **we do not eliminate any arm.**
>
> 5. "compare the lower bound of Theorem 5.6 with Proposition 2 in Kaufmann et al. (2018)"
>
>     Summary of result:
>     + Sum up the lower bound in our theorem 5.6, we get $\mathbb{E}\tau\geq \Omega(\sum_{a=2}^K\frac{\log 1/\Delta_{1,0}^2 }{\Delta_{1,a}^2})$, meaning the term $\log 1/\Delta_{1,0}^2$ in the Theorem 5.3 might be required.
>     + In Kaufmann et al. (2018), they derive the lower bound $\Omega(\frac{1-K^3\delta}{K\max\\{\Delta_{1,0}^2, \Delta_{0,K}^2\\} })$
>
>     We have $\log 1/\Delta_{1,0}^2$ in the numerator and no K in the denominator, suggesting that our bound is stronger. But we require more assumptions.
>     + Only $\mu\_1$ is above $\mu\_0$
>     + $\mu\_1-\mu\_0$ is sufficiently small
>     + we are required to output an arm while they do not
>
> 6. "Discussions on lilHDoC"
>     + Synergizing HDoC with the lil rule, and their framework requires a warm up stage
>     + Derive an extra high-probability upper bound for each $N\_a(\tau)$,
>
>        $\Pr(N\_a(\tau) < O(\frac{ \log\frac{K}{\delta}+\log\log\frac{K}{\delta\Delta_{0,a}^2} }{\Delta_{0,a}^2})) > 1-\delta$
>       + in the case $\Delta_{0,a}=0$, this upper bound is infinity
>       + still suboptimal as $\log\log\frac{1}{\delta}$ exists, and the upper bound is infinity for some $\nu\in \mathcal{S}^{pos}\cup \mathcal{S}^{neg}$
>     + We will include this discussion in our literature review.
>
> 7. "lilHDoC, HDoC and APGAI might be $(\Delta, \delta)$-PAC algorithm"
>
>    Here we rigorously prove lilHDoC is not $(\Delta, \delta)$-PAC. Consider a two-arm instance with $\mu_1>\mu_0=\mu_2$. Arm 1 follows unit variance Gaussian and arm 2 returns constant reward.
>    + With non-zero prob p, $\text{UCB}_1(t) < \mu_0$ holds after the warm up stage.
>    + Then arm 1 will get removed from the arm set. In this case, lilHDoC will keep pulling arm 2 without termination
>    + In an event with positive prob, $\tau=+\infty$. We can conclude $\mathbb{E}\tau=+\infty$
>
>    Similar idea applies to APGAI, HDoC. They are not $(\Delta, \delta)$-PAC.
>
> 8. "Compare with TaS, MS, lilHDoC in numeric part"
>
>     While we implement lilHDoC and an adapted version of TaS, the algorithm MS is incompatible with our model
>     + TaS is not designed for 1-identification. The only available stopping condition is provided in Theorem 10, Degenne & Koolen 2019 (S-TaS), which **doesn't provide an explicit value for "large enough C"**
>
>        We use the pulling rule in TaS, while adopt the stopping rule of S-TaS, using a lower bound on required C. **Equivalently, we remove the "Sticky" part in S-TaS**.
>     + lilHDoC's length of warm up stage is already much larger than the SEE in our numerical settings.
>
>        We set T=200 as the warm up pulling times for each arm.
>     +  MS only determines if an instance is positive or negative, but does not identify a good arm in the former case. MS does not solve the GAI problem, unlike SEE which both solve GAI and identify if an instance is positive or negative. Thus, MS and SEE cannot be compared on the same ground.
>
>     Point 5 for reviewer K8WR shows the results.
>
> References
>
> + Jamieson et.al 2014, lil’ UCB : An Optimal Exploration Algorithm for Multi-Armed Bandits
> + Kaufmann et al. 2018, Sequential Test for the Lowest Mean: From Thompson to Murphy Sampling
> + Degenne & Koolen et al. 2019, Pure Exploration with Multiple Correct Answers

---

> > ### Comment · Reviewer_d2eW · 2025-04-02
> >
> > I thank the authors for their thorough and detailed answers, as well as the additional experiments. At the time being, I will keep my positive score.
> >
> > For the sake of discussion, I detailed some follow-up comments. Feel free to use the extra space to add comments on questions not previously addressed due to space limit.
> >
> > **1**.  In the revised version, it would be great to include a more detailed discussion on this lower bound for instances with multiple good arms. This would allow substantiating and nuancing the currently misleading claims on matching upper and lower bounds for all instances.
> > Does the sum really involve the term $\Delta_{1,m}$ that is independent of $a$ ? Given the gap in the non-asymptotic dependency of the lower/upper bound for those instances, do the authors have an intuition as regards whether the lower and/or the upper bound could be improved ?
> >
> > **3**. I don’t expect the lower bound to hold for all sub-Gaussian instances. Yet, the Gaussian assumption should still be explicitly mentioned in the Theorems 5.5 and 5.6.
> >
> > **8**. We thank the authors for including lilHDoC in their experiments. Here are two clarifications on statements made for TaS and MS.
> > - **TaS**. The sentence “The only available stopping condition is provided in Theorem 10” is false. You can use the stopping rule from Jourdan and Reda (2023), see equation (5) in Section 4. Their Lemma 2 ensures that it is $\delta$-correct for sub-Gaussian for any sampling rule. Using a loose upper bound on $C$ probably explains why TaS has poor performance in the provided additional experiments, which are not consistent with the empirical results in Appendix I.5 of Jourdan and Reda (2023).
> > - **MS**. From an empirical perspective, the sentence “MS and SEE cannot be compared on the same ground” is misleading. As detailed in my initial review, MS has been adapted for GAI with good empirical performance, see Appendix I.5 of Jourdan and Reda (2023). From a theoretical perspective, this modified MS and SEE cannot be compared yet.
> >
> > **Miscellaneous**.
> > On the “linear” instance, the mean of APGAI is lower than the one of SEE. Is bold used to highlight the one with the lowest mean + std ?

---

> > > ### Author Response · Authors · 2025-04-07
> > >
> > > Thanks for your careful and detailed reply. Firstly we want to answer some missing questions in our first rebuttal.
> > >
> > > 1. "why restrict $\mu\_a\in [0, 1]$"
> > >    + About the upper bound, $\mu\_a\in [0, 1]$ guarantees $\Delta\_{i,j} < 1$, further $\log 1/\Delta\_{i,j}^2 >0$. If $\Delta\_{i,j}>1$, we need to turn to $\lceil ( \log \lceil 1/\Delta\_{i,j}^2 \rceil^+)\rceil^+,\sum\_a \lceil 1/\Delta\_{1,a}^2 \rceil^+$, which is strenuous, but the analysis still holds.
> > >    + About the lower bound, $\mu\_a\in [0, 1]$ is required. We need to construcut "hard" enough problem instance to show any algorithm must suffer some complexity.
> > >
> > > 2. "“subjective” choice of default values, i.e., $C,\beta\_k,\alpha\_k, \delta\_k$"
> > >
> > >    We analyze the requirements of $C,\beta\_k,\alpha\_k, \delta\_k$ in appendix B.1.
> > >    + Exponential decreasing/increasing speed for $\delta\_k, \beta\_k$ is required in our theoretical analysis.
> > >    + Taking $\beta\_k=2^k,\alpha\_k=5^k, \delta\_k=1/3^k$ seems to be regular choices.
> > >
> > > 3. "why $T\_k^{ee}$ so large"
> > >    + Recall defition $T\_k^{ee}=1000(C+1)^2K\beta\_k\log(4K/\delta\_k)$.
> > >    + The constant 1000 is mainly for **simplifying the calculation in Line 809-871, which is for the upper bound for** $L^{pos}\_{ee}, L^{pos}\_{et}, L^{neg}$. 1000 is large enough such that solving those inequalities becomes easier.
> > >    + If we remove the coefficient 1000, it will only affects the constant factor in our upper bound, and the theoretical analysis still holds.
> > >
> > > 4. "No sharing sample between ee and et, wasteful"
> > >
> > >    We admit it is possible to further improve our current result by merging the samples collected in exploration and exploitation periods.
> > >    + Our current design is to **simplify the proof**, as it is easier to treat $\kappa^{ee}, \kappa^{et}$ independently.
> > >    + If we merge these two phases, we may need to distinguish the smaller one between $\delta / \alpha\_{\kappa^{et}}, \delta\_{\kappa^{ee}}$, which makes the current analysis more complicated
> > >
> > > 5. On the “linear” instance, the mean of APGAI is lower
> > >
> > >    The mean stopping time of APGAI is indeed smaller in the group Linear. But we want to clarify our result only estimates its lower bound of empirical mean. Larger forced stopping threshold results in larger value. We will elaborate more on APGAI's result in our revision
> > >
> > > We will also follow your suggestions about discussing Gaussian assumption, and the the lower bound if multiple good arms exist. Now, we turn to discuss TaS and MS.
> > >
> > > We acknowledge your reply and we indeed miss the numeric performance of two combinations TaS+GLR, MS+GLR, where GLR is a stopping rule in Lemma 2, Jourdan & Reda (2023). Here we clarify the following facts.
> > > + GLR stopping rule guarantees TaS+GLR, MS+GLR are both $\delta$-PAC. But unlike other benchmarks, there aren't theoretical analysis regarding $\tau$. Applying the pulling rules in these two algorithm is based on heuristic experience
> > > + Given current numeric result, our proposed **SEE is the best in all the algorithms with performance guarantee on $\tau$**
> > > + According to numeric result in Jourdan & Reda (2023), we admit it is possible these two will significantly outperform SEE in positive instances where K is not large. But we fail to rerun all the numeric experiments for these two algorithms because of the time limit
> > >
> > > Here we present two groups of experiment, to discuss the pros and cons of TaS+GLR, MS+GLR. Take mu0=0.5, repeating times 1000, delta=0.001, i.i.d noise N(0, 1)
> > > + Linear, the reward vector is an arithmetic array with mu1=0.3, muK=0.7.
> > >
> > >    The gap is larger comapred to Linear Group in current submission, as the time limit requires to shorten the experiments
> > > + AllBetter, mu1=...=muK=0.7
> > >
> > > For group Linear,
> > > | Method         | K=50          | K=100              | K=150        | K=200             |
> > > | -------------- | ------------------ | ------------------- | ------------------ | ------------------ |
> > > |SEE(This work)|4323$\pm$91|**6539$\pm$110**|**8960$\pm$139**|**10582$\pm$141**|
> > > |TaS+GLR|3881$\pm$39|7528$\pm$69|11632$\pm$107|15403$\pm$142|
> > > |MS+GLR|**3851$\pm$45**|6680$\pm$72|9280$\pm$95|11839$\pm$117|
> > >
> > > Some observation
> > > + TaS+GLR, MS+GLR perform pretty well when K=50
> > > + When K gets larger, TaS+GLR, MS+GLR becomes worse than SEE
> > >
> > > The phenomenon is clearer in the instance AllBetter, in which TaS+GLR, MS+GLR don't perform well
> > > | Method         | K=50          | K=100              | K=150        | K=200             |
> > > | -------------- | ------------------ | ------------------- | ------------------ | ------------------ |
> > > |SEE(This work) |**4077$\pm$60**|**6198$\pm$100**|**7963$\pm$131**|**9988$\pm$167**|
> > > |TaS+GLR|4854$\pm$54|9422$\pm$94|14309$\pm$142|19211$\pm$192|
> > > |MS+GLR|6103$\pm$78|10609$\pm$125|14615$\pm$165|18519$\pm$201|
> > >
> > > In our revision, we will set up more experiments regarding **arm number, reward vector** and release the code for numeric experiments. And we still want to emphasize **our main contribution still lies on the new theoretical analysis.**

---

### Official Review · Reviewer_K8WR · 2025-03-08

**Overall Recommendation:** 3

**Summary:**

This paper studies the 1-identification problem, a multi-armed bandit exploration problem with the goal of identifying an arm whose mean exceeds a given threshold. The paper introduces a new algorithm that achieves near-optimal non-asymptotic sample complexity. Theoretical guarantees establish its efficiency in both positive and negative instances. The authors also conduct numerical experiments that demonstrate SEE outperforms some baseline algorirthms.

**Claims And Evidence:**

I find it concerning SEE does not outperform all other baselines, even in the synthetic experiments. Based on the evidence provided by the authors, the APGAI method performs overall better, hence the limited impact of this work.

**Essential References Not Discussed:**

N/A

**Experimental Designs Or Analyses:**

The experimental design of the synthetic benchmark described in Section 6 and Appendix E.1 seems sound.

**Methods And Evaluation Criteria:**

The method was only evaluated on the synthetic benchmarks. Evaluating it on real-world datasets, e.g., Yahoo! Today Module User Click Log, would strengthen the results.

**Other Comments Or Suggestions:**

I believe the paper could be written much more clearly. For example, (the list is not exhaustive)
* Abstract begins with: _Motivated by an open direction in existing literature_ - all papers are
* L015: _or to output None_ - this is clear only to people using Python
* L104: _It is obvious to see (∆, δ)-PAC is stronger than δ-PAC._ - you should be more precise in what stronger means
* Page 4: All three algorithms took too much time to study, and their presentation could be simplified with a diagram
* The paper could use a Conclusions section summarizing the main takeaways and discussing future work

**Other Strengths And Weaknesses:**

See other comments.

**Questions For Authors:**

Can you evaluate your method and the baselines on a real-world dataset and share the results?

**Relation To Broader Scientific Literature:**

I am not familiar with the literature in the 1-identification domain, but based on these empirical results, the method seems insignificant.

**Theoretical Claims:**

I did not check the proofs.

---

> ### Author Rebuttal · Authors · 2025-03-29
>
> Our primary contributions lie in the theory towards optimal performance on the 1 identification problem, and there could be misunderstanding in the numerical performance of AP-GAI. We are thankful for the perspective that motivates us to clarify more, and we look forward to discuss more in the next iteration. Let us provide the following clarifications:
>
> 1. As discussed in the reviewing process, the main contribution is **we design a new algorithm with best performance guarantee**, compared to all the existing algorithms.
>
> 2. Regarding numerical performance
>     + HDoC's curve is slightly below (within 3 standard error) SEE in the group "Unqiue, $K=10$" . In other groups, its curves are all above SEE.
>     + LUCB\_G's curves are all above SEE.
>     + APGAI's curves are below SEE in group "AllWorse" and "Linear, $K\geq 30$", Other are all above SEE.
>
> 3. The numerical performance is of AP-GAI has to be interpreted carefully, different from other algorithms.
>
>     + It is generally impossible for an algorithm to outperform all the others across all the scenarios.
>
>        For example, a naive algorithm that keeps pulling arm 1 can defeat all the existing algorithms, if the optimal arm is exactly arm 1.
>
>    + In the plots in section 6, our proposed SEE outperforms all other benchmarks except AP-GAI. It appears from the figure that APGAI is can outperform our proposed SEE in some of the plots
>
>        +  However, as stated in the section 6 and the appendix E, the plotted points in the Figure are only **lower bounds of the  the realized empirical stopping time $\tau$ of AP-GAI**, thus the plotted lines for AP-GAI only serves as a lower bound but not the actual performance of AP-GAI.
>
>        +  In the "Unique" experiment group, AP-GAI always fails to terminate even after $8000\cdot K$ rounds, making its performance much worse than all the others. In our evaluation, AP-GAI is the only algo that suffers from the non terminating issue.
>
>        + In other groups,  with some small but non-zero probability (Quarter 14/1000-52/1000, Linear 1/1000-3/1000), AP GAI stuck into non-stopping pulling procedure and we have to terminate it before the real termination. This results in the large error bar in the graph.
>
>        + Considering stability on overall performance, we feel that SEE is better than AP-GAI, since non terminating is a serious issue, and effectively means that the expected stopping time can be infinite. In our revision, we will emphasize the non terminating issue for clarification.
>
> 4. The notations and definitions ("None", delta-PAC, ...) follow the convention in existing research, such as Kano et.al 2017, Degenne & Koolen 2019. We thank the reviewers for suggestion in improving the intuition about SEE. While the complicated algorithm design is necessary, we plan to provide more intuitions in our revision.
>
> 5. Additional numerical experiments
>
> Following the reviews, we implement
> + two extra benchmarks Adapted-TaS(A-TaS) and lilHDoC (see 8th point in our reply to reviewer d2eW)
>
>    Due to the time and space limit, we only present the result in K=30, $\delta=0.001$, with 1000 independent repeating times
> + an extra RealLfie instance in appendix I.1.1 of Jourdan & Reda 2023, **whose mean reward vector comes from real-world therapeutic protocols data**, $\delta=0.001$, with 1000 independent repeating times
>
> Following table reports the empirical stopping times. We ignore the decimal number.
>
> | Method       | All Worse | Unique | One Quarter | Linear | RealLife(Extra) |
> |---------------|--------------|----------|----------|----------|------|
> | SEE(This work)  | 19885 $\pm$ 60 | **11029 $\pm$ 180** | **7106 $\pm$ 154** | **9266 $\pm$ 150** | **281 $\pm$ 2** |
> | HDoC | 23601 $\pm$ 42 | 12023 $\pm$ 122 | 11313 $\pm$ 108 | 13620 $\pm$ 143 | 472 $\pm$ 3 |
> | LUCB\_G | 23601 $\pm$ 42 | 18401 $\pm$ 179 | 15114 $\pm$ 136 | 18234 $\pm$ 176 | 395 $\pm$ 3 |
> | APGAI | **16496 $\pm$ 33** | 143266 $\pm$ 3548 | 14716 $\pm$ 1504 | 8762 $\pm$ 1262 | 1568 $\pm$ 571 |
> | A-TaS(Extra) | 280904 $\pm$ 377 | 32111 $\pm$ 180 | 37393 $\pm$ 277 | 38562 $\pm$ 302 | 1766 $\pm$ 8  |
> | lilHDoC(Extra) | 37099 $\pm$ 50 | 14930 $\pm$ 108 | 17542 $\pm$ 125 | 19363 $\pm$ 160 | 3610 $\pm$ 0 |
>
> SEE ranks 2nd in the "Allworse" group and is the best in all the other groups. The extra numeric experiments also suggest the superior numerical performance of our proposed SEE.
>
> References
> + Kano et.al 2017, Good arm identification via bandit feedback
> + Degenne & Koolen 2019, Pure Exploration with Multiple Correct Answers
> + Jourdan & Reda 2023, An Anytime Algorithm for Good Arm Identification

---

> > ### Comment · Reviewer_K8WR · 2025-04-03
> >
> > I want to thank the authors for all the clarifications. I also read other reviews and it is more clear to me where the paper contributions lie. I also commend another empirical evaluation in such a short time. I updated my score accordingly.

---

> > > ### Author Response · Authors · 2025-04-07
> > >
> > > Thanks for your additional effort in the evaluation and the appreciation to our contributions. We will improve our manuscript based on your suggestion.

---

### Official Review · Reviewer_eELW · 2025-03-11

**Overall Recommendation:** 4

**Summary:**

This paper addresses the problem of 1-identification in stochastic multi armed bandits. In particular, given a reward threshold $\mu_{0}$ an algorithm solving the 1-identification problem has to return an arm whose associated expected reward is greater than $\mu_{0}$ whether it exists. The authors propose the fixed confidence SEE (Sequential-Explore-Exploit) method, exhibiting a non-asymptotic sample complexity. Additionally, the authors propose a lower bound for the sample complexity in the considered setting, showing that their proposal matches (up to logarithmic factors) the lower bound, thus being near-optimal. Finally, the authors validate their method against baselines, evaluating the empirical stopping time of the methods.

## Update after rebuttal: I believe that including the comments provided by the authors during the rebuttal will enhance the paper. I maintain my positive score.

**Claims And Evidence:**

The claims made by the authors are supported by both theoretical and experimental evidence.

**Essential References Not Discussed:**

The related works section is complete and enriched by additional discussion in the appendix. The authors provide the reader with a complete overview of the contribution of the recent literature. I feel (up to my knowledge) all the essential references are discussed properly in this work.

**Experimental Designs Or Analyses:**

I checked the validity and the soundness of the reported experiments. The only issue I feel to highlight is in the fact that in the experimental section the authors state that they made 1000 runs of the same experiment, but the reported plots do not show the confidence intervals. I suggest to show also the confidence intervals of the presented curves, in order to assess the statistical significance of the results.

**Methods And Evaluation Criteria:**

The methods and the evaluation criteria are appropriate for validating the claims done by the authors.

**Other Comments Or Suggestions:**

Typo:
1. line 133 (right)
2. line 149 (right)
3. line 182 (left)
4. line 262 (left)

**Other Strengths And Weaknesses:**

**Strengths**
This paper makes a step towards the closure of the 1-identification problem for stochastic multi armed bandits. The paper is complete in the sense that the authors not only provide a novel method and its theoretical analysis, but also they provide the lower bound for the setting allowing for checking the optimality of the proposed method (and of other predecessors). Additionally, I appreciated the rich related works section and the parallel done with the best arm identification, motivating why it is not convenient to just apply a BAI method. Finally, I have appreciated the presence of the sketches of the proofs.

**Weaknesses**
1. The paper is well-written and easy to follow until Section 4, where the algorithm is explained. Indeed, the presented pseudo-codes are a little bit hard to follow, even if in the following part it is explained. I suggest to insert in the main paper a more conceptual version of the pseudo-code, leaving the current proposed version in the Appendix.
2. This excess of complexity is a feeling that remains even for the section about the main theoretical results. I suggest just to report the core results, without the presented level of details in the main paper, for instance for what concern lines 349--364 (left).
3. At line 304 (left) the authors cite a lemma in appendix, I suggest at least to say what this lemma is about in the context of the main paper.
4. Besides the comment that I have previously posted on the experimental validation, and even if I think the amount of experiments is adequate for a theoretical paper, I suggest the author to compare their method to some algorithm thought to address best arm identification in order to further highlight the inefficiency of such methods in this setting.
5. Minors: lack of running title; lack of text in Lemma 5.1; SEE in the abstract is expanded in Sequential-Exploration-Exploitation but in the following it is expanded in Sequential-Explore-Exploit; Lemma 4.2 needs to be restated in my opinion.

**Questions For Authors:**

See previous sections.

**Relation To Broader Scientific Literature:**

The authors propose, as they highlight and up to my knowledge, the first method solving the 1-identification problem providing non-asymptotic expected stopping time (in the fixed confidence setting). Moreover, the authors seems to provide the first lower bound for the sample complexity in this setting and they show their method to be nearly optimal.

**Theoretical Claims:**

I briefly went through the proofs, and the theoretical claims seems to be sound.

---

> ### Author Rebuttal · Authors · 2025-03-28
>
> Thank you very much for your suggestions. We will correct the typos, and we look forward to discuss more in the next iteration. We will also remove part of the sketch proof to allow space for algorithm description.
>
> 1. "the reported plots do not show the confidence intervals"
>
>     In fact, all figures indeed include the error bars, but many of these error bars are too small to be visible, especially in the experiment group "All Worse". Please check the notebook Visualize-Demo-Delta\_0p15.ipynb in the supplement files for details.
>
> 2. "compare their method to some algorithm thought to address best arm identification"
>
>     + LUCB\_G algorithm, one of the benchmarks, can be considered **an adaptation of BAI algorithm UCB** in Jamieson \& Nowak 2014.
>
>     + In fact, a BAI algorithm without adapted stopping rule would fail to terminate with a qualified arm in the 2 arm instance $\mu_1=\mu_2>\mu_0$. Thus, we only focus on BAI benchmark with adaptation like the previous point.
>
> 3. (Also for d2eW) "The paper is well-written and easy to follow until Section 4,"
>
>     We will try our best to provide more intuitions on SEE in our revision. The complicated design seems to be needed for our current algorithm framework. In the following, we will show a simpler version, and discuss why this version fails to work. Then, we explain why we adopt the current design. An **informal and simpler SEE** is as follows. (We only consider positive instances)
>
> --alg starts--
>
> For phase index $k=1,2,\cdots$
> +  (Exploration) Run LUCB\_G with previous exploration history and tolerance level $\delta_k$. Stops when
>      +  LUCB\_G stops and return $\hat{a}_k$, or
>      +  The total pulling times in all exploration phase $\geq T_k^{\text{ee}}$
> +  (Exploitation) If $\hat{a}_k\in [K]$, keeps pulling $\hat{a}_k$. Stops when
>      +  $\text{LCB}_{\hat{a}\_k}^{et}(\delta) > \mu_0$, or
>      +  Total pulling times of $\hat{a}_k$ $\geq T_k^{\text{et}}$
>
> --alg ends--
>
> where LUCB\_G is in Kano et.al 2017, very similar to the alg UCB in BAI area. We wish at the phase $k$ such that $k\geq \max\{\kappa^{ee}, \kappa^{et}\}$ (i.e concentration inequality (3) holds), $\beta_k\geq \Omega(1/(\omega^2 \Delta_{1,0}^2))$, we have
> + the LUCB\_G algorithm can guarantee $\mu_{\hat{a}\_k}\geq \omega \mu_1+(1-\omega)\mu_0$.
> + Then the above algorithm terminates at the end of phase k.
>
> To achieve this, we need to guarantee **at the start of an exploration phase k,**   $LCB\_{a}^{ee}(\delta\_k) < \mu\_0$ **holds for all**  $a\in [K]$. However, the above informal SEE **cannot** guarantee this.
> + It is possible that at the end of phase k-1, we have $LCB\_{\hat{a}\_{k-1}}^{ee}(\delta\_{k-1}) > \mu\_0$,
> + the last collected sample of arm $\hat{a}\_{k-1}$ is so large, such that $LCB\_{\hat{a}\_{k-1}}^{ee}(\delta\_{k}) > \mu\_0$ also holds.
>
> This issue ruins the theoretical analysis of the above informal SEE. To fix this issue, we introduce a **temporary container** Q.
> + If the LUCB\_G returns arm $\hat{a}\_{k-1}$, we transfer the lasted collected sample of $\hat{a}\_{k-1}$ into Q from the exploration history $\mathcal{H}^{ee}$.
> + Notice that $LCB^{ee}\_{\hat{a}\_{k-1}}(\delta\_{k-1}) < \mu\_0$ holds **without the latest collected sample**, then at the start of the next , we can guarantee $LCB\_{a}^{ee}(\delta\_k) < \mu\_0$ **holds for all**  $a\in [K]$.
>
> This is indeed line 14-17 in our Algorithm 2.
>
> If the phase k pull $\hat{a}\_{k-1}$ again,
> + we transfer back the latest collected sample from Q to the $\mathcal{H}^{ee}$
> + as the concentration ineq (3) requires the empirical mean $\hat{\mu}\_{a,t}^{ee}=\sum_{s=1}^t X\_{a,s}^{ee}/t$ to be **consecutive summation of indexes of collected sample**. We cannot drop any collected sample.
>
> This is line 5-9 in our Algorithm 2.
>
> To rigorously formalize the above idea, we have to spend much space in showing the details. Also, we improve the concentration inequality in Kano et.al 2017, which ends up with our current section 4.
>
> There are also other ways to fix the issue, such as
> + Drop all the collected sample in previous phases, and prove independent concentration ineq for each phase
> + Use an extra concentration event, bounding $|\hat{\mu}\_{a,t}-\hat{\mu}\_{a,t-1}|$
> + Create multiple algorithm copies like Katz \& Jamieson 2020 or Chen & Li 2015
>
> But all these methods will result in one or more problems
> + lead to worse numeric performance
> + impose harsh restrictions on the parameters $C, \beta\_k, \delta\_k, \alpha\_k$
> + strictly larger logarithm factor
>
> Considering all these pros and cons, we adopt our current algorithm design.
>
> References
> + Katz & Jamieson 2020, The true sample complexity of identifying good arms
> + Chen & Li 2015, On the optimal sample complexity for best arm identification
> + Kano et.al 2017, Good arm identification via bandit feedback"
> + Jamieson & Nowak 2014, Best-arm identification algorithms for multi-armed bandits in the fixed confidence setting

---

### Official Review · Reviewer_M2Aq · 2025-03-12

**Overall Recommendation:** 3

**Summary:**

In this paper, the authors consider the identification problem a pure exploration problem with bandit feedback, where the objective is to determine where there exists an arm whose mean reward is at least a known threshold or to output None if it believes such an arm does not exist. They proposed an algorithm termed sequential-exploration-exploitation and provided a non-asymptotic analysis of the sample complexity. More precisely, they provided non-asymptotic lower and upper bounds of the expected sample complexity and proved that they match up to a constant. Finally, they conducted experiments on synthetic environments and demonstrated that the proposed method overall outperforms the baselines.

## update after rebuttal
My major concern was the poor readability of the paper. However, the authors provided a more intuitive explanation of the algorithm in their response, and I believe the paper could be improved in light of the points raised during the author-reviewer discussion phase. Therefore, I will keep my positive score.

**Claims And Evidence:**

The main contribution os this paper is non-asymptotic analysis of the sample complexity with matching lower and upper bounds. The claims are supported by Theorem 5.3 and theorems in Sec 5.2ms in Sec 5.2.

**Essential References Not Discussed:**

To the best of my knowledge, related works are adequately discussed.

**Experimental Designs Or Analyses:**

As I wrote above, the experimental design is standard for a theoretical paper.

**Methods And Evaluation Criteria:**

Theoretical results show the proposed method is $\delta$-PAC with a nearly optimal sample complexity. Although the experiments use synthetic environments, considering the theoretical nature of the paper, I believe it is a standard experimental setting.

**Other Comments Or Suggestions:**

Related to the second weakness, is it possible to add theoretical properties (as propositions or lemmas) of each sub algorithms instead of providing very detailed descriptions of the algorithms?

**Other Strengths And Weaknesses:**

- Strengths
    - They provided a non-asymptotic analysis of the sample complexity and proved that the proposed algorithm is nearly optimal.
    - Experimental results on several problem instances support the effectiveness of the proposed method.
- Weaknesses
    - As far as I understand, the lower bound provided in Theorem 5.5 holds only for some problem instances $\nu$. Also, the same theorem requires the condition $\mu_1 > \mu_0 \ge \mu_2 \ge \dots \mu_K$ which does not cover all cases of positive instances.
    - The readability of the paper could be improved. For instance, after Theorem 5.3, the authors provided a sketch of a proof over a half page. It is partially due to the short review period, but, I was unable to follow technical details. Would it be possible to move technical details to the appendix and focus on the main idea of the algorithms and proofs?

**Questions For Authors:**

1. By the statement of Theorem 5.5, there exists a positive instance $\nu$ such that the inequality holds.  Does the lower bound provided in Theorem 5.5 hold whenever $\mu_1 > \mu_0 \ge \mu_2 \ge \dots \mu_K$ holds, or does it hold for only specific instances?  Such an analysis is standard for related problem settings (e.g. BAI problem)?

**Relation To Broader Scientific Literature:**

I believe that the 1-identification problem is practically important, and an optimal algorithm for it would be beneficial for even outside the ML community.

**Theoretical Claims:**

I have not checked proofs. A sketch of a proof is provided after Theorem 5.3, I have not checked the correctness of it.

---

> ### Author Rebuttal · Authors · 2025-03-27
>
> Thank you very much for your suggestions. Let us respond below to the points raised, and we look forward to discuss more in the next iteration.
>
> 1. "If $\mu_1>\mu_0\geq\mu_2\geq...\geq\mu_K$, does the lower bound work for all permutation of [K]"
>
>      We cannot expect the current lower bound holds for any permutation of [K], but we can only guarantee there exists a permutation such that our lower bound holds.  Consider a "naive" algorithm which keeps pulling arm 1 until
>     + LCB_1>$\mu_0$ holds or
>     + UCB_1<$\mu_0$ holds.
>
>     If UCB_1<$\mu_0$, it turns to keep pulling arm 2 until similar conditions hold. Then if the mean reward of arm 1 is indeed $\mu_1$, its pulling complexity is $\Theta(\frac{\log\frac{1}{\delta}}{\Delta^2_{0,1}})$. The lower bound $\Omega\left(\frac{\log\frac{1}{\delta}}{\Delta_{0,1}^2} + \sum_{a=2}^K\frac{1}{\Delta_{1,a}^2}\right)$ does not hold. But we can "punish" this naive algorithm by considering another permutation. Let the mean reward of arm K is $\mu_1$, then this algorithm will suffer complexity $\mathbb{E}\tau\geq \Omega\left(\sum_{a=1}^K\frac{\log\frac{1}{\delta}}{\Delta_{0,a}^2}\right)$, which is much larger than the lower bound.
>
> 2. "The lower bound if $\mu_1>\mu_0\geq\mu_2\geq...\geq\mu_K$ does not hold"
>
>      + For the instance with multiple qualified answer, i.e. $m:=|\{a:\mu_a>\mu_0\}|>1$, we only know the lower bound $\Omega(\frac{\log\frac{1}{\delta}}{\Delta_{0,1}^2}+\frac{1}{m}\sum_{a=m+1}^K\frac{1}{\Delta_{1,m}^2})$.
>      + The proof is similar to Theorem 5.5, by combining the Theorem 1 in Katz \& Jamieson 2020 and Kaufmann et.al 2016. In this case, the $\delta$-dependent part of our upper bound is nearly optimal, but the $\delta$-independent part is loose.
>      + Our current result is meaningful. Our result is the first to achieve nearly optimality of $\delta$-dependent part in this case.
>         +  Also, $\delta$ dependent and independent term in the bound are tight, for the case of unique qualified instance
>         +  **None of the existing algorithms can achieve same theoretical performance as us**.
>
> 3. "The readability of the paper could be improved."
>
>     Thanks for your suggestion. In our revision, we will focus more on explaining the intuitions and the reasons behind our algorithm design, which is carefully chosen so that our current bounds can be achieved. We welcome the reviewer to check out the 3rd point of our response to the reviewer eELW.
>
> (Primarily for addressing Reviewer d2eW) Here we also list the summary of pulling and stopping rules across different algorithms. The main difference between ours and existing work is
> + we adopt decreasing tolerance level in the UCB expression. Equivalently speaking, increasing radius of the confidence interval.
> + **we do not eliminate any arm.**
>
> Comparison of sampling rule
> + HDoC: lilHDoC pull arm by $\arg\max\_{1\leq a\leq K} \hat{\mu}\_{a,t}+\sqrt{\frac{2\sigma^2 \log t}{N\_a(t)}}$.
> + LUCB-G: pull arm by $\arg\max\_{1\leq a\leq K} \hat{\mu}\_{a,t}+\Theta(\sqrt{\frac{2\sigma^2 \log \frac{\log N\_a(t)}{\delta}}{N\_a(t)}})$.
> + APGAI: pull arm by $\arg\max\_{1\leq a\leq K} \sqrt{N\_a(t)}(\hat{\mu}\_{a,t}-\mu\_0)^+$, if $\max \hat{\mu}\_{a,t} > \mu_0$,
>
>    pull arm by $\arg\max\_{1\leq a\leq K} \sqrt{N\_a(t)}(\mu\_0-\hat{\mu}\_{a,t})^+$, if $\max \hat{\mu}\_{a,t} \leq \mu_0$,
> + Our proposed SEE:
>    + UCB rule in exploration, but the tolerance level decreases (radius of confidence interval increases) at each phase,
>    + Keep pulling the same arm in exploitation.
>
> Comparison of stopping rule
> + HDoC, lilHDoC, SEE:
>    + LCB defined by $\delta$ is above $\mu\_0$ for an arm, output qualified arm,
>    + UCB defined by $\delta$ is below $\mu\_0$ for all arms, output None.
> + APGAI:
>    + $\max\_{1\leq a\leq K} \sqrt{N\_a(t)}(\hat{\mu}\_{a,t}-\mu\_0)^+ > \sqrt{2c(t,\delta)}$,
>    + $\min\_{1\leq a\leq K} \sqrt{N\_a(t)}(\mu\_0-\hat{\mu}\_{a,t})^+ > \sqrt{2c(t,\delta)}$.
>
> The radius of our confidence interval is also defined by lil rule, but we improve the constant compared to the Jamieson et.al 2014. See appendix D.2.
>
> References
> + Katz & Jamieson 2020, The true sample complexity of identifying good arms
> + Kaufmann et.al 2016, On the Complexity of Best-Arm Identiﬁcation in Multi-Armed Bandit Models
> + Jamieson et.al 2014, lil’ UCB : An Optimal Exploration Algorithm for Multi-Armed Bandits

---

### Decision · Program_Chairs · 2025-05-01

**Decision:**

Accept (poster)

**Comment:**

This paper studies the fixed-confidence “1-identification problem,” which is one of the important problems in pure exploration in multi-armed bandits. In this problem, the algorithm is given a certain threshold value, and if it believes that there exists an arm whose mean exceeds that threshold, it outputs one such arm; if it believes none exist, it outputs “none.” The authors propose a new algorithm called SEE and provide a non-asymptotic theoretical analysis. Furthermore, they prove a lower bound and demonstrate near-optimality.

The reviewers pointed out issues with the statement and presentation of the results (Reviewers M2Aq and d2eW), the algorithm description (Reviewer eELW), and a lack of comprehensiveness in the experiments (Reviewers eELW, K8WR, and d2eW). However, it was discussed that these can be addressed through revisions in the camera-ready version.

Considering that non-asymptotic theoretical analyses have already been provided in the closely related topic of good arm identification, the contribution may seem somewhat incremental. The authors also discuss the assumption of $\mu_a \neq \mu_0$, but this appears to be a minor difference.

Nevertheless, I believe that the research topic and results are indeed important to the community and have practical implications.